# Nuclear paxillin functions as a molecular switch for alternative splicing in neurons during a critical period of brain development

Chien-Hsin Chu [1,2], Guan-Zhu Pan[1,2], Ching-Yen Tsai[1], Chen-Hsin Albert Yu[1], Hsin-Nan Lin[1], Hung-Lun Chiang [1], Chen Chen[1], Li-Ching Chen[1], Xuan-Dieu Thi Pham [1], Chien-Ling Lin [1] & Pei-Lin Cheng [1✉]

## Abstract

**During a critical period of postnatal brain development, neural circuits undergo significant refinement coincident with widespread alternative splicing of hundreds of genes, which undergo altered splice site selection for the generation of isoforms essential for synaptic plasticity. Here, we reveal that neuronal activity-dependent phosphorylation of paxillin at its serine 119 (p-paxillin[S119]) acts as a molecular switch in the nucleus for the control of alternative splicing during this period. We show that following NMDA receptor activation, nuclear p-paxillin[S119] is recruited to nuclear speckles, where it interacts with splicing factors, such as U2AFs. Neuronal paxillin expression is required for timely alternative splicing of synaptic factors, including Snap25. Young mice lacking paxillin S119 phosphorylation exhibit significantly reduced levels of Snap25-5b isoforms, impaired presynaptic function at hippocampal Schaffer collateral-CA1 synapses, and deficits in short-term learning and memory. These findings support the idea that nuclear p-paxillin[S119] is a critical mediator of alternative splicing programs in postnatal neurons that are essential for neural plasticity during a critical period of brain development.**

**Keywords** Paxillin S119 Phosphorylation; Activity-Dependent Alternative Splicing; Nuclear Speckle Localization; Critical Period Synaptic Plasticity
**Subject Categories** Chromatin, Transcription & Genomics; Development; Neuroscience

## Introduction

In mice, the postnatal brain undergoes a developmental transition in a sensitive critical period when young neurons are most responsive to environmental or experiential influences (Semple et al, 2013). The ability to control the duration and closure of this period determines the degree of brain plasticity. Rodent brains exhibit two distinct waves of mRNA alternative splicing (AS), before and during early phases of circuit formation (Jacko et al, 2018; Joglekar et al, 2021; Weyn-Vanhentenryck et al, 2018). The first, occurring around birth, influences neuronal morphology, while the second, which occurs in postnatal week 2 and coincides with the onset of experience-dependent sensitive/critical periods, modulates numerous mRNA isoforms critical for synaptic refinement and electrophysiological properties. Simultaneous onset of AS events with the opening of a critical period suggests a feedback mechanism by which initial synaptic activity facilitates production of RNA isoforms, optimizing or refining synaptic responsiveness and connectivity. In this regard, premature or delayed AS transitions in early postnatal neurons could undermine brain function.

Neuronal circuit maturation requires precisely timed transitions, potentially mediated by switch-like mechanisms that synchronize activation of AS programs. Identifying those switches remains a significant challenge. Candidate switch factors likely exhibit several key characteristics: (1) their expression/activation patterns could be spatially and temporally concurrent with opening or closing of a critical time window; (2) they may show changes in the ability to associate with spliceosomes or RNA binding proteins (RBPs), outcomes that could alter splice site selection crucial for synaptic activity in a particular brain area; or (3) their activation could be determined by neurotrophic factors and/or experience-dependent synaptic activation at the transition required for stage progression. Intriguingly, paxillin, a protein typically associated with cytoskeletal dynamics or activity of endocytic vesicles at growing axonal tips (Chang et al, 2017; Chen et al, 2022; Huang et al, 2004; Lopez-Colome et al, 2017; Tsai et al, 2021), has emerged as a potential modulator of dynamic AS in nuclei of mature neurons. While its exact function in this capacity remains elusive, evidence suggests that paxillin localizes to the nucleus and interacts with RBPs and transcriptional regulators to influence gene expression programs critical for cell proliferation and differentiation (Dong et al, 2009; Ma and Hammes, 2018; Sathe et al, 2016). Moreover, paxillin's subcellular localization and activity in functional networks are regulated via phosphorylation in a cellular context-dependent manner (Lopez-Colome et al, 2017), such as when cells enter specific phases of the cell cycle. These observations suggest

---

[1]Institute of Molecular Biology, Academia Sinica, Taipei, Taiwan. [2]These authors contributed equally: Chien-Hsin Chu, Guan-Zhu Pan. ✉E-mail: plcheng@imb.sinica.edu.tw

that paxillin expressed in mature neurons may be responsive to neurotrophins and/or neuronal activity and serve as a modulator of signals regulating gene expression sent from the periphery to the nucleus.

Here, to investigate these possibilities, we identified a brain-enriched phosphorylation site on paxillin at serine 119 (p-Paxillin$^{S119}$). Phosphorylation at this site, stimulated by BDNF and/or CDK5 activation, facilitated paxillin nuclear translocation in cultured young neurons via proline-tyrosine nuclear localization signals (NLSs). Nuclear p-Paxillin$^{S119}$ formed condensates localized to nuclear speckles, where it associated with splicing factors, suggesting direct regulation of splicing. Importantly, during sensitive periods, sensory input, such as tactile stimulation of the whiskers or NMDA receptor activation, was both necessary and sufficient to induce p-Paxillin$^{S119}$ formation and maintain it in the nucleus, with peak levels occurring in different brain regions depending on the onset of those periods. Paxillin KD in young neurons delayed the splicing switch for genes functioning in synaptic plasticity, including those encoding synaptosomal-associated protein of 25 kDa (Snap25), various Ankyrins (ANKs), NMDA receptor, and the GABAb receptor. Furthermore, RBP-eCLIP motif analysis and co-immunoprecipitation (co-IP) identified multiple RBPs, such as FUS, ELAVL4, NOVA1/2, matrin3, and Rbfox2, that physically interact with paxillin in neurons. Notably, neonatal mice lacking paxillin$^{S119}$ phosphorylation starting at postnatal week 2 exhibited delayed Snap25 isoform switching across various brain regions, altering electrophysiological properties and impairing neural plasticity. Collectively, these findings demonstrate that activity-dependent paxillin S119 phosphorylation modulates AS programs crucial for proper brain plasticity.

# Results

## Spatiotemporal distribution of a brain-enriched form of phosphorylated paxillin$^{S119}$ in neurons

We previously reported that paxillin expressed in newborn mouse neurons functions via mechanisms distinct from those seen in other adherent cells (Chang et al, 2017; Chen et al, 2022). In light of these functional differences, we hypothesized that differential post-translational modifications—specifically phosphorylation—might contribute to the unique neuronal activity of paxillin. To identify variants of phosphorylated paxillin specifically enriched in neurons or brain tissue, we immunoprecipitated (IP'd) paxillin from postnatal day 14 (P14) mouse brain, heart, and liver tissues and subjected precipitates to phosphopeptide enrichment and mass spectrometry-based phospho-proteomic analysis (Fig. EV1). Among numerous paxillin phosphorylation sites identified, phospho-serine 119 was specifically enriched in brain samples, and that motif and surrounding sequences were conserved across vertebrates (Fig. EV1A). Western blotting with phospho-serine119-specific antibodies confirmed enrichment of phospho-paxillin at serine119 (p-Paxillin$^{S119}$) in mouse brain tissues (Fig. EV1C–E). By contrast, phosphorylation of the adjacent tyrosine 118 (p-Paxillin$^{Y118}$), a known focal adhesion kinase phosphorylation site, was seen across various tissues (Fig. EV1C–E). We validated the specificity of antibodies targeting p-Paxillin serine119 using a conditional mouse line harboring a floxed stop cassette followed by a mutant form of paxillin with a serine-to-alanine substitution at position 119

(S119A), a change driven by tamoxifen-inducible nestin-CreEr$^{T2}$ recombinase (Appendix Fig. S1A). Tamoxifen administration at postnatal day 5 (P5), P7, and P9 promoted loss of nuclear p-Paxillin$^{S119}$ staining in P60 NeuN-positive neurons from Nestin-CreEr$^{T2}$; paxillin$^{S119A/fl}$ mice (see Appendix Fig. S1A). Western blotting of lysates from Neuro2a cells ectopically expressing FLAG-tagged paxillin variants confirmed that the affinity of p-Paxillin$^{S119}$ antibodies for phosphomimetic paxillin$^{S119D}$ was significantly higher than that for WT or phosphorylation-deficient (S119A) paxillin (Appendix Fig. S1E).

Next, we analyzed spatiotemporal expression patterns of brain-enriched p-Paxillin$^{S119}$ in primary neuronal cultures using p-Paxillin$^{S119}$-specific antibodies (Figs. 1 and EV2). Analysis of p-Paxillin$^{S119}$ staining in cultured cortical neurons at various developmental stages [from day 3 to day 14 in vitro (DIV3–DIV14); Fig. 1A–C] revealed a distinct nuclear speckle pattern colocalizing with staining for splicing factors, such as SC-35 and phosphor-Ser-Arg-rich (SR) proteins, but not with Histones (Fig. 1D). These patterns emerged during a defined period in cultured neurons, from baseline levels prior to DIV3, to levels increasing and peaking around DIV7, and then to a decline in levels before DIV14 (Figs. 1A–C and EV2). By contrast, the nuclear content of p-Paxillin$^{Y118}$ or other serine-phosphorylated paxillin variants (such as S83, S178, or S272) remained low or showed no significant change over the same time period in cultured neurons (Fig. EV2). Such time-dependent nuclear patterns of p-Paxillin$^{S119}$ were not seen in non-neuronal cells in the same primary cultures (Fig. EV2C), suggesting they are specific for differentiated neurons.

## Paxillin phosphorylation at serine119 permits its nuclear translocation in mature neurons

Next, we employed pharmacological and genetic strategies to characterize mechanisms regulating p-Paxillin$^{S119}$ nucleocytoplasmic transport, first by asking whether blocking nuclear export would prolong p-Paxillin$^{S119}$ nuclear accumulation after DIV7, a period when nuclear p-Paxillin$^{S119}$ levels peak. To do so, we treated primary neuronal cultures at DIV7 with one of two nuclear export signaling (NES) inhibitors, 10 nM Leptomycin B (LMB) or 500 nM KPT-330, and then assessed nuclear p-Paxillin$^{S119}$ levels at DIV10, when they normally decline. As shown in Fig. 1E,F, neither LMB nor KPT-330 significantly altered nuclear levels of p-Paxillin$^{S119}$ in DIV10 neurons relative to untreated cultures. By contrast, pretreating DIV5 neurons 3 h with importazole (5 μM; IPZ), which inhibits nuclear import, was sufficient to decrease nuclear p-Paxillin$^{S119}$ levels in cortical neurons on DIV7 relative to those seen in untreated cultures, suggesting that paxillin harbors NLSs (Fig. 1G–J; Appendix Fig. S2). We next asked whether paxillin serine119 phosphorylation facilitated its association with nuclear-import receptor β1, a component of the nuclear import machinery, by performing co-IP of lysates of differentiated Neuro2a cells expressing FLAG-tagged paxillin variants, such as phospho-deficient (S119A) or phosphomimetic (S119D) forms (Fig. 1K). In the presence of the cAMP-elevating agent forskolin, we found that co-IP'd importin β1 levels were significantly lower in the presence of the S119A variant relative to wild-type (WT) or S119D paxillin, indicating that paxillin$^{S119}$ phosphorylation is crucial for importin β1 interaction (Fig. 1K). Conversely, pretreating DIV3 or DIV5 cortical neurons with LB100, an inhibitor of serine/threonine protein phosphatase 2A (PP2A), significantly increased both cytoplasmic and nuclear levels of

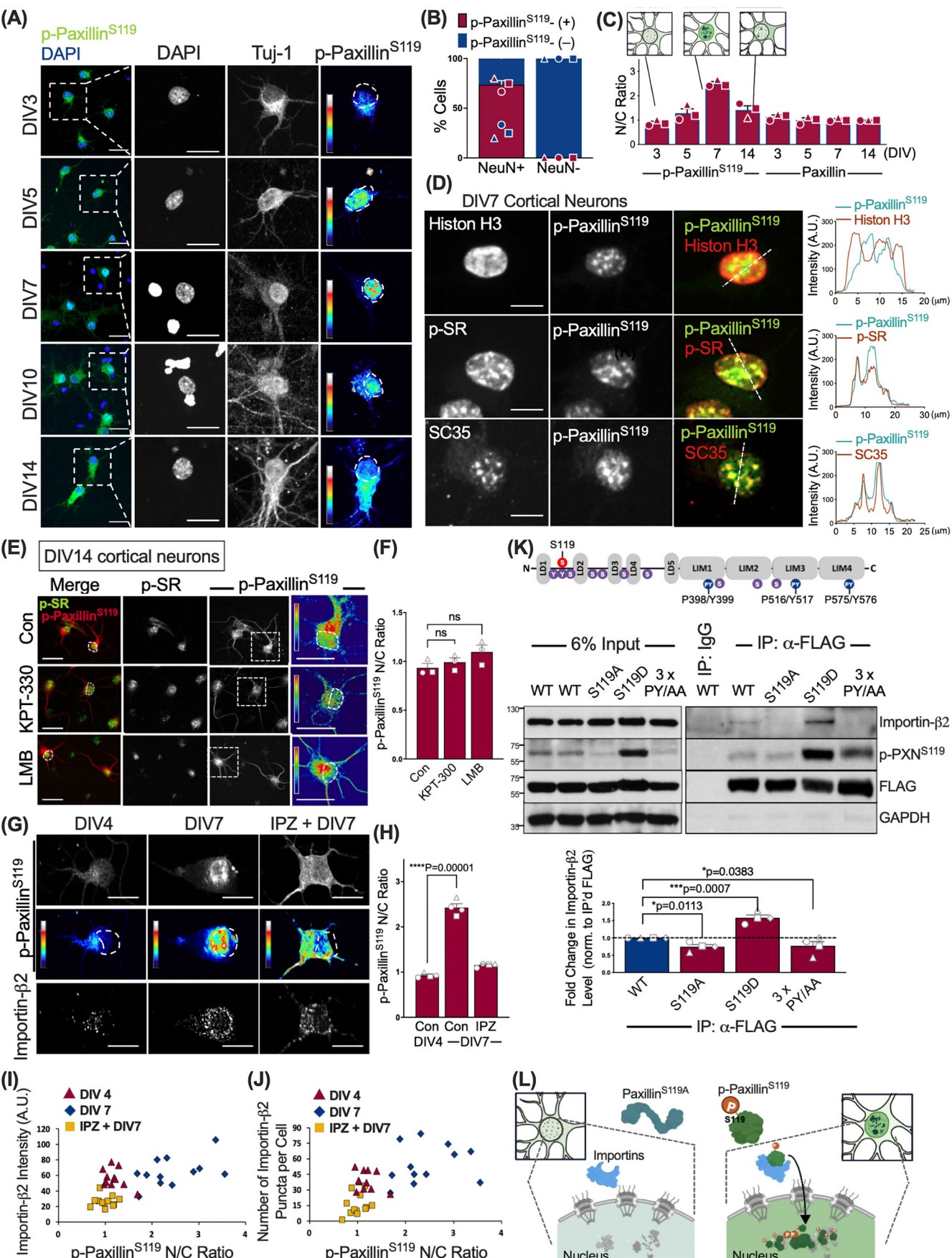

**Figure 1.  p-Paxillin^S119 localizes to distinct nuclear speckles localization in rat primary cultured neurons at DIV7.**

(A–C) Subcellular distribution of endogenous p-Paxillin^S119 in newly polarized cortical neurons over the first two weeks of in vitro culture. (A) Representative images of primary cortical cultures grown in standard neurobasal medium and immunostained for p-Paxillin^S119 (green) and the neuronal marker Tuj-1, with DAPI counterstain (blue) at the indicated time points. Scale bar, 20 µm. (B, C) Characterization of paxillin S119 phosphorylation patterns in primary neuronal cultures. Histogram showing percentage (± SEM, n = 3 cultures) of NeuN-positive (neuronal) and -negative (non-neuronal) cells exhibiting p-Paxillin^S119 nuclear accumulation at DIV7. (C) Histogram summarizing nuclear to cytoplasmic (N/C) ratio of p-Paxillin^S119 and paxillin in primary neuronal cultures at the indicated time points. Data represent mean ± SEM (n = 3 cultures, >30 cells per group). (D) p-Paxillin^S119 colocalizes with RBPs in nuclear speckles in DIV7 neurons. Immunofluorescence images show staining for p-Paxillin^S119, histone H3, SRRM2 (SC-35), and phospho-SR proteins (p-SR). Fluorescence intensity profiles (right panel) across p-Paxillin S119 puncta show colocalization of p-Paxillin^S119 with SC-35 and p-SR, but not with histone H3. Scale bar, 10 µm. (E–J) Nuclear localization of p-Paxillin^S119 requires importin β2 activity. Primary cortical cultures were treated with the nuclear export inhibitors Selinexor ("KPT-330"; 500 nM for 16 h at DIV7) or leptomycin B ("LMB", 10 nM for 16 h at DIV7; E, F) or the nuclear import inhibitor importazole ("IPZ", 5 µM for 3 h at DIV5; G, H) and immunostained for p-Paxillin^S119, p-SR, and importin β2 at indicated time points. (F, H) Histogram showing the N/C ratio (± SEM; n = 3–4 cultures; >20 cells per group; ***p < 0.001, ns not significant by multiple unpaired t-tests) in neurons from experiments shown in (E, G). (I, J) Plots showing correlation of N/C ratio of p-Paxillin^S119 with importin β2 intensity (I) and puncta density (J) in neurons under the indicated conditions. (K) Serine119 phosphorylation and PY-NLS motifs facilitate paxillin interaction with importin β2. Western blots of FLAG immunoprecipitates from Neuro2A cells transfected with FLAG-tagged wild-type paxillin (WT) or indicated mutants. Immunoprecipitates were analyzed by immunoblotting with antibodies against importin β2, p-Paxillin^S119, and FLAG. A schematic of putative phosphorylation sites and PY-NLS motifs on paxillin protein is shown above. Histograms summarize quantitative measurements of fold changes (± SEM; n = 3 independent experiments; *p < 0.05, **p < 0.01, ***p < 0.001 compared to WT control group by multiple t-tests) in importin β2 binding. 3x PY/AA refers to a triple PY to AA mutant (paxillin^P318A/Y319A, P517A/Y518A, P575A/Y576A). (L) Schematic illustrating p-Paxillin^S119 interactions with importins, which, in turn, facilitates the process of nuclear translocation and complex formation with nuclear proteins. Created with BioRender. Source data are available online for this figure.

p-Paxillin^S119 when assessed at DIV7 (Appendix Fig. S2A), likely due to sustained S119 phosphorylation.

There are no reports of a canonical paxillin NLS. Thus, using silico predictions of an atypical NLS, we identified 3 potential proline-tyrosine (PY)-type NLSs at positions P398/Y399, P516/Y517, and P575/Y576, and then assessed their function along with S119. Neuro2a cells ectopically expressing GFP-tagged WT paxillin exhibited nuclear translocation of paxillin upon differentiation induced by serum starvation in the presence of forskolin (Appendix Fig. S3). Moreover, transduction of Neuro2a cells with alanine mutants of each PY pair (PY to AA) in GFP-paxillin constructs decreased paxillin nuclear-to-cytoplasmic (N/C) ratios, suggesting that these NLSs are functional (Appendix Fig. S3). We also observed a significant reduction in nuclear levels of serine119 phosphorylation-deficient alanine mutants (paxillin^S119A) relative to the WT protein, suggesting that paxillin serine 119 phosphorylation is crucial for its nuclear translocation (Appendix Fig. S3). Accordingly, we asked whether PY-NLS is required for paxillin to associate with the nuclear importer importin-β2 by performing in vivo co-IP of ectopically-expressed FLAG-tagged WT paxillin and its variants. We found that IP'd paxillin harboring AA at all three PY sites in the NLSs ("3x PY/AA") exhibited significantly reduced paxillin binding to importin-β2 than did the WT protein. A similar reduction in binding was observed for the S119A variant, whereas mutation of single PY sites to AA did not significantly alter importin-β2 binding (Fig. 1K). These findings suggest that paxillin phosphorylation at S119 permits importin-β2/paxillin interaction, enabling paxillin nuclear translocation in young neurons (Fig. 1L).

## Signals associated with neuronal maturation promote paxillin phosphorylation at S119

The timing (DIV5–DIV10) of paxillin^S119 phosphorylation in cultured primary neurons aligned with axon maturation (Fig. 1), suggesting that paxillin^S119 serves as a substrate for serine/threonine kinases driving axon development, such as those involving autocrine BDNF signaling and CDK5 activation. To determine whether cAMP/PKA, CDK5/p25, or BDNF/TrkB signaling pathways regulate paxillin^S119 phosphorylation, we employed pharmacological inhibitors and in vitro kinase assays. In DIV14 cortical

neuronal cultures—in which nuclear p-Paxillin^S119 levels had already declined—bath application of forskolin or BDNF significantly elevated nuclear p-Paxillin^S119 staining compared to untreated controls (Fig. 2A,B). In contrast, in DIV7 cultures—in which nuclear p-Paxillin^S119 levels are still increasing—treatment with inhibitors of PKA, TrkB, or CDK5 abolished nuclear p-Paxillin^S119 (Fig. 2C,D).

Consistent with the observations, western blot analysis of paxillin IP'd from Neuro2a cells ectopically expressing FLAG-tagged WT paxillin revealed that phosphorylation at S119, but not Y118, was significantly higher in Neuro2A cells treated with forskolin or BDNF compared to untreated controls (Fig. 2E,F), an increase abolished when cells were pretreated with the PKA inhibitor KT5720 or the TrkB inhibitor K252a (Fig. 2E,F). We next asked whether CDK5/p25 activity alone was sufficient to promote paxillin S119 phosphorylation using in vivo CDK5 kinase assays of Neuro2A cells co-expressing FLAG-tagged paxillin variants, CDK5, and/or CDK5 coactivators. As shown in Fig. 2G,H, S119 phosphorylation levels of IP'd WT paxillin, but not the S119A variant, significantly increased in Neuro2a cells overexpressing CDK5 and its coactivator p25, suggesting that activated CDK5 phosphorylates paxillin at serine119. Conversely, the CDK5-dependent increase in p-Paxillin^S119 levels was eliminated when we co-expressed a dominant-negative form of p25 (Fig. 2G,H). These findings support the idea that paxillin is a potential substrate of these kinases.

## Synaptic activity sustains nuclear Paxillin^S119 phosphorylation in vitro and in vivo

In cultured primary neurons, nuclear p-Paxillin^S119 is transiently upregulated in the second week (DIV5–DIV10) and then declines before DIV14, indicating that nuclear localization is not constitutive in vitro (Fig. 1A–C) and suggesting that maintaining p-Paxillin S119 in the nucleus requires synaptic stimulation, which is minimal in most neuronal culture systems. To investigate this possibility, we treated DIV14 cultures of primary neurons with NMDA and glycine to activate the NMDA receptor and observed significantly increased levels (>2-fold increases relative to the group treated with regular medium; Fig. 2A,B,I,J) of nuclear p-Paxillin^S119 with a clear speckle staining pattern. By contrast, this NMDA-induced effect was not seen in comparable analysis in the presence of the NMDA

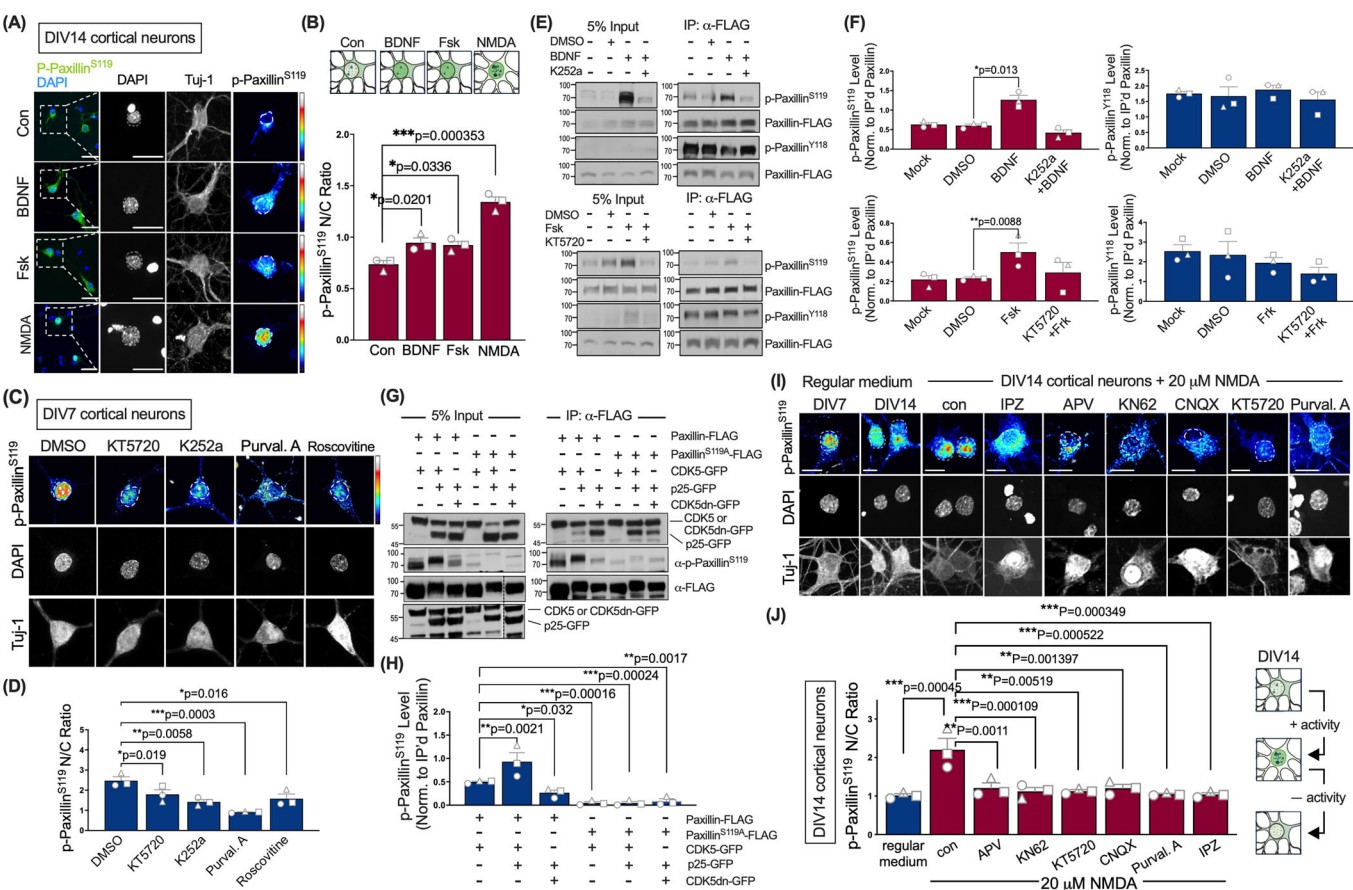

**Figure 2. Neuronal maturation signaling and synaptic activity induce paxillin S119 phosphorylation and nuclear speckle localization.**

(A) Images of DIV14 cortical neuronal cultures exposed to 20 ng/ml BDNF, 10 μM forskolin ("Fsk"), or 20 μM NMDA, as indicated, followed by immunostaining for p-Paxillin[S119] and Tuj-1, with DAPI counterstain. Scale bar, 20 μm. (B) Quantitative assessment of the p-Paxillin S119 N/C ratio in neurons from experiments shown in (A). Data represent mean ± SEM ($n = 3$ cultures; 27–35 cells per group; *$p < 0.05$, ***$p < 0.001$, compared to control, one-way ANOVA followed by Tukey's multiple comparisons test). Top panel, schematic illustrating representative p-Paxillin[S119] (green) staining patterns in neurons under the indicated conditions. (C, D) Serine/threonine kinase activity of CDK5, BDNF/TrkB, or cAMP/PKA is required to enhance paxillin serine119 phosphorylation and nuclear translocation in DIV7 neurons. (C) Similar to (A), except DIV7 cortical neurons were treated with the PKA inhibitor KT5720 (5 nM), the TrkB inhibitor K252a (200 nM), or the CDK5 inhibitors Purvalanol A (10 μM) or Roscovitine (10 μM) for 30 min, as indicated. Scale bar, 5 μm. (D) Histogram summarizing N/C ratio of p-Paxillin[S119] in DIV7 neurons from experiments, as shown in (C). Data represent mean ± SEM ($n = 3$ cultures; 27–50 cells per condition; *$p < 0.05$, **$p < 0.01$, ***$p < 0.001$, compared with the DMSO control, one-way ANOVA followed by Dunnett's multiple comparisons test). (E–H) Serine/threonine kinase activity of CDK5, BDNF/TrkB, or cAMP/PKA signaling is sufficient to induce paxillin phosphorylation at S119. (E, F) In vivo kinase assays were performed using Neuro2A cells pretreated with or without the TrkB inhibitor K252a or the PKA inhibitor KT5720, in the presence or absence of 20 ng/ml BDNF or 10 μM forskolin (fsk), as indicated. (F) Quantitation of p-Paxillin[S119] and p-Paxillin[Y118] levels in FLAG immunoprecipitants. Data represent mean ± SEM ($n = 3$ biological replicates; *$p < 0.05$, **$p < 0.001$, compared with the control DMSO group by multiple $t$-test). (G, H) Similar to (C, D), except Neuro2A cells were expressing FLAG-tagged wild-type paxillin, FLAG-tagged S119 phosphorylation-deficient paxillin[S119A], GFP-tagged CDK5, GFP-tagged dominant-negative CDK5 ("CDK5dn"), and/or GFP-tagged p25, as indicated. Immunoblots of FLAG immunoprecipitants show increased p-Paxillin[S119] levels when CDK5 and coactivator p25 are co-expressed, but not with CDK5dn. Data represent mean ± SEM ($n = 3$ biological replicates; *$p < 0.05$, **$p < 0.01$, ***$p < 0.001$, compared with the CDK5 alone group by multiple unpaired $t$-tests). (I, J) Activation of neurons by NMDA is required for p-Paxillin[S119] nuclear localization in DIV14 neurons. (I) similar to (A, C), except DIV14 cortical neuronal cultures exposed to 20 μM NMDA were pretreated with inhibitors of importin-β (5 μM IPZ), NMDA receptor (50 μM APV), CaMK-II (3 μM KN62), AMPA/kainate receptor (10 μM CNQX), cAMP/PKA (5 nM KT572), or CDK (10 μM Purval. A), as indicated. Scale bar, 10 μm. (J) Quantitative assessment of the p-Paxillin S119 N/C ratio in neurons from experiments shown in (I). Data represent mean ± SEM ($n = 3$ cultures; 25–35 cells per condition; **$p < 0.01$, ***$p < 0.001$, compared to NMDA-treated control, one-way ANOVA followed by Dunnett's multiple comparisons test). Right panel, schematic illustrating representative p-Paxillin[S119] (green) staining patterns in neurons under the indicated conditions. Source data are available online for this figure.

receptor antagonist APV, the AMPA/kainate receptor antagonist CNQX, calmodulin-dependent protein kinase II (CaMK-II) inhibitor KN62, or the importin inhibitor IPZ (Fig. 2I,J). These findings support the idea that evoked neuronal activity mediates increases in nuclear p-Paxillin[S119] in DIV14 neurons.

To test whether this responsiveness is confined to an early developmental phase, we further examined NMDA-induced nuclear translocation of p-Paxillin[S119] in older cultures. Consistent with a

critical window, neurons at DIV28 showed minimal or no nuclear p-Paxillin[S119] accumulation following either acute (25-min) or prolonged (6-h) NMDA stimulation (Fig. EV3). While nuclear p-Paxillin[S119] levels peaked at DIV7 and were strongly induced by NMDA stimulation at DIV14, CDK5 remained predominantly cytoplasmic and its nuclear-to-cytoplasmic ratio did not mirror this pattern (Fig. EV3), suggesting that S119 phosphorylation directly drives Paxillin nuclear import via importin β2 rather than via CDK5

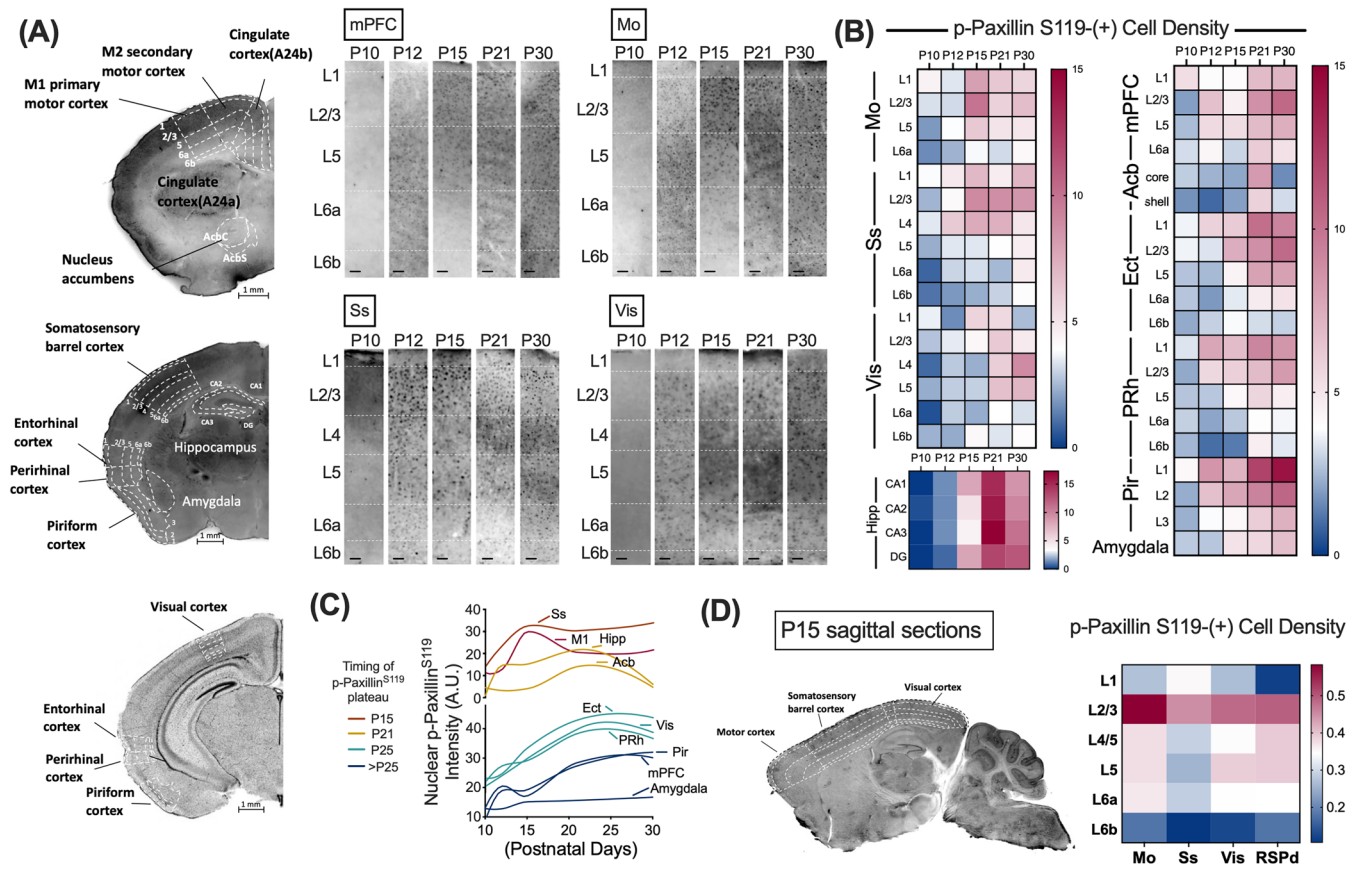

**Figure 3. Developmental upregulation of paxillin S119 phosphorylation in mouse brain.**

(A) Immunohistochemistry of coronal sections of mouse cortex showing p-Paxillin[S119] staining at the indicated postnatal days. (B) Quantitation of density of p-Paxillin[S119]-positive neurons in coronal sections. Heatmap shows the percentage ($n = 3$ cortices per brain area) of p-Paxillin[S119]-positive neurons in the indicated brain regions, at the indicated postnatal days. mPFC medial prefrontal cortex, Ect ectorhinal cortex, PRh perirhinal cortex. Pir piriform cortex. (C) Akima spline fits of average p-Paxillin[S119] intensity. Traces ($n = 3$ cortices per brain area) show staggered intensity peaks across brain regions at P15 (red), P21 (yellow), P25 (green-blue), and P30 (dark blue. (D) Immunohistochemistry of sagittal mouse P15 cortical sections. Heatmap on the right shows a high-to-low gradient of p-Paxillin[S119] density ($n = 3$ cortices) from motor ("M1"), somatosensory cortex ("Ss"), and visual cortex ("Vis") to dorsal retrosplenial cortex ("RSPd").

nuclear entry. Together, these findings demonstrate that NMDA receptor activation maintains levels of nuclear p-Paxillin[S119] within a defined, activity-sensitive period during neuronal maturation.

Accordingly, we asked whether tactile sensory input would increase p-Paxillin[S119] levels. To do so, we used a neonatal whisker trimming approach to induce unilateral sensory deprivation by plucking all whiskers in row C on one side. Five days later at P21, we assessed neuronal activity (based on c-Fos expression) and p-paxillin S119 levels in the barrel cortex using fluorescence immunohistochemistry. We observed a reduction in both c-Fos expression and paxillin S119 phosphorylation specifically within the row C region of the barrel cortex contralateral to the whisker trimming (Appendix Fig. S4), indicating that expression of both is dependent on sensory input and associated neural activity.

We next asked whether paxillin[S119] phosphorylation in postnatal brain tissue is concurrent with initiation of physiological synaptic input. To do so, we assessed temporal phosphorylation of paxillin S119 in mouse brain in vivo by co-staining with the nuclear stain DAPI plus antibodies against p-Paxillin[S119] and/or NeuN, a marker of post-mitotic neurons, at various time points in the first postnatal

month. As shown in Fig. 3A,B; Appendix Fig. S5, while pan-Paxillin expression remained relatively constant from P10 to P30, nuclear p-Paxillin[S119] was seen in neurons in various brain regions at different times after postnatal day 10 (P10) and was detectable up to P30. We observed that p-Paxillin[S119] peaks in motor and somatosensory cortices by P15 (Fig. 3C,D), in the amygdala, hippocampal dentate gyrus, and CA3 by P21, and in the visual cortex by P30 (Fig. 3C). These patterns reflect that sensitive periods of plasticity are staggered across each region.

## Serine 119 phosphorylation modulates paxillin interaction with splicing factors

In rodents, postnatal week 2 begins a sensitive period when numerous genes associated with neuronal plasticity exhibit altered splice site usage (Jacko et al, 2018; Semple et al, 2013; Weyn-Vanhentenryck et al, 2018). During this period, p-Paxillin[S119] colocalizes with phosphor-SR splicing factors and active spliceosome-enriched nuclear speckles in young neurons (Fig. 1D). Thus, we asked whether nuclear paxillin modulates these splicing events using biochemical and RNA-sequencing

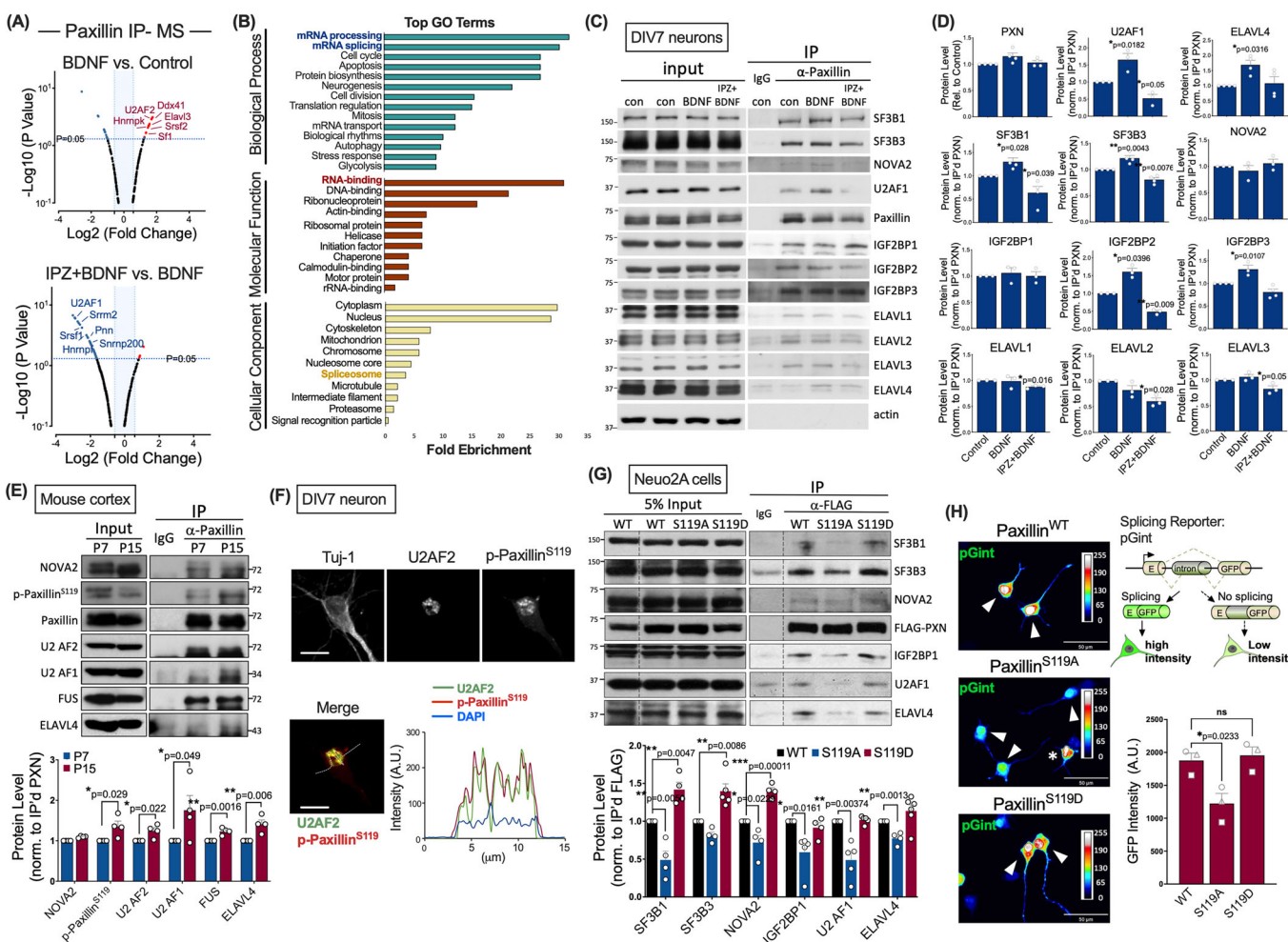

**Figure 4. Paxillin[S119] phosphorylation facilitates paxillin interaction with splicing factors.**

(A–D) BDNF treatment and nuclear import signaling promote paxillin interaction with splicing factors. (A) Paxillin-associated proteins IP'd from lysates of DIV7 neuronal cultures pretreated with or without the nuclear import inhibitor IPZ (5 μM, 16 h), and then stimulated with BDNF (20 ng/ml, 2 h), followed by LC-MS/MS analysis. Volcano plots showing log2-fold change in abundance ratio of identified paxillin-associated proteins (y-axis) plotted against statistical significance (−log10 $p$ value, x-axis). The top panel compares BDNF-stimulated to control cultures; the bottom panel compares IPZ-pretreated/BDNF-stimulated to BDNF-stimulated cultures. (B) Gene Ontology (GO) term enrichment analysis was performed on paxillin-bound proteins whose levels decreased (cut-off ≤0.78) in IPZ-pretreated/BDNF-stimulated relative to BDNF-stimulated cultures. (C) Western blots showing levels of paxillin IP'd proteins upon indicated treatments. (D) Quantification of data shown in blots in (C). Data represent mean ± SEM from three independent experiments. *$p < 0.05$, **$p < 0.01$, compared with the control group by multiple unpaired $t$-tests. (E) Similar to (C), except P7 and P15 brain lysate was used. (F) Image showing a DIV7 primary cultured neuron immunostained with antibodies against U2AF2, p-paxillin[S119], or Tuj-1. Intensity traces along the white dashed line show punctate staining of U2AF2 and p-Paxillin[S119] colocalized in the nucleus. Scale bar, 10 μm. (G) Similar to (C), except lysates of Neuro2A cells transfected with FLAG-tagged paxillin variants were IP'd with FLAG antibody. Histogram quantifies FLAG IP data ( ± SEM, $n = 4$ cultures; normalized to the corresponding wild-type ("WT") paxillin group; *$p < 0.05$, **$p < 0.01$ by multiple unpaired $t$-tests). (H) mRNA splicing efficiency is altered in differentiated Neuro2A cells ectopically expressing paxillin[S119A]. A schematic of the GFP splicing reporter (pGint) is shown above. Representative images depict differentiated Neuro2A cells co-transfected with pGint and plasmids encoding either paxillin wild-type (WT) or its variants. GFP fluorescence intensity is displayed using a "royal" look-up table, highlighting the range from low (blue) through medium (green) to high (red) intensity. Arrowhead, paxillin-expressing cell. Asterisk, paxillin[S119A]-nonexpressing cell. Scale bar, 50 μm. Histogram summarizes average GFP intensity ( ± SEM; $n = 3$ cultures; 21–30 cells per group; ****$p < 0.0001$ compared to WT, one-way ANOVA followed by Tukey's multiple comparisons test). Source data are available online for this figure.

approaches. To define factors interacting with paxillin during this period, we IP'd paxillin and performed proteomic analysis using liquid chromatography-tandem mass spectrometry (LC-MS/MS) in both nuclear and cytoplasmic fractions of DIV7 cultured neurons treated with BDNF to promote paxillin phosphorylation and/or pretreated with IPZ to prevent p-Paxillin[S119] nuclear translocation (Fig. 4A). We then performed comparative analysis of proteins enriched in the nucleus and dependent on nuclear transporter activity [cut-off: (IPZ pretreatment

+BDNF)/(BDNF) <0.78). Among identified paxillin-interactors, we observed significant enrichment of Gene Ontology (GO) terms related to factors involved in RNA splicing and processing (Fig. 4B). To validate interactions, we performed co-IP analysis in primary neuronal cell lysates or brain lysates of paxillin at different time points (Fig. 4C–E) and observed enhanced paxillin association with splicing factors, such as SF1, U2AF1, U2AF2, HNRNPK, SRSFs, and ELAVL4, following BDNF treatment of cultured neurons or in P15 brain lysates, compared to that

seen in respective untreated controls or P7 lysates. By contrast, paxillin interaction with RBPs significantly decreased when nuclear import was blocked by IPZ treatment in cultured neurons (Fig. 4C,D), suggesting that these interactions occur in the nucleus. Physical association of p-Paxillin[S119] with U2AF complexes in nuclear speckles was confirmed by co-immunofluorescent staining of U2AF2-p65 and p-Paxillin[S119] in DIV7 neuron cultures: as shown in Fig. 4F, the condensed/punctate-like pattern of p-Paxillin[S119] staining highly overlapped with that of U2AF2-p65.

To further assess whether paxillin/splicing factor interactions require S119 phosphorylation, we performed in vivo co-IP in lysates of Neuro2a cells expressing Flag-tagged forms of WT or phospho-deficient (S119A) paxillin in the presence of forskolin to facilitate PKA-mediated serine phosphorylation (Fig. 4G). Anti-Flag IP confirmed that WT paxillin forms complexes with various RBPs, and that the S119A variant showed reduced binding affinity for SF3B1, IGF2BP1, U2AF1, and ELAVL4, suggesting that serine 119 phosphorylation enhances paxillin interaction with these factors.

To validate whether S119 phosphorylation is sufficient to alter mRNA splicing efficiency, we employed a GFP-based splicing reporter (pGint) (Bonano et al, 2007) that contains an intron between split GFP cDNAs; higher GFP intensity signifies more efficient splicing. Differentiated Neuro2A cells were co-transfected with pGint and plasmids encoding paxillin WT or S119A/D variants. As shown in Fig. 4H, cells expressing the S119A mutant exhibited lower GFP intensity relative to those expressing WT or S119D, suggesting reduced splicing efficiency. These findings support the notion that S119 phosphorylation enhances both the physical and functional interplay between paxillin and the splicing machinery. To determine whether Paxillin's ability to regulate splicing depends on the actin cytoskeleton, we treated Neuro2A cells and DIV7 primary neurons with Latrunculin A and observed robust disruption of the cytoplasmic actin network, based on phalloidin staining (Fig. EV4). However, neither the pGint reporter signal nor Paxillin's nuclear speckle localization was altered by actin disruption (Fig. EV4). Notably, expression of the S119A mutant reduced pGint splicing in the presence or absence of Latrunculin A treatment. These results suggest that S119 phosphorylation promotes Paxillin's splicing regulatory activity, which functions largely independently of cytoplasmic actin dynamics.

## Accurate timing of splice site switching requires paxillin expression

Given that paxillin physically interacts with splicing components in vitro and in vivo, we asked whether neuronal paxillin expression regulates the timing of AS programs. To do so, we performed RNA sequencing (60+ million reads/sample) and rMATs analysis of cortical cultures with and without paxillin KD (siRNA) and/or BDNF treatment to assess changes in exon inclusion. We then determined percent spliced-in (PSI) values for cassette exons at key developmental stages (DIV0/E15, DIV3, DIV5, and DIV10) to monitor splice site switching transitions in vitro (Figs. 5A and EV5). Splice site usage and cassette exon patterns in these cultures closely mirrored cortical maturation in vivo, with DIV3, DIV5, and DIV10 correlating with P0, P4-P7, and P15+ cortices, respectively (Fig. EV5A). We then analyzed changes in PSI ($\Delta$PSI or $\Delta\psi$) at these intervals to identify increases or decreases for each cassette exon and delineate patterns of splice site switching. Comparison of these patterns in scrambled control (SC)-siRNA

versus paxillin KD groups allowed us to categorize cassette exons into four groups—Early, Late, Reversed, and Unaffected—based on responses to paxillin KD and reflective of acceleration, delay, reversal or no effect switch timing (Figs. 5A and EV5B). Analysis of the size of these groups revealed that paxillin KD primarily delayed the timing of AS events, while BDNF treatment accelerated inclusion of alternative cassette exons (Fig. EV5C). Moreover, 1607 of 3334 cassette exons displayed altered splicing patterns ("switching time points") after paxillin KD, with 290 in "Early", 974 in "Late", and 343 in "Reversed" switching groups. We observed significant enrichment of GO terms related to cell component functioning in synapses and synaptosomes among identified paxillin KD-affected late or reversed cassette exons (Fig. 5B). To confirm the functional relevance of observed splicing changes, we isolated synaptosome fractions from hippocampus of Nestin-CreERT2; Paxillin[S119A/fl] and control mice and found that levels of protein variants of Glutamate Receptor NMDAR1 (NR1) containing N1 and C1 domains decreased in Nestin-CreERT2; Paxillin[S119A/fl] relative to control mice (Appendix Fig. S6A,B).

We next asked which trans-acting RBPs associate selectively with splicing events altered by paxillin KD. To do so, we analyzed cassette exons affected by paxillin KD and the adjacent 200-base pairs of intronic sequences (5' and 3') for RBP-binding site enrichment (Figs. 5 and EV5B). To avoid ambiguities in RBP region annotation, we focused our analysis on 46 RBPs with eCLIP data available from oRNAment and CLIPdb. Using Fisher's exact test to assess statistical significance, we refined candidate RBPs by cross-referencing with those identified in our paxillin IP/mass spectrometry (IP-MS/MS) analysis (Figs. 4 and EV5C,D). Applying these criteria, we found that the neuron-enriched RBPs NOVA1/2 and Rbfox2 and the core splicing factor U2AF2 were most significantly associated with isoform switching events altered by paxillin KD relative to controls (Fig. EV5C,D). Notably, within the Late switching group of cassette exons, we observed that transcripts encoding SNAP25, ANK2, ANK3, NR1 and the metabotropic glutamate receptor 5 (GRM5) contained cis-binding elements of at least one of these RBPs around splice sites of paxillin KD-affected cassette exons. These findings support the idea that paxillin modulates the timing of AS programs that govern neuronal activity and synaptic strength.

## Paxillin[S119] phosphorylation is required for the timely expression of the Snap25 5b isoform

Transcripts of the presynaptic SNARE protein Snap25 harbor two mutually exclusive exons, Exon 5a and 5b, which are differentially expressed due to AS. While the 5a isoform is expressed constitutively starting at embryonic stages, Snap25 5b expression emerges at P7 and becomes the dominant isoform (Bark et al, 2004; Bark et al, 1995). Our RNA-seq and rMATs analysis revealed that neuronal paxillin KD significantly impaired the induction of Snap25 5b (Fig. 5C,D) and delayed the isoform switch, with or without BDNF treatment. We also evaluated Snap25 5a and 5b exon inclusion ratios ($\Delta\psi$) using RT-PCR and restriction enzyme digestion (Appendix Fig. S7). In control neurons, the 5b to 5a inclusion ratio reversed between DIV5 and DIV10, coinciding with the critical period of synaptic plasticity (Fig. 5C,D; Appendix Fig. S7). In contrast, paxillin KD neurons at DIV10 exhibited significantly reduced 5b expression, supporting a role for paxillin in regulating Snap25 AS. To further examine whether

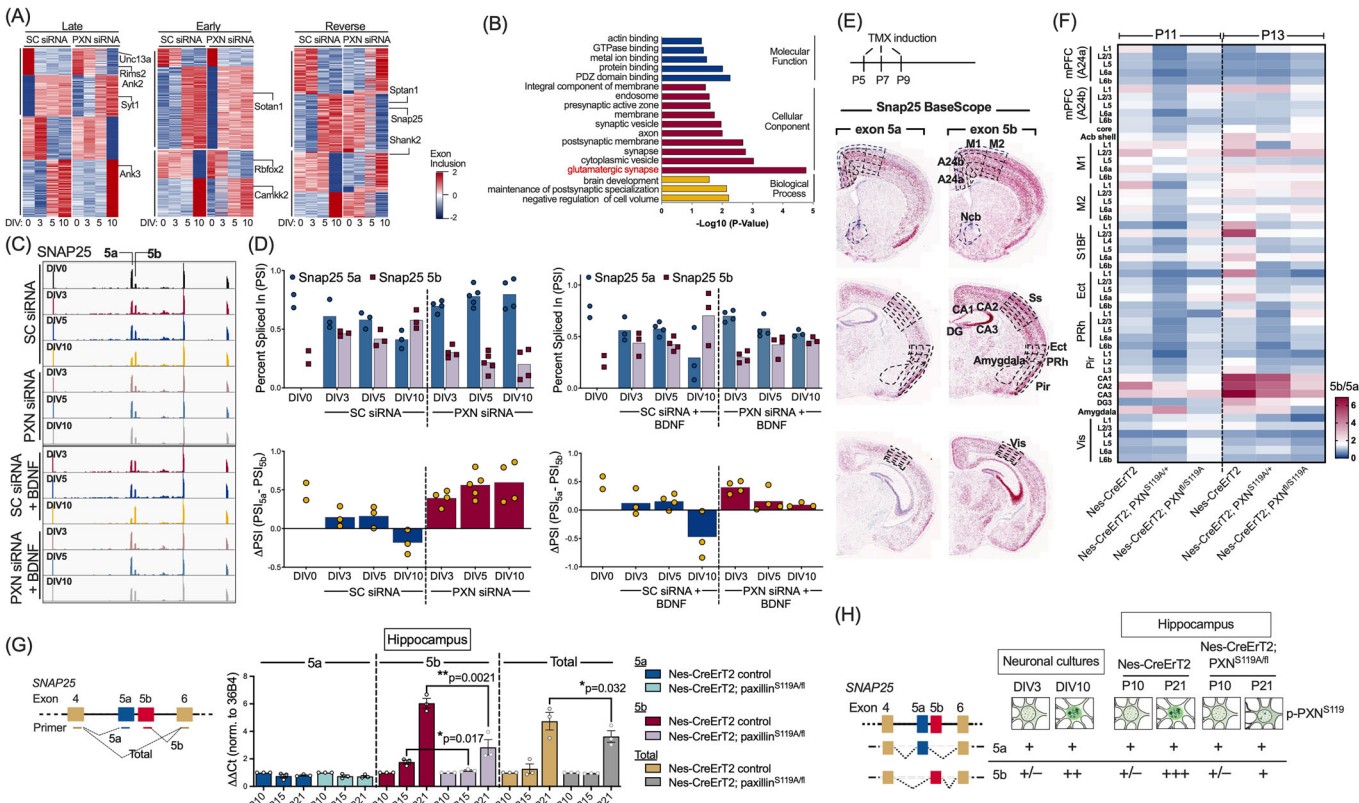

**Figure 5. Paxillin loss alters alternative splicing and suppresses upregulation of the Snap25 5b isoform.**

(A) Inclusion profiles of cassette exons at different time points derived from control (SC) or paxillin siRNA-treated mouse primary cortical neuronal cultures. Heatmap depicts the relative inclusion ratio of cassette exons, calculated as the percent spliced in (PSI) for each time point. AS switching time is defined as the transition at which the sign of the next PSI value reverses. Cassette exons were categorized as "early," "late," or "reverse" based on the relative AS switching time in paxillin siRNA-treated compared to control (SC siRNA) cultures. (B) Gene Ontology (GO) enrichment analysis was performed on genes exhibiting delayed switching (late group) in paxillin siRNA-treated cells relative to control (SC siRNA) cells. (C–F) Deficiency in phosphorylation of paxillin S119 suppresses Snap25 5b mRNA isoform expression. (C) Example of developmental changes in inclusion levels of Snap25 exons 5a and 5b. Note that paxillin siRNA-treated neuronal cultures on DIV10 exhibit decreased exon 5b inclusion relative to control cultures. (D) Histogram summarizing relative inclusion (PSI) as well as changes in relative inclusion (ΔPSI) between exons 5a and 5b, from all experiments as shown in (C). Data represent mean (n = 2–5 independent cultures per group). (E) BaseScope™ images showing Snap25 exon 5a- and 5b-specific detection in mouse brain sections. Punctate red signals, Snap25 5a or 5b mRNAs. Scale bar, 200 mm. (F) Heatmap depicting quantitation of BaseScope data, calculated as intensity ratios of Snap25 5b to 5a in indicated brain regions from mice with tamoxifen-induced neuronal paxillin[S119A] cKI and/or paxillin cKO (Nes-CreErT2; PXN[S119A/+] ; Nes-CreErT2; PXN[fl/+]; Nes-CreErT2; PXN[S119A/fl]) on P11 or P13, as indicated. Mice were administered Tamoxifen at P5, P7, and P9. Data represent mean ± SEM (n = 3 cortices). Scale bar, 200 mm. (G) Real-time qPCR analysis of Snap25 exon 5a, 5b, and total mRNA expression in mouse hippocampal tissue at the indicated postnatal days. A schematic of the primer locations is shown on the left. Data represent mean ± SEM (n = 3 mice per genotype; *p < 0.05, **p < 0.01 relative to P10; comparisons made using multiple unpaired t-tests). (H) Table summarizing relative Snap25 5a and 5b mRNA expression before and after the AS switch. Expression level indicated as: "+++" >5x > "++" >2x > "+" >1x > "+/−" > 0 relative to 5a at DIV3 or P10. Schematics (above) show representative p-Paxillin S119 (green) staining patterns in neurons at the corresponding stages.

activity-dependent regulation of Snap25 AS persists in more mature neurons, we performed isoform-specific RT-qPCR across later developmental time points (DIV14, DIV21, and DIV28) following NMDA receptor activation (Fig. EV3E). Consistent with RNA-seq and rMATs data, DIV14 neurons showed a significant NMDA-induced increase in *Snap25-5b* expression and an elevated 5b/5a ratio. In contrast, such NMDA-induced enhancement was reduced at DIV21 and nearly absent by DIV28, indicating that loss of splicing responsiveness occurs with maturation. Additionally, prolonged NMDA exposure (6 h) reduced total *Snap25* mRNA levels at all ages, likely due to transcript degradation, and did not further increase levels of the 5b isoform. These findings confirm that Paxillin[S119] phosphorylation contributes to timely Snap25 5b induction within a defined, activity-sensitive window during neuronal development.

To assess a function for paxillin S119 phosphorylation in Snap25 splice site switching across different brain regions during crucial developmental periods, we engineered 2 inducible mouse lines--one with targeted loss of paxillin S119 phosphorylation in neurons (*Nestin-CreER[T2]; paxillin[S119A/fl]*) and another with loss of paxillin S119 phosphorylation in activity-trapped neurons (*TRAP2; paxillin[S119A/fl]*). Deficiencies were timed to occur post-Tamoxifen induction at P5, P7, and P9 prior to p-Paxillin[S119] nuclear translocation (Appendix Figs. S1A,B and S8). Nanopore long-read cDNA sequencing of hippocampal tissue collected from P10, P15, and P21 Nestin-CreERT2 mice confirmed that production of alternatively spliced isoforms of both Snap25 and NMDAR1 decreased in the hippocampus of Nestin-CreERT2; Paxillin[fl/S119A] relative to control mice (Appendix Fig. S6). We utilized BaseScope

with probes specific to either Snap25 5a or 5b isoforms and analyzed their distribution in brain slices at P11 and P13 across various regions (Fig. 5E,F). In P13 control mice, the 5b isoform predominated over the 5a in the hippocampal CA3 region, dentate gyrus, and cortical areas. Conversely, deficiency in S119 phosphorylation (Nestin-ER$^{T2}$; paxillin$^{S119A/fl}$) decreased the 5b/5a ratio in these brain regions (Fig. 5E,F). To further validate the developmental AS switch between Snap25 5a and 5b isoforms observed from P10 to P21, we performed RT-qPCR using 5a- and 5b-specifc primers. Consistent with Base-Scope findings, Nestin-CreERT2; paxillin$^{S119A/fl}$ hippocampi expressed significantly lower levels of the 5b isoform at P15 and P21 compared to control hippocampi (Fig. 5G,H), strongly suggesting that loss of S119 phosphorylation prior to critical periods prevents switching to the Snap25 5b isoform.

## Early loss of postnatal Paxillin$^{S119}$ phosphorylation impairs synaptic plasticity and working memory

Given that Snap25 is essential for presynaptic neurotransmitter release, a delay in the transition to its 5b isoform in early postnatal brain may alter homeostatic plasticity of neural circuits (Irfan et al, 2019). To assess this possibility, we conducted electrophysiological behavioral analyses to evaluate how paxillin S119 phosphorylation impacts synaptic activity within the hippocampal Schaffer collateral-CA1 pathway. Mice with inducible deficiency in paxillin S119 phosphorylation (Nestin-CreERT2; paxillin$^{S119A/fl}$ or Nestin-CreERT2; paxillin$^{S119A/+}$) and those with reduced paxillin expression (Nestin-CreERT2; paxillin$^{fl/+}$) were administered tamoxifen at P5, P7, and P9, timed to precede anticipated nuclear translocation of the phosphorylated protein (Fig. 6A). When mice were one-month old, we prepared acute hippocampal slices maintaining the Schaffer collateral-CA1 pathway to perform field recordings to assess input-output relationships (Fig. 6B), paired-pulse facilitation (Fig. 6C), and long-term potentiation (Fig. 6D,E). We first performed extracellular field recordings from the CA1 area, pairing tetanic stimulation of Schaffer collateral afferents and assessing input-output relationships between the slope of field excitatory post-synaptic potentials (fEPSPs) and the amplitude of the afferent fiber volley over a range of stimulus intensities. As shown in Fig. 6B, relative to control littermates, mice deficient in phosphorylated paxillin S119 (Nestin-ERT2; paxillin$^{S119A/fl}$) exhibited a significantly reduced fEPSP slope in response to stimuli, suggesting reduced synaptic efficacy. Conversely, mice with lower levels of phosphorylated paxillin S119 (Nestin-CreERT2; paxillinS119A/+) or reduced levels of total Paxillin (Nestin-CreERT2; paxillin$^{fl/+}$) exhibited fEPSP slopes comparable to those seen in control littermates.

We then asked whether paxillin$^{S119}$ phosphorylation altered short-term synaptic plasticity by assessing paired-pulse facilitation (PPF) at Schaffer collateral synapses by analyzing the amplitude of response evoked by paired presynaptic spikes over a range of intervals between spikes (Fig. 6C). The PPF index was expressed as the ratio of the slope of fEPSP from the second spike to that of the first. Consistently, responses recorded from CA1 synapses of p-Paxillin S119-deficient Nestin-ERT2; paxillin$^{S119A/fl}$ Schaffer collaterals exhibited significantly lower fEPSPs than Nestin-ERT2 controls, with PPF induction abolished. By contrast, PPF induction in Nestin-CreERT2; paxillin$^{fl/+}$ and control mice was comparable. Interestingly, Nestin-CreERT2; paxillin$^{S119A/+}$ mice exhibited enhanced PPF, suggesting abnormally decreased release probability.

The timing of the synaptic sensitive period is similar to that of the appearance of nuclear p-Paxillin S119 in young postnatal brain (Fig. 3). Thus we analyzed a potential requirement for paxillin$^{S119}$ phosphorylation in inducing N-methyl-D-aspartate receptor (NMDAR)-dependent long-term plasticity in young mice (Fig. 6D,E) by evaluating LTP in the hippocampal Schaffer collateral-CA1 pathway in P35 mice administered tamoxifen at P5 and P7, prior to critical developmental periods. To trigger LTP, we used high-frequency stimuli (HFS; 3 trains at 100 Hz, each lasting 1 s with 20-second intervals). As shown in Fig. 6D, mice lacking paxillin S119 phosphorylation (Nestin-ERT2; paxillin$^{S119A/fl}$) or those with lower paxillin expression (Nestin-CreERT2; paxillin$^{fl/+}$) exhibited lower evoked fEPSP slopes after HFS compared to control littermates. We also observed impaired long-term depression (LTD) induced by low-frequency tetanic stimulation (1 Hz for 15 min) in P16 mice with reduced or absent S119 phosphorylation, indicating that overall neuronal plasticity during critical periods requires S119-phosphorylated paxillin.

Finally, disrupted synaptic plasticity seen in Nestin-ERT2; paxillin$^{S119A/fl}$ mice could perturb working memory. To test this possibility in active neurons, we employed Targeted Recombination in Active Populations (TRAP2) mice to create activity-dependent paxillin S119A knock-in (TRAP2; paxillin$^{S119A/fl}$) or KD (TRAP2; paxillin$^{fl/+}$) mice. TRAP2 mice were administered Tamoxifen at postnatal days 5, 7, and 9, during initiation of sensitive developmental periods. Adolescent (P35-P40) mice either deficient in paxillinS119 phosphorylation (TRAP2; paxillin$^{S119A/fl}$) or expressing lower paxillin levels (TRAP2; paxillin$^{fl/+}$) exhibited normal locomotor and social behaviors (Fig. 7A–D); however, in open-field tests, adolescent TRAP2; paxillin$^{S119A/fl}$ mice exhibited anxiety-like behavior, spending more time in the center area compared to control littermates (Fig. 7B). Furthermore, sensitive period-TRAPed mice with p-Paxillin$^{S119}$ deficiency (TRAP2; paxillin$^{S119A/fl}$) showed decreased preference for a texture-rich, interesting object in young adulthood, with or without exposure to physical restraint stress (Fig. 7E,F). Notably, young adult TRAP2; paxillin$^{S119A/fl}$ mice at P65 exhibited impaired discrimination between novel and familiar objects based on visual surface feature cues in a novel object recognition test (Fig. 7E,F). Consistently, sensitive period-TRAPed mice with reduced paxillin expression or paxillin S119 phosphorylation (i.e., TRAP2; paxillin$^{S119A/fl}$, TRAP2; paxillin$^{S119A/+}$, TRAP2; paxillin$^{f/+}$) showed no tendency to enter a new arm in a Y-maze test at P65 (Fig. 7G), indicating impaired hippocampal-dependent spatial working memory. By contrast, TRAP2; Paxillin$^{S119A/fl}$ mice that underwent tamoxifen induction at a later stage (3 months old, i.e., 1 week prior to behavioral testing) exhibited comparable performance in open field, Y-maze, and novel object recognition tests (Appendix Fig. S9). Together, these findings suggest that paxillin S119 phosphorylation in sensitive period-TRAPed neurons is critical for normal cognitive functions in adolescent and young adult mice.

## Discussion

Refinement of neural circuits at early developmental stages requires that neurons initiate an AS program to enhance synaptic efficacy and strengthen neuronal connections. Here, we report a neuron-specific mechanism by which paxillin phosphorylated at serine 119 (p-Paxillin$^{S119}$) acts downstream of BDNF/cAMP/Cdk5 signaling and synaptic activity to initiate such a program in the nucleus, by interacting with the splicing machinery to promote timely AS of genes

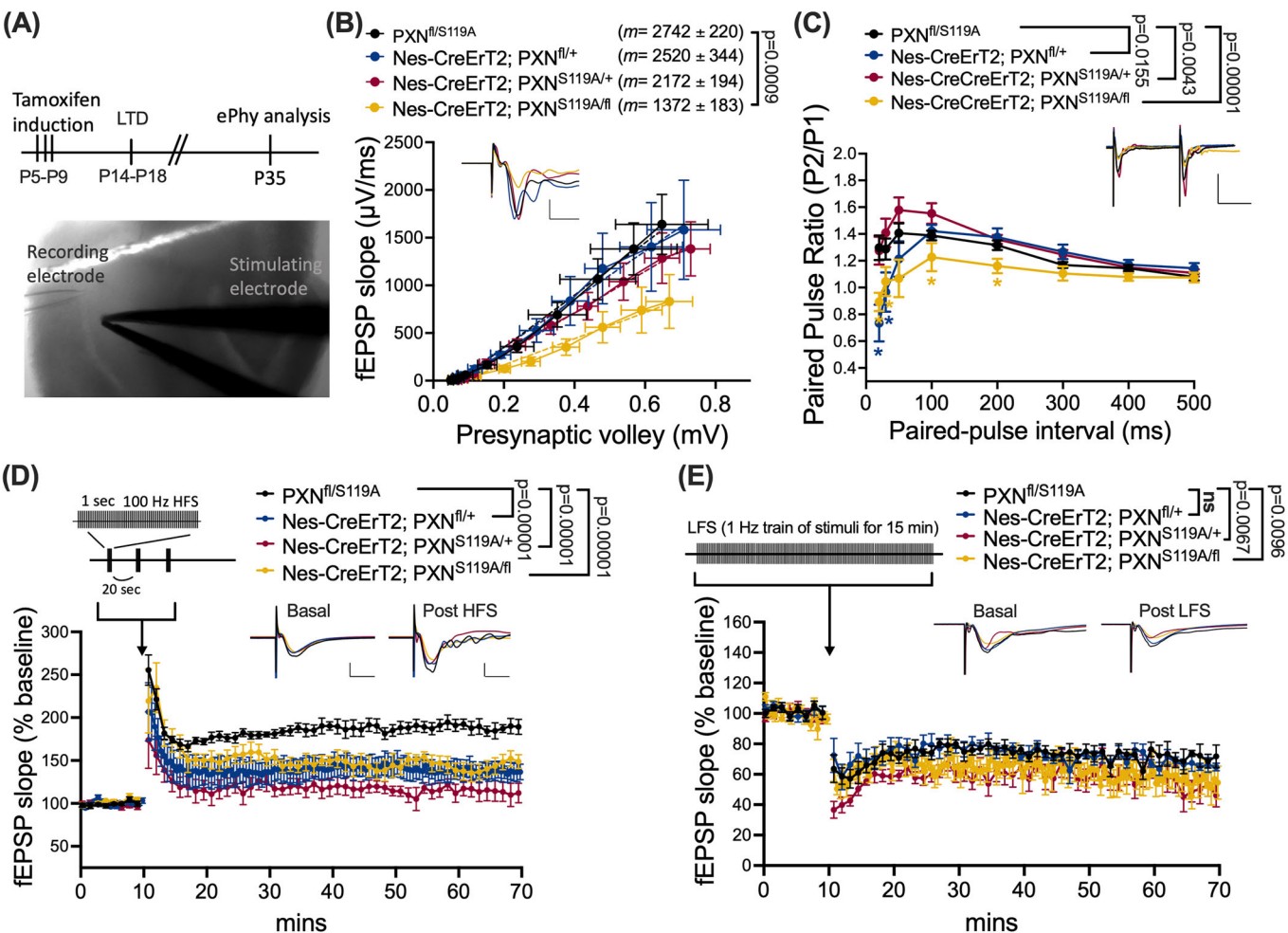

**Figure 6. Mice with early postnatal paxillin S119 phosphorylation deficiency show altered synaptic properties at Schaffer collateral synapses.**

(A) Representative differential interference contrast (DIC) images of a hippocampal slice depicting the position of the stimulating electrode in Schaffer collateral fibers and recording pipettes in CA1. The schematic at the top depicts the experimental timeline of tamoxifen injection (at P5, P7, and P9) and subsequent electrophysiological recordings. (B) Input-output (I/O) relationships of field excitatory postsynaptic potentials (fEPSPs), indicating reduced synaptic efficacy in *Nes-CreErT2; PXN^{S119A/fl}* mice. Representative traces from control and neuronal paxillin S119A cKI and/or paxillin cKO doubly heterozygous mice, recorded in response to a 1.1 mA stimulus, are superimposed (scale bars: 1 mV on the y-axis and 5 ms on the x-axis). Data were presented as mean ± SEM (n = 9 control; n = 6 *Nes-CreErT2; PXN^{fl/+}*; n = 7 *Nes-CreErT2; PXN^{S119A/+}*; and n = 7 *Nes-CreErT2; PXN^{S119A/fl}*). The slope (m) of the linear fit (dashed line) serves as an index of synaptic efficacy. Statistical comparison of slopes (***p < 0.001) was performed using linear regression analysis in GraphPad Prism. (C) Paired-pulse facilitation was plotted with P2/P1 against varying intervals, where P1 and P2 represent the amplitude of response (slopes of fEPSP) of the first and second stimulus, respectively. Data presents mean ± SEM (n = 10 for control, n = 6 for Nes-CreErT2; PXN^{fl/+}, n = 8 for Nes-CreErT2; PXN^{S119A/+}, n = 6 for Nes-CreErT2; PXN^{S119A/fl}. *p < 0.05, **p < 0.01, compared to the control group, one-way ANOVA followed by Dunnett's multiple comparison test. Scale bars: 0.5 mV (y-axis) and 50 ms (x-axis). (D) Nes-CreErT2; PXN^{S119A/fl} mice show impaired long-term potentiation (LTP) of Schaffer collateral-CA1 synapses. LTP was induced by high-frequency stimulation (1 s of HFS at 100 Hz, 20 s interval, three trains). Plot shows fEPSP slope (mean ± SEM, normalized to the first 10 min EPSP magnitude before HFS; n = 10 for control, n = 6 for Nes-CreErT2; PXN^{fl/+}, n = 8 for Nes-CreErT2; PXN^{S119A/+}, n = 6 for Nes-CreErT2; PXN^{S119A/fl}; ***p < 0.001, compared to control, one-way ANOVA followed by Dunnett's multiple comparison test). Scale bars: 0.5 mV (y-axis), 10 ms (x-axis). (E) Nes-CreErT2; PXN^{S119A/fl} mice show impaired long-term depression (LTD) of Schaffer collateral-CA1 synapses. LTD was induced by low-frequency stimulation (LFS; 1 Hz for 15 min). Plot shows fEPSP slope (mean ± SEM, normalized to the first 10 min EPSP magnitude before LFS; n = 10 for control, n = 6 for Nes-CreErT2; PXN^{fl/+}, n = 8 for Nes-CreErT2; PXN^{S119A/+}, n = 6 for Nes-CreErT2; PXN^{S119A/fl}; ns non-significant, **p < 0.01, compared to control, one-way ANOVA followed by Dunnett's multiple comparison test. Scale bars: 0.5 mV (y-axis), 10 ms (x-axis).

governing synaptic plasticity. Blocking paxillin S119 phosphorylation significantly delayed AS switching of numerous genes crucial for synaptic activation, including those encoding SNAP25, several Anks, and the GABAb receptor. As a consequence, presynaptic transmission and long-term plasticity via Schaffer collateral input were altered in mice lacking phosphorylated paxillin S119 starting at the second postnatal week, leading to learning and memory deficits. Our findings strongly suggest that paxillin^{S119} phosphorylation plays an important

role in regulating AS during a critical period, an activity essential for the development and function of neural circuits (Fig. 7H).

These findings illustrate the unique regulation of paxillin localization and function in neurons. Others have reported that in fibroblasts or epithelial-like cancer cells (Dong et al, 2009; Noh et al, 2021; Sathe et al, 2016; Sen et al, 2012; Sen et al, 2010), paxillin's nuclear localization is regulated by serine phosphorylation of its nuclear export LD motif and/or nuclear-localizing LIM domain (Deakin and Turner, 2008; Dong et al,

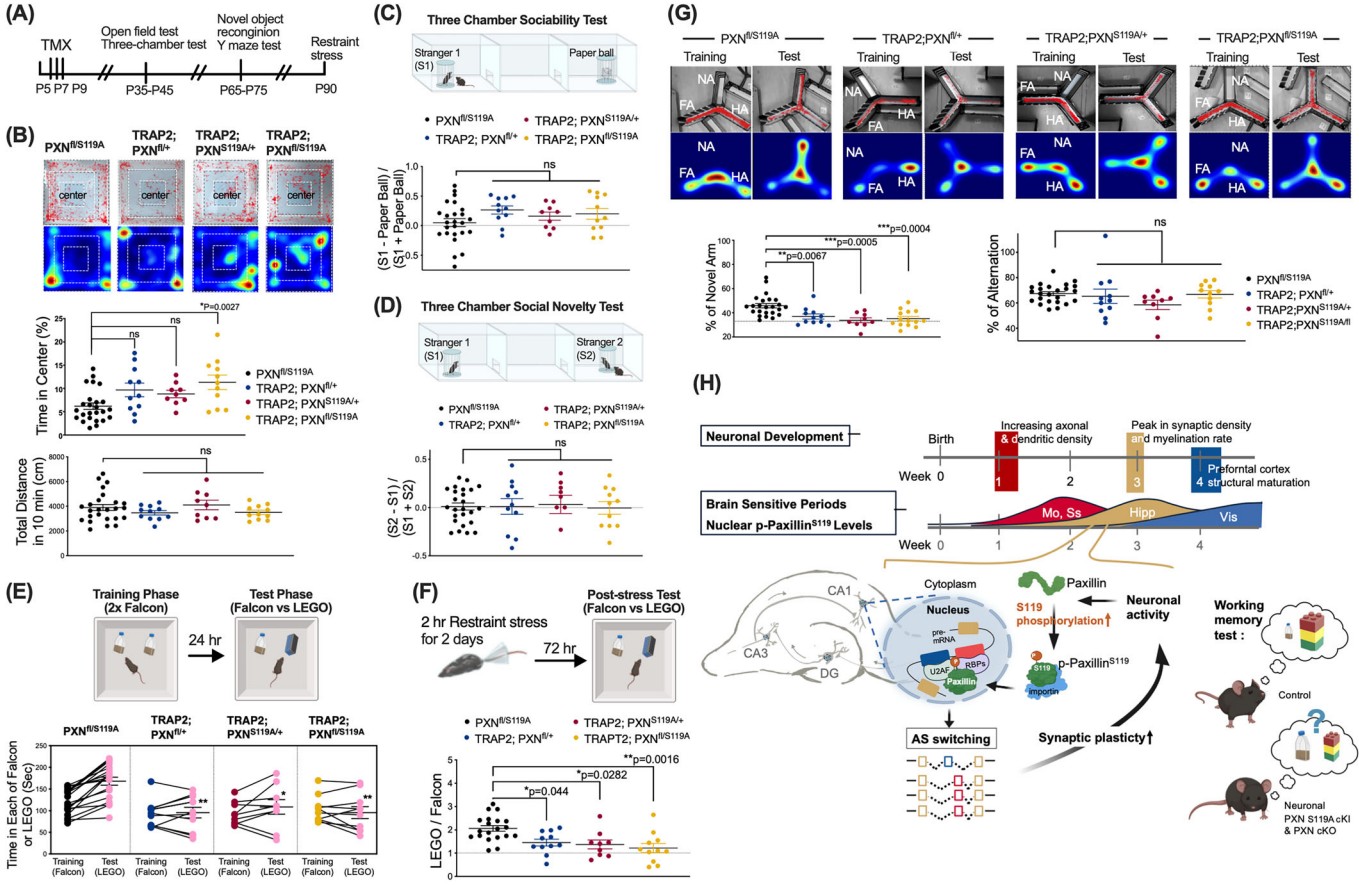

**Figure 7. Mice with early postnatal p-Paxillin$^{S119}$ deficiency exhibit anxiety-like behavior and deficits in working memory.**

(A) Schematic showing induction of TRAP2 mice via tamoxifen injections at the indicated days, followed by a sequence of behavioral tests. (B) Open-field test of TRAP2; PXN$^{S119A/fl}$ mice and littermate controls. TRAP2 mice spent more time in the center of the arena than did littermate controls, suggesting anxiety-like behavior. Top panels, representative traces and heatmaps indicating mouse movement recorded in 10 min test sessions. Middle panel, plot shows percentage of time ( ± SEM; $n > 8$ mice per genotype; *$p < 0.05$ compared to controls by multiple unpaired $t$-tests) spent in the center area by indicated mouse groups. Bottom panel, plot shows the total distance traveled by mouse groups (±SEM; $n > 8$ mice per genotype; ns not significant compared to controls by multiple unpaired $t$-tests) in a 10-min session. (C, D) Three-chamber test shows comparable sociability (C) and preference for social novelty (D) in TRAPed control and TRAPed paxillin S119A and/or paxillin cKO mice. Data represents mean ( ± SEM; $n > 8$ mice each group; ns, non-significant compared to controls by multiple unpaired $t$-tests). (E–G) TRAP2; PXN$^{S119A/fl}$ mice exhibit impaired working memory. (E, F) Novel object recognition test designed to measure the ability to discriminate a novel (LEGO) from a familiar (Falcon tube) object before (E) and after (F) restraint stress. TRAP2; PXN$^{S119A/fl}$ mice spent significantly less time exploring the novel relative to the familiar object. Data compared mean time (±SEM; $n > 8$ mice per genotype; *$p < 0.05$, **$p < 0.01$ compared to control test (LEGO); multiple $t$-test) between TRAPed paxillin S119 cKI/paxillin cKO and TRAPed control mice. (G) The Y-maze spontaneous alternation test is used to evaluate spatial working memory by measuring the percentage of consecutive entries into all three arms without repetition. TRAP2; PXN$^{S119A/fl}$ mice showed a smaller percentage of entries into the novel (NA) compared to the familiar arm (NA). Data represents mean ( ± SEM; $n > 8$ mice per genotype; ns non-significant; *$p < 0.05$, **$p < 0.01$ compared to controls by multiple unpaired $t$-tests). (H) A model depicting paxillin phosphorylation at serine 119 (S119) as a molecular switch for alternative splicing (AS) transitions and synaptic plasticity. The timeline highlights key neuronal milestones, with rising nuclear p-Paxillin$^{S119}$ levels coinciding with the onset of experience-dependent sensitive periods across different brain circuits. Neuronal activity induces paxillin S119 phosphorylation, facilitating its nuclear import and condensation within nuclear speckles. There, p-Paxillin$^{S119}$ interacts with splicing factors to regulate AS-driven synaptic refinement. Mice lacking neuronal paxillin S119 phosphorylation exhibit impaired working memory, underscoring its role in timely circuit maturation and cognitive function.

2009; Ma and Hammes, 2018). However, mechanisms controlling paxillin nuclear transport in differentiated neurons in physiological conditions remain unclear. Neurons cultured on uniform substrates in standard medium form many fewer mature adhesion foci compared to fibroblasts or epithelial cells, a decrease that may permit recruitment of paxillin to other protein complexes in response to external signals or internal cellular activities (Chang et al, 2017; Sathe et al, 2016; Turner, 2000). Our findings in differentiated neurons reveal that paxillin phosphorylation at serine 119 and its subcellular localization are highly dependent on a neuron's developmental stage. For example, in immature neurons (days 5–7 in vitro), we observed that paxillin S119

phosphorylation occurs transiently after axon determination, and p-Paxillin$^{S119}$ nuclear localization is importin β2-dependent. Such transient serine phosphorylation is regulated by autocrine BDNF/cAMP signaling (Cheng et al, 2011) and may be coordinated with postnatal neuronal Cdk5 activation (Meyerson et al, 1992; Pao and Tsai, 2021; Takahashi et al, 2003). Those levels then decline to basal levels by the second week of culture.

Conversely, once mature, neurons require synaptic activation via NMDA receptors to induce increases in nuclear p-Paxillin$^{S119}$. This activity coincides with significant remodeling of splicing factors and occurs dependently of importin β2 activity. Other

serine-phosphorylated forms of paxillin, such as S83 and S178, remain constitutively localized in the nucleus, regardless of neuronal maturation status. Furthermore, although paxillin phosphorylation at S273 reportedly blocks NES activity and facilitates its nuclear transport in fibroblasts (Dong et al, 2009), we found that paxillin phosphorylated at S273 was predominantly cytoplasmic in primary cultured neurons, although it became transiently nuclear in neuroblastoma N2a cells in non-mitotic phases of the cell cycle. Overall, except for S119 seen in mature neurons, none of the serine-phosphorylated forms of paxillin tested here were responsive to NMDA receptor activation. Notably, nuclear levels of pan-paxillin did not show a clear or significant increase across developmental stages or after NMDA stimulation, unlike p-Paxillin$^{S119}$. This may reflect technical limitations, such as dilution by unphosphorylated paxillin or epitope masking by phosphorylation-dependent conformational changes, or the possibility that a substantial fraction of nuclear paxillin is phosphorylated by kinases within the nucleus. These limitations should be considered when interpreting differences between total and phosphorylated paxillin.

A critical question related to early postnatal brain development is how synaptic plasticity influences AS programs, and vice versa, in order to initiate sensitive periods. While intrinsic factors like age and genetics influence such critical periods, experience-dependent neural activity plays a pivotal role in their timing and duration (Berardi et al, 2000; Desai et al, 2002; Hooks and Chen, 2007; Nakamura et al, 2020). Development of neural circuits requires sensory and cognitive input to strengthen connectivity, while exposure of animals to enriched environments or to neurotrophin signaling can extend plasticity (Akaneya et al, 1997; Kang et al, 1997; Lessmann et al, 2003; Lodovichi et al, 2000; Wang et al, 2013). Moreover, critical periods are not uniform across neural systems (Dehorter and Del Pino, 2020; Pedrosa et al, 2022; Reh et al, 2020). Brain regions mature in a rostro-caudal sequence, and a critical/sensitive period for a particular stage cannot begin until input generated in preceding stages in maturing neurons is available. Such a cascade of sensitive periods, each occurring at specific hierarchical levels, shapes the development of brain functions as corresponding pathways acquire plasticity. Our work illustrates how these events are coordinated by synaptic activation-associated switches of AS programs at the onset of sensitive periods, although it is also known that dynamic spatiotemporal patterns of RBP expression regulate splicing events. Specifically, we observed that activation of NMDA receptors in dissociated DIV7 neuronal cultures enables p-Paxillin$^{S119}$ to form complexes with nuclear speckle components and

remodels the splicing machinery. Our findings suggest that a feedback loop governs entry of neuronal networks into a plastic state: specifically, activation of postsynaptic NMDA receptors increases nuclear levels of p-Paxillin$^{S119}$, in turn modulating the splicing machinery and resulting in a select set of splicing events that shape synaptic connectivity. Indeed, we found that RBP-binding cis-elements associated with delayed AS events after paxillin knockdown and residing in cassette exons or their adjacent introns are linked to the core splicing factor U2AF2 and neuron-specific RBPs, such as RbFox2 and NOVA1/2. Moreover, paxillin phosphorylation at S119 enhances its interaction with RBPs. Thus, the ability of NMDA receptor activation to increase the nuclear pool of p-Paxillin$^{S119}$ may enable proper temporal switching of splice site selection in response to synaptic input. Accordingly, the timing of nuclear p-Paxillin$^{S119}$ upregulation in different brain areas coincides with the onset of critical periods in those regions. As a result, mice deficient in neuronal p-Paxillin$^{S119}$ during the first postnatal week exhibited a delay in the switch from the SNAP25 5a to 5b isoform, decreasing the probability of neurotransmitter release at CA1 synapses. Also, TRAP2 mice deficient in p-Paxillin$^{S119}$ showed impaired novel object recognition ability, supporting the physiological relevance of this modification.

Taken together, our study demonstrates that p-Paxillin$^{S119}$ provides a vital link between synaptic activity and splicing control during a critical developmental window. By bridging external signals and the splicing machinery in the nucleus, phosphorylated paxillin in neurons ensures that key synaptic genes undergo appropriate AS transitions at the right time. These findings place paxillin$^{S119}$ alongside other neuronal regulatory mechanisms that coordinate gene isoform expression and circuit maturation. In the broader context, unveiling p-Paxillin$^{S119}$'s function underscores how intracellular signaling cascades and specific RBPs converge to shape synaptic properties crucial for working memory. Given that disruptions in synaptic maturation and splicing regulation are implicated in neurodevelopmental disorders such as autism spectrum disorder and schizophrenia, our results raise the possibility that altered p-Paxillin$^{S119}$ signaling could contribute to these pathophysiologies. The developmental stage-specific and activity-dependent features of this pathway highlight potential therapeutic windows for restoring proper splicing dynamics and synaptic refinement, offering new avenues for intervention in neurodevelopmental disease.

# Methods

**Reagents and tools table**

| Reagent/resource | Reference or Source | Identifier or Catalog Number |
| --- | --- | --- |
| **Experimental models** | | |
| *Fostm2.1(icre/ERT2)Luo/J; Fos2A-iCreER (TRAP2)* | The Jackson Laboratory | RRID:IMSR_JAX:030323 |
| *C57BL/6-Tg(Nes-cre/ERT2)KEisc/J; Nes-cre/ERT2* | The Jackson Laboratory | RRID:IMSR_JAX:016261 |
| Human embryonic kidney 293T cells | ATCC | Cat# CRL-3216, RRID:CVCL_0063 |
| Neuro2a cells | ATCC | Cat# CCL-131, RRID:CVCL_0470 |
| Sprague Dawley timed-pregnant rats | BioLASCO | N/A |
| ECOS101 DH5α competent cell | Yeastern Biotech Co. | Cat#FYE678 |

| Reagent/resource | Reference or Source | Identifier or Catalog Number |
|---|---|---|
| **Recombinant DNA** | | |
| Paxillin-pEGFP | addgene | RRID:Addgene_15233 |
| Paxillin(S119A)-pEGFP | This paper | N/A |
| Paxillin(S11DA)-pEGFP | This paper | N/A |
| Paxillin(P398A/Y399A)-pEGFP | This paper | N/A |
| Paxillin(P516A/Y517A)-pEGFP | This paper | N/A |
| Paxillin(P575A/Y576A)-pEGFP | This paper | N/A |
| Flag-Paxillin | addgene | RRID:Addgene_15212 |
| Flag-Paxillin(S119A) | This paper | N/A |
| Flag-Paxillin(S119D) | This paper | N/A |
| Flag-Paxillin(P398A/Y399A) | This paper | N/A |
| Flag-Paxillin(P516A/Y517A) | This paper | N/A |
| Flag-Paxillin(P575A/Y576A) | This paper | N/A |
| Flag-Paxillin(P398A/Y399A,516A/Y517A;P575A/Y576A) | This paper | N/A |
| pcDNA3-cdk5GFP | Addgene | RRID:Addgene_1346 |
| pcDNA3-dnCDK5 | Addgene | RRID:Addgene_1344 |
| pcDNA3.1-P25C-GFP | Addgene | RRID:Addgene_1343 |
| Gateway™ pDONR/Zeo Vector and Gateway™ pDESTTM15 Vector, Gateway™ Cloning System | Thermo Fisher Scientific | Cat#11803-012, Cat# K37120, Cat# K37020 |
| pLenti-C-mGFP | OriGeneTechnologies | Cat#PS100071 |
| pGFP-C-shLenti shRNA Vector | OriGeneTechnologies | Cat#TR30023 |
| **Antibodies** | | |
| Anti-β-actin mouse monoclonal antibody (1C7) | Abbkine | Cat# A01010, RRID:AB_2737288 |
| Anti-Paxillin antibody [E228] | Abcam | Cat#ab32115, RRID:AB_777116 |
| anti-alpha Tubulin (ab89984) | Abcam | Cat#ab89984, RRID:AB_10672056 |
| Fox2 / RBM9 antibody | Abcam | Cat#ab57154, RRID:AB_2285090 |
| Anti-mCherry antibody [1C51] | Abcam | Cat# ab125096, RRID:AB_11133266 |
| Mouse Anti-SC-35 monoclonal antibody, unconjugated, clone SC-35 | Abcam | Cat#ab11826, RRID:AB_298608 |
| anti-paxillin clone 349 (612405) | BD Biosciences | Cat#612405, RRID:AB_647289 |
| Rabbit anti-SF3b155/SAP155 Antibody, Affinity Purified | Bethyl | Cat#A300-996A, RRID:AB_805834 |
| Rabbit anti-Matrin 3 Antibody, Affinity Purified | Bethyl | Cat#A300-591A, RRID:AB_495514 |
| Rabbit anti-RBM9 Antibody, Affinity Purified | Bethyl | Cat#A300-864A, RRID:AB_609476 |
| GAPDH antibody | GeneTex | Cat#GTX100118, RRID:AB_1080976 |
| PTBP1 monoclonal antibody (1) | Invitrogen | Cat#32-4800, RRID:AB_2533082 |
| c-Myc monoclonal antibody (9E10) | Invitrogen | Cat#MA1-980, RRID: AB_558470 |
| chicken polyclonal βIII tubulin (AB9354) | Merck Millipore | Cat#AB9354, RRID:AB_570918 |
| Anti-actin antibody, clone C4 (MAB1501) | Merck Millipore | Cat#MAB1501, RRID:AB_2223041 |
| Mouse anti-phosphoepitope SR proteins monoclonal antibody, unconjugated, clone 1H4 | Millipore | Cat#MABE50, RRID: AB_10807429 |
| Anti-NeuN antibody | Millipore | Cat# ABN91, RRID:AB_11205760 |
| U2AF35 rabbit Polyclonal antibody | proteintech | Cat#10334-1-AP, RRID:AB_2211314 |
| NOVA2 rabbit Polyclonal antibody | proteintech | Cat#55002-1-AP, RRID:AB_10858926 |
| ELAVL2 antibody | Proteintech | Cat# 14008-1-AP, RRID:AB_2096356 |
| HuC Elavl3 antibody | Proteintech | Cat# 55047-1-AP, RRID:AB_10859256 |
| U2AF35 Monoclonal antibody | Proteintech | Cat#CL647-60289, RRID:AB_2934985 |
| Transportin-1 antibody | Proteintech | Cat#20679-1-AP, RRID:AB_10694291 |
| SF3B3 antibody | Proteintech | Cat#14577-1-AP, RRID:AB_2270189 |

| Reagent/resource | Reference or Source | Identifier or Catalog Number |
|---|---|---|
| IGF2BP1 antibody | Proteintech | Cat# 22803-1-AP, RRID:AB_2879173 |
| IGF2BP2 antibody | Proteintech | Cat# 11601-1-AP, RRID:AB_2122672 |
| IGF2BP3 antibody | Proteintech | Cat# 14642-1-AP, RRID:AB_2122782 |
| Anti-Paxillin Antibody | PhosphoSolutions | Cat#PM1071, RRID:AB_2174719 |
| Paxillin (Ser-83), phosphospecific antibody | PhosphoSolutions | Cat#PP1341, RRID:AB_2253333 |
| Anti-paxillin (Ser-178), phosphospecific antibody | PhosphoSolutions | Cat#PP1051, RRID: AB_2174596 |
| Human/mouse/rat GRIN1/NMDAR1 antibody | R&D systems | Cat# PPS011B, RRID:AB_3659611 |
| Mouse/Rat GRIN1/NMDAR1 C1 splice variant antibody | R&D systems | Cat# PPS080, RRID:AB_2232504 |
| Human/mouse/rat GRIN1/NMDAR1 C2' splice variant antibody | R&D systems | Cat# PPS082, RRID:AB_2112159 |
| Mouse/rat GRIN1/NMDAR1 N1 splice variant antibody | R&D systems | Cat# PPS083, RRID:AB_2112007 |
| anti-FLAG M2 | Sigma-Aldrich | Cat#F1804, RRID:AB_262044 |
| Anti-phospho-Paxillin (pSer272) antibody produced in rabbit | Sigma-Aldrich | Cat#SAB4301321, RRID:N/A |
| U2AF65 (MC3) antibody | Santa Cruz Biotechnology | Cat#sc-53942, RRID:AB_831787 |
| matrin-3 (2539C3a) antibody | Santa Cruz Biotechnology | Cat#sc-81318, RRID:AB_2141656 |
| U1 snRNP 70 antibody (C-3) | Santa Cruz Biotechnology | Cat#sc-390899, RRID:AB_2801569 |
| MBNL2 (3B4) antibody | Santa Cruz Biotechnology | Cat#sc-136167, RRID:AB_2140469 |
| FUS/TLS (4H11) antibody | Santa Cruz Biotechnology | Cat#sc-47711, RRID:AB_2105208 |
| HuR (3A2) ELAVL1 antibody | Santa Cruz Biotechnology | Cat# sc-5261, RRID:AB_627770 |
| HuD (E-1) Elavl4 antibody | Santa Cruz Biotechnology | Cat#sc-28299, RRID:AB_627765 |
| Histone H3 antibody (1 G1) | Santa Cruz Biotechnology | Cat# sc-517576, RRID:AB_2848194 |
| Cdk5 antibody (J-3): sc-6247 | Santa Cruz Biotechnology | Cat# sc-6247, RRID:AB_627241 |
| Paxillin polyclonal antibodies(PA5-34910) | Thermo Fisher Scientific/Invitrogen | Cat#PA5-34910, RRID: AB_2552260 |
| TrkB polyclonal antibody | Thermo Fisher Scientific | Cat#PA5-78405, RRID:AB_2736725 |
| CD171 (L1CAM) polyclonal antibody | Thermo Fisher Scientific | Cat#PA5-85876, RRID:AB_2802677 |
| p-PaxillinS119 antibody | This paper | |
| **Oligonucleotides and other sequence-based reagents** | | |
| paxillin siRNA C: 5'- GAUGAGCAGUCCACAGCGA -3' | This paper | N/A |
| paxillin siRNA D: 5'- CAGCCUAGUGGUUCCAGAU -3' | This paper | N/A |
| Acell non-targeting Pool | Dharmacon | DAMD-001910-10-20 |
| S-to-A point mutation: PXNS119A-F: GAGGAGGAACACGTCTACGCCTTCCCCAACAAGCAGAAG PXNS119A-R: CTTCTGCTTGTTGGGGAAGGCGTAGACGTGTTCCTCCTC | This paper | N/A |
| S-to-D point mutation: PXN-S119D-F: GAGGAGGAACACGTCTACGACTTCCCCAACAAGCAGAAG PXN-S119D-R: CTTCTGCTTGTTGGGGAAGTCGTAGACGTGTTCCTCCTC | This paper | N/A |
| PXN KO genotyping: PXN-int1 GT-Fw: GGAGGAGCTTAGAGCACGGAA PXN-int2 GT-Rv: CACTGGAGCTGTACACGGG | This paper | N/A |
| PXNS119A KI genotyping: Pxn-arm800-GT-Fw: GGGTTCTGTCCTCAGTACTGAC Pxn-arm800-GT-Rv: CCTCAGTCCTTCAAACAGCACG Pxn-16026-GT-Fw: GATACGCTCACCAGCAGCC Pxn-16516-GT-Rv: GTTCACAGTGATGGCCGGGA | This paper | N/A |

| Reagent/resource | Reference or Source | Identifier or Catalog Number |
|---|---|---|
| P398A-Y399A-f:CGAGCGGGACGGACAAGCCGCCTGTGAAAAGGACTACC P398A-Y399A-r:GGTAGTCCTTTTCACAGGCGGCTTGTCCGTCCCGCTCG | This paper | N/A |
| P516A-Y517A-f:CGAGCACGACGGGCAGGCAGCTTGTGAGGTGCACTACC P516A-Y517A-r:GGTAGTGCACCTCACAAGCTGCCTGCCCGTCGTGCTCG | This paper | N/A |
| P575A-Y576A-f:GGAGCAGAACGACAAGGCCGCCTGTCAGAGCTGCTTCC P575A-Y576A-r:GGAAGCAGCTCTGACAGGCGGCCTTGTCGTTCTGCTCC | This paper | N/A |
| L142A-L145A-f:CCCTGGGCAGCAACGCCTCTGAGGCGGACCGGCTGTTAC L142A-L145A-r:GTAACAGCCGGTCCGCCTCAGAGGCGTTGCTGCCCAGGG | This paper | N/A |
| qPCR_Snap25AB_F-1:GTAAAGATGCTGGCATCAGG | This paper | N/A |
| qPCR_Snap25_5A_R:AATCTTTTAAATTTTTCTCGGCC | This paper | N/A |
| qPCR_Snap25_5B_F:CAGAAAAGAATTTGACGGACC | This paper | N/A |
| qPCR_Snap25AB_R-1:TCTGCTCCCGTTCATCCAC | This paper | N/A |
| qPCR_Snap25AB_F-1:GTAAAGATGCTGGCATCAGG | This paper | N/A |
| qPCR_Snap25AB_R-1:TCTGCTCCCGTTCATCCAC | This paper | N/A |
| qPCR_36B4_F:GCTCCAAGCAGATGCAGCA | This paper | N/A |
| qPCR_36B4_R:CCGGATGTGAGGCAGCAG | This paper | N/A |
| PCR_ms_Snap25AB_F-1:GTAAAGATGCTGGCATCAGG | This paper | N/A |
| PCR_ms_Snap25AB_R-1:TCTGCTCCCGTTCATCCAC | This paper | N/A |
| BaseScope™ Probe- BA-Mm-Snap25-tv2-1zz-st-C2: AAATTTAAAAGATTTAGGCAAATGCTGTGGCCTTTTCATATGTCCTT | ACDBio | Cat#851891-C2 |
| BaseScope™ Probe- BA-Mm-Snap25-tv1-1zz-st-C2: TGACGGACCTAGGAAAATTCTGCGGGCTTTGTGTGTGTCCC | ACDBio | Cat#851831-C2 |
| **Chemicals, Enzymes and other reagents** | | |
| Recombinant human/murine/Rat BDNF | PeproTech | Cat#450-02 |
| BSA Alexa Fluor 647 conjugate | Invitrogen | Cat#A34785 |
| Forskolin, 7-deacetyl-7-[O-(N-methylpiperazino)-γ-butyryl]-, dihydrochloride | Calbiochem | Cat#344273 |
| KT5720 | Calbiochem | Cat#420320 |
| Pim1/AKK1-IN-1, also named LKB1/AAK1 dual inhibitor | MedChemExpress | Cat#HY-10371 |
| PP2 | Abcam | Cat#ab120308 |
| PP3 | Abcam | Cat#ab120617 |
| K252a | Abcam | Cat#ab120419 |
| Importazole | Cayman Chemical | Cat#21491 |
| MG132 | Calbiochem | Cat#474790 |
| MDL-28170 (Calpain inhibitor III) | MedChemExpress | Cat#HY-18236 |
| Purvalanol A | Calbiochem | Cat#540500 |
| Roscovitine | Sigma | Cat#R7772 |
| Leptomycin B | Cayman Chemical | Cat#10004976 |
| Selinexor(KPT-330) | Selleckchem | Cat#S7252 |
| Gem21 NeuroPlexTM | GEMINI Bio-products | Cat#400-160 |
| poly-L-Lysine | Sigma-Aldrich | Cat#P2636-1G |
| laminin | Corning | Cat#354232 |
| SYLGARD® 184 SILICONE ELASTOMER KIT | Dow Corning | Material#1673921 |
| N'-N'-methylenebisacrylamide solution | Sigma-Aldrich | Cat#M1533 |
| Acrylamide solution | Sigma-Aldrich | Cat#A4058 |
| (3-aminopropyl)triethoxysilan | Sigma-Aldrich | Cat#A3638 |
| Phusion high-fidelity DNA polymerases | Thermo Fisher Scientific | Cat#F530L |
| Phalloidin Labeling Probes; Alexa Fluor™ 647 | Thermo Fisher Scientific/ Invitrogen | Cat#A22287 |

| Reagent/resource | Reference or Source | Identifier or Catalog Number |
|---|---|---|
| NMDA | TOCRIS | Cat#0114 |
| D-AP5 | TOCRIS | Cat#0106 |
| KN62 | TOCRIS | Cat#1277 |
| CNQX | TOCRIS | Cat#0190 |
| **Software** | | |
| ImageJ | Schneider et al, 2012 | https://imagej.nih.gov/ij/ |
| Imaris 9.3.1 (Bitplane) | Oxford Instruments | https://imaris.oxinst.com |
| ezTrack | Pennington et al, 2019 | https://github.com/denisecailab/ezTrack |
| **Other** | | |
| HisTrap column | GE Healthcare | Cat#17-5247-01 |
| Glutathione Sepharose 4B | GE Healthcare | Cat#17-0756-01 |
| Protein G agarose | Roche | 11243233001 |
| square-wave electroporation generator | BTX Inc. | model ECM 830 |
| carboxylate-modified FluoSpheres™ microspheres (0.2 μm, blue fluorescing (365/415) | Thermo Fisher Scientific | Cat#F8805 |
| BaseScope™ Duplex Detection Reagent Kit | ACDBio | Cat#323800 |
| Mendeley DATA | Mendeley data | Reserved DOI: 10.17632/9mckw4t9f3.2 |
| Raw and analyzed data of RNA-seq | Gene Expression Omnibus | https://www.ncbi.nlm.nih.gov/geo/query/acc.cgi?acc=GSE288599 Username:reviewer_pxd060131@ebi.ac.uk Password:cR9FHFPHORrP |
| Protein interaction IP-MS data | PRIDE | Project accession: PXD060131 Token:cR9FHFPHORrP |
| Behavior dataset | BioImage Archive | https://www.ebi.ac.uk/biostudies/studies/S-BIAD2140 |

## Methods and protocols

### Animals

All animal experiments were conducted according to protocols approved by the Academia Sinica Institutional Animal Care & Utilization Committee. All mice were generated on a C57BL/6J genetic background. Both sexes of mice were used in experiments. Mice were housed for harem breeding when necessary (one male and two females) and maintained in a 14-h light/10-h dark cycle in sterile ventilated cages with access to food and water *ad libitum* at IMB animal facilities. Conditional paxillin-floxed mice (PXN^fl/fl^; PXN cKO) and stop-floxed paxillin^S119A^ knock-in (PXN^S119A/+^; PXN S119A cKI) mouse lines were created by a CRISPR/Cas9 approach targeting *PXN* gene exons 2 and 1 with flanking Loxp sites, respectively (see Appendix Fig. S1A). Restriction fragment length polymorphism (RFLP) and sequencing were employed to validate CRISPR edits, and no off-target mutations were detected. Heterozygous mice were backcrossed for more than ten generations on a C57BL/6J background. PXN^S119A/+^ mice were maintained as a heterozygous colony. Lines of *PXN^S119A/+^* or *PXN^fl/+^* mice were crossed, each with either Nestin-Cre/ER^T2^ (JAX Stock # 016261) or with Fos^2A-iCreER^ (TRAP2) (JAX Stock # 030323) mice. Doubly heterozygous mice were bred with each other to obtain a designated conditional paxillin S119A cKI and paxillin cKO (PXN^S119A/fl^) mouse line, which exhibited the expected Mendelian ratio. Specifically, for TRAP2-driven manipulations, TRAP2; Paxillin^S119A/+^ mice were crossed with TRAP2; Paxillin^fl/fl^ or TRAP2;

Paxillinfl/+ mice. This breeding strategy ensured that all possible experimental and control genotypes were present within the same litters. Controls and experimental animals were littermates and underwent identical tamoxifen administration and handling. A shared control group was used across comparisons to maintain statistical power and genotype matching while avoiding redundant controls. Data were collected until each group included at least eight biological replicates.

To prepare tamoxifen stock, 20 mg/mL tamoxifen (Sigma; Cat. #T5648) in corn oil (Sigma; Cat. #C8267) was dissolved by repeated sonication at 30 °C for 5 min and vortexing until complete dissolution was achieved. The working tamoxifen solutions (10 mg/mL) were always used the day they were prepared. To induce neuron-specific (Nestin line) or activity-dependent (TRAP2 line) paxillin S119A cKI and/or paxillin cKO within a sensitive developmental window, mice received intraperitoneal (i.p.) injections of tamoxifen (25 μg/g body weight in corn oil) every other day for three doses, starting at postnatal day 5.

### Cell culture

Primary cortical neurons were prepared from rat embryos at embryonic day (E) E18 and cultured in Neurobasal medium (Gibco; Cat. #21103049) supplemented with Gem21 NeuroPlex (Gemini Bio-Products; Cat. #400-160-010) (Dotti et al, 1988; Hsu et al, 2015). The Neuro2a mouse neuroblastoma line (CCL-131) was obtained from ATCC and was cultured in MEM (Gibco; Cat. #11090-81 or Corning; Cat. #10-010-CM) supplemented with 10%

FBS (Biological Industries) and 1 mM Sodium pyruvate (Gibco; Cat. #11360-070). Cell cultures were regularly confirmed to be mycoplasma-free.

### Plasmids, antibodies, and materials

pcDNA3-cdk5GFP, pcDNA3-dnCDK, and pcDNA3.1-P25C-GFP plasmids were gifts of Li-Huei Tsai (Addgene plasmids # 1346, #1344, and #1343), paxillin-pEGFP was a gift of Rick Horwitz (Addgene plasmid # 15233) (Laukaitis et al, 2001). Potential paxillin phosphorylation sites were identified by mass spectrometry proteomic analysis. A paxillin siRNA pool containing two siRNAs (paxillin C: 5′-GAUGAGCAGUCCACAGCGA -3′, paxillin D: 5′-CAGCCUAGUGGUUCCAGAU -3′) and a scramble siRNA (Acell non-targeting siRNA #1: 5′-UGGUUUACAUGUCGACUAA-3′) were purchased from Dharmacon (Lafayette, CO). S-to-A or S-to-D point mutations of paxillin[S119] were introduced using the Quik-Change® II XL site-directed mutagenesis protocol (Stratagene, La Jolla, CA) and Flag-paxillin (a gift from Rick Horwitz; Addgene plasmid # 15212) (Webb et al, 2005). A polyclonal antibody recognizing phosphorylated paxillin Ser119 was custom-made by LTK BioLaboratories (Taipei, Taiwan), raised by immunizing rabbits with synthetic phosphopeptides (VGEEEHVY[pSer119] FPNKQKSA, and then subjected to peptide affinity purification and cross-adsorption to ensure antibody specificity and affinity.

### Shotgun proteomic analysis

For mass spectrometry sample preparation, lysates of rat primary neuronal cultures were prepared at DIV7, after a 2 h incubation with 20 ng/mL BDNF (PeproTech). To inhibit nuclear import, cells were pretreated 16 h with importazole (5 uM; Cayman Chemical) followed by addition of 20 ng/mL BDNF for 2 h. To define paxillin phosphorylation sites, snap-frozen brain, liver, and heart tissues were homogenized 1–2 min in a pre-chilled 25 ml tissue grinding jar (RETSCH; Cat. #15739089) with a 2 cm grinding ball (RETSCH; Cat. #10324771) using a mixer mill (RETSCH MM400) at 25 Hz. The resulting powder was dissolved in 10 ml lysis buffer (0.5% NP-40, 50 mM Tris, pH 8.0, 150 mM NaCl, 1 tablet EDTA-free protease inhibitors). Tissue debris was removed by centrifugation at $2000 \times g$ for 10 min at 4 °C. Supernatants were ultracentrifuged for 80 min at $100,000 \times g$ at 4 °C in a SW 41 Ti swinging-bucket rotor (Beckman). Supernatants were filtered sequentially through 0.8 μm and 0.4 μm filters and then precleaned using a protein G sepharose (Roche) cartridge. One milligram of paxillin polyclonal antibodies (Invitrogen, CAT# PA5-34910) were loaded onto the cartridge and incubated 1 h at room temperature. Paxillin IP'd from lysates was separated on 10% SDS-PAGE gels and stained for 20 min with Coomassie blue under gentle agitation.

After gel destaining, bands corresponding to paxillin and associated proteins were visualized and excised for in-gel digestion overnight in modified trypsin/Lys-C (Promega) at 37 °C. Peptide mixtures were then subjected to shotgun proteomic identification by using NanoLC−nanoESI-MS/MS on a nanoAcquity system (Waters, Milford, MA) connected to the Orbitrap Elite hybrid mass spectrometer (Thermo Electron, Bremen, Germany) equipped with a PicoView nanospray interface (New Objective, Woburn, MA). Peptide mixtures were loaded onto a 75 μm ID, 25 cm length C18 BEH column (Waters, Milford, MA) packed with 1.7 μm particles with a pore size of 130 Å and separated using a 60 min segmented gradient that increased solvent B (0.1% formic acid in acetonitrile)

from 5 to 25% in 55 min, followed by 25 to 35% in 5 min, at a 300 nl/min flow rate and a column temperature of 35 °C. Solvent A was 0.1% formic acid in water. The mass spectrometer was operated in data-dependent mode. Briefly, survey full scan MS spectra were acquired in the Orbitrap (m/z 350–1600) with resolution set to 120 K at m/z 400 and automatic gain control (AGC) target at 106. The 20 most intense ions were sequentially isolated for CID MS/MS fragmentation and detection in the linear ion trap (AGC target at 104) with previously selected ions dynamically excluded for 60 s. Ions with single and unrecognized charge state were also excluded.

### Proteome discoverer label-free quantification

MS and MS/MS raw data were processed by Proteome Discoverer (v 3.0.1.27; Thermo Scientific, Waltham, MA, USA) and searched against the Swiss-Prot protein sequence database with the Mascot server (v.2.8.2; Matrix Science, Boston, MA, USA). Taxonomy was set as Rattus. Search criteria used were trypsin digestion, static modifications set as carbamidomethyl (C), phosphorylation (STY), variable modifications set as oxidation (M) and allowing up to 2 missed cleavages, mass accuracy of 10 ppm for the parent ion and 0.6 Da for the fragment ions mass tolerance. Label-free quantification was performed in Proteome Discoverer without normalization, using both unique and razor peptides to calculate protein abundance. A pairwise ratio-based approach was employed to compare protein levels, and a background-based t-test was conducted to calculate ratio $p$ values.

### RNA isolation and quantitative real-time PCR (qRT-PCR)

Total RNA was extracted from primary cultured neurons or brain tissue using TRIzol reagent (Invitrogen, Cat. #15596018) and purified with a RNeasy Kit (Qiagen) based on the manufacturer's instructions. Sample quality and quantity were assessed using a Nanodrop 2000 system (Thermo Fisher Scientific) to verify the A260/A230 and A260/A280 ratios. Samples with A260/A280 ratios within the 1.8–2.0 range were considered high quality. Subsequently, 600 ng RNA was reverse transcribed using SuperScript IV Reverse Transcriptase (Invitrogen, #18080044) and RNaseOUT Recombinant Ribonuclease Inhibitor (Invitrogen, #10777019).

Quantitative PCR was performed with LightCycler 480 SYBR Green I Master (Roche, Cat. #04887352001), employing the following amplification conditions: initial pre-incubation at 95 °C for 3 min, followed by 36 cycles of 95 °C for 30 s, 55 °C for 30 s, and 72 °C for 60 s. All experiments were conducted in triplicate. Relative gene expression was determined using the 2-ΔΔCt method and normalized to the control sample, with the data presented as fold changes in gene expression. Mouse acidic ribosomal phosphoprotein P0 (36B4) served as the endogenous control for sample normalization.

### RNA sequencing, cassette exon analysis, and Nanopore long-read sequencing

Total RNA was extracted using a RNeasy Mini Kit (Qiagen; Cat. #74104), and its concentration was determined using a Nanodrop 2000 spectrophotometer and a Qubit fluorometer (both from Thermo Fisher Scientific). RNA quality was evaluated using a 2100 Bioanalyzer (Agilent). Only samples with an RNA integrity number (RIN) >7.5 were further analyzed. One microgram total RNA per sample was used for library preparation using the SureSelect XT HS2 mRNA Library Prep Kit (Agilent Technologies, USA) on the Agilent Bravo Automated Liquid Handling Platform. Libraries were

sequenced on an Illumina NovaSeq 6000 platform using 148-bp paired-end reads, generating over 60 million reads per sample. Paired-end raw reads were aligned to the mouse reference genome (GCF_000001635.26; GRCm38.p6) using STAR (v2.7.10), and gene-level read counts were obtained using RSEM (v1.3.3). Subsequent transcriptomic data analysis—encompassing normalization, differential gene expression testing, and K-means clustering—was performed using the BBSC portal, a cloud-based R platform providing specialized bioinformatics applications for RNA-seq analysis from the Bioinformatics Core of the Institute of Molecular Biology, Academia Sinica. Aligned BAM files were analyzed with rMATS (v4.1.2) to quantify exon inclusion and exclusion events. For each cassette exon, rMATS calculates a percent spliced-in (PSI or ψ) value, reflecting exon inclusion across samples. The relative inclusion rate was determined as $[(PSI_{t=n} - PSI_{\mu}) / PSI\_\sigma]$, where $PSI\mu$ is the mean PSI and $PSI_{\sigma}$ the standard deviation of PSI values obtained from DIV0, DIV3, DIV5, and DIV10 neuronal cultures. The "switching time" for an alternative splicing event was defined as the point at which the sign of the next PSI value reversed. Fisher's exact test was used to assess the statistical significance of trans-acting RBPs associated with exons and adjacent 200-base-pair intronic sequences affected by paxillin KD, focusing on 46 RBPs with eCLIP data available in oRNAment and CLIPdb. Gene Ontology (GO) terms analysis was executed with a P-value threshold of 0.05, applying the Benjamini–Hochberg method for False Discovery Rate (FDR) correction. RNA-seq data have been deposited in the GEO repository with GEO accession number GSE288599.

For Nanopore sequencing, 120 ng of total RNA per sample was used for cDNA synthesis using the SMART-Seq mRNA with UMI kit (Takara, Cat. #634763), following the manufacturer's instructions. Amplification was performed using 8 PCR cycles, and cDNA fragments shorter than 500 bp were excluded using 0.5× AMPure XP beads (Beckman Coulter). Full-length cDNAs were then prepared for long-read sequencing using the Oxford Nanopore Ligation Sequencing Kit with Native Barcoding (Kit 24 V14, SQK-NBD114.24). Barcoded cDNA libraries were loaded onto a PromethION flow cell (FLO-PRO114M, pore version R10.4.1) and sequenced on a PromethION P2 Solo platform for 72 h. Reads were aligned to the mouse reference genome (mm10) using minimap2 with the parameters -ax splice -t 36 -MD --secondary = yes -Y. Alignments were filtered to retain only primary alignments with a mapping quality (MAPQ) > 50, and then sorted and indexed using SAMtools. Transcript-level read counts were estimated using IsoQuant (v3.6.3) with the mm10 reference genome and the following parameters: "--data_type nanopore --report_novel_unspliced true --report_canonical all --count_exons".

### Protein lysate preparation and immunostaining

Cells were transfected using a lentivirus-based system or Lipofectamine 2000 (Invitrogen) based on the manufacturer's instructions. Cells were lysed in RIPA buffer (Sigma-Aldrich; Cat. #R0278) containing a complete protease inhibitor cocktail (Roche; Cat. #4693116001) and the phosphatase inhibitor PhosSTOP (Roche; Cat. #4906837001). For immunostaining and immunohistochemistry, cells or 50-μm brain cryosections were fixed 12 min in 4% PFA, permeabilized 12 min in 0.3% Triton X-100, and blocked 1–2 h with 3% BSA. Fixed sections were processed for immunostaining using standard procedures. Images were analyzed and

processed for presentation using brightness and contrast adjustments with ImageJ software (National Institutes of Health) (Rossner and Yamada, 2004; Schneider et al, 2012).

### Synaptosome fractionation

Fresh mouse hippocampi were homogenized using a Dounce homogenizer in 0.3 mL of ice-cold buffered sucrose solution containing 10 mM HEPES (pH 7.5), 320 mM sucrose, 10 mM DTT, 5 mM EDTA, and 1 mM EGTA, supplemented with 1× cOmplete™ Protease Inhibitor Cocktail (Roche, Cat. #11697498001) and 1× PhosSTOP™ Phosphatase Inhibitor Cocktail (Roche, Cat. #4906837001). The homogenate was centrifuged at $800 \times g$ for 10 min at 4 °C to pellet nuclei and cell debris (P1; "Pellet"). The supernatant was transferred to a fresh tube and centrifuged again at $800 \times g$ for 10 min at 4 °C, followed by a second centrifugation at $9200 \times g$ for 15 min at 4 °C. The resulting pellet (P2; "Syn") was collected as the crude synaptosome fraction.

### Detection of Snap25 exons 5a and 5b in mouse brain slices

P11.5 and P13 mice were transcardially perfused with 20 mL 4% PFA in PBS. Brains were dissected, post-fixed in 10% neutral-buffered formalin (NBF) for 24 h at room temperature, and then processed 24 h for paraffin embedding. Serial sections (5 μm) were cut, deparaffinized in xylene and 100% ethanol, and air-dried before incubation with hydrogen peroxide ($H_2O_2$). Target retrieval was performed according to the manufacturer's instructions using RNAscope™ Target Retrieval Reagents (Advanced Cell Diagnostics, Inc), followed by digestion with Protease III (BaseScope™ Duplex Reagent Kit, Advanced Cell Diagnostics, Inc) at 40 °C for 30 min. Brain sections were hybridized for 2 h at 40 °C in a HybEZ oven (Advanced Cell Diagnostics) with either the Snap25 exon 5a probe (BA-Mm-Snap25-tv2-1zz-st-C2) diluted 1:50 or the Snap25 exon 5b probe (BA-Mm-Snap25-tv1-1zz-st-C2) diluted 1:100, each combined with a positive control probe solution (Mm-PPIB-3ZZ). Amplification was carried out using AMP1 through AMP12 reagents (BaseScope Duplex Reagent Kit), with AMP7 incubation reduced to 15 min. Red signals served to detect expression of Snap25 exon 5a or 5b, and green signals marked the positive control. Sections were counterstained and mounted with Vectamount mounting medium (Vector Laboratories), ready for imaging.

### Field potential recordings in hippocampal slices

Mice (P14-P18 for LTD; P35 for all other experiments) were deeply anesthetized with isoflurane before decapitation. Hippocampal slices (450 μm thick) were prepared and maintained in artificial cerebrospinal fluid (aCSF) composed of 119 mM NaCl, 2.5 mM KCl, 26.2 mM NaHCO₃, 1 mM NaH₂PO₄, 1.3 mM MgSO₄, 2.5 mM CaCl₂, and 11 mM glucose, continuously bubbled with 95% O₂/5% CO₂ at room temperature. A surgical cut was made to isolate CA1 by severing CA1-CA3 connections. Picrotoxin (PTX, 100 μM) was added to the aCSF to block GABAA receptor–mediated inhibitory currents. Extracellular field potentials were recorded in the CA1 stratum radiatum using borosilicate glass microelectrodes (Warner Instruments G150F-4) filled with 3 M NaCl (3–4 MΩ resistance). Schaffer collateral fibers were stimulated with 40-μs square-wave pulses delivered through a bipolar tungsten electrode placed 100–400 μm from the recording site. Pulses, ranging from 10 to 200 μ for a maximal duration of 150 μsec, were delivered through

a stimulus isolator unit (WPI). Recordings were performed using patch-clamp amplifiers (Molecular Devices Axon MultiClamp 700B), digitized and acquired with WinWCP software. Signals were low-pass filtered at 1 kHz and sampled at 20 kHz using the Clampfit module (pClamp v10.5; Molecular Devices). Recordings were maintained for up to 4 h post-slicing. For input-output (I-O) relationship experiments, slices were stimulated eight times every 15 s with incremental current amplitudes (from 0.2 to 1.1 mA, in 0.1-mA increments). Presynaptic volley amplitude and initial fEPSP slope were measured to assess the I-O curve. For paired-pulse facilitation (PPF) experiments, pairs of pulses repeated six times were delivered at varying intervals. Slopes of the first and second fEPSP were measured to calculate the paired-pulse ratio. Long-term potentiation (LTP) experiments: Following a 10-min baseline recording (15-s intervals) of fEPSPs (0.5–1.0 mV amplitude), LTP was induced by high-frequency stimulation (HFS; 3 trains of 100 Hz for 1 s). Single-pulse stimulation (15-s intervals) resumed for 60 min to assess potentiation. Long-term depression (LTD) experiments: Following a 10-min baseline recording (15-s intervals), LTD was induced by low-frequency stimulation (LFS; 1 Hz for 15 min). Post-LFS fEPSPs were recorded for 60 min to quantify the magnitude of depression.

## Mouse behavior assessments

Mice used for behavioral experiments were selected based on genotype and littermate availability to ensure balanced and unbiased group assignment. All animals were age-matched and free of overt health issues at the time of testing. Both male and female mice were included in behavioral assessments, with gender balanced across experimental groups unless otherwise noted. No sex-based differences were observed in preliminary analyses; therefore, data from both sexes were combined. Beginning at P5, test mice received IP tamoxifen injections (25 μg/g body weight in corn oil) every other day for three doses to induce paxillin S119A cKI and/or paxillin cKO before mice entered a sensitive phase of neuronal development. After this regimen and standard weaning, we conducted behavioral evaluations using four paradigms based on institutional and ethical guidelines. Test mice were brought to the behavioral testing room in test conditions and allowed to habituate for at least 16 h. During testing, overhead lighting was minimized to avoid anxiogenic effects, and the light intensity was measured at the center of all apparatuses to ensure evenly lighting. Video analyses were performed using ezTrack (Pennington et al, 2019), a free open-source behavioral tracking pipeline.

In the open-field test, each mouse (at P35–P45) was placed in a square polyvinyl chloride arena (40 cm × 40 cm) for 10 min to measure overall locomotor activity, anxiety-like behavior (percentage of time spent in center vs. periphery), and exploratory patterns.

The three-chamber social behavior test was used to assess sociability and preference for social novelty by quantifying time spent interacting with an unfamiliar mouse compared to an inanimate object or a familiar mouse. The apparatus consisted of three connected compartments (each 20 cm × 40 cm), separated by transparent walls with small openings that could be opened or closed. In the habituation phase, test mice (at P35–P45) were placed in the central chamber and allowed 10 min of free exploration of the three compartments with side chambers empty. For the sociability test, an age-, sex-, and strain-matched unfamiliar wild-type mouse (Stranger 1) was placed under a wire pencil cup in one

side chamber, while a paper ball (under a similar pencil cup) was in the opposite. Test mice were then given 10 min to explore both side chambers, with time spent in each recorded. Then, the paper ball was replaced with another unfamiliar wild-type mouse (Stranger 2), while Stranger 1 remained in its original compartment. Test mice again had 10 min of free access to all three compartments, and time spent interacting with each mouse was assessed to evaluate social novelty preference.

The novel object recognition (NOR) test was carried out in a polyvinyl chloride arena (40 cm × 40 cm) designed to assess the animal's memory of previously encountered objects and consisted of habituation, familiarization, and test phases. During the first test, mice (at P65–P75) were allowed to explore the arena for 5 min to minimize novelty-related anxiety. On the following day, familiarization was conducted by placing two identical Falcon tubes at opposite corners of the same half of the arena. Test mice were then placed in the center and permitted to explore for 8 min, and the time spent investigating each tube was recorded. Two hours later, test mice were returned to the arena for the test phase, in which one tube (familiar object) was replaced with a novel object (a LEGO), and the time spent interacting with each over 8 min was used to calculate a recognition index, indicating memory for the familiar object. To prevent bias from olfactory cues, the arena and objects were thoroughly cleaned with 70% ethanol between trials. The NOR test was conducted under both baseline conditions and after imposing subthreshold stress via a physical restraint protocol, in order to assess stress-induced changes in performance. On 2 consecutive days, mice (at P90) were individually placed in a ventilated plastic bag for two hours, limiting movement while ensuring adequate ventilation. Following each session, mice were returned to their home cages and allowed free access to food and water.

The Y-maze spontaneous alternation test was used to assess short-term spatial working memory. The Y-maze consisted of three identical arms, each marked by a unique visual cue. In the training phase, one arm was blocked, allowing mice to freely explore only the "home" or "familiar" arm for 5 min. Then, mice were returned to the home cage for 1 h. In the test phase, mice were reintroduced to and allowed free access to all three arms, including a newly opened "novel" arm, for 5 min. The percentage of alternations was calculated by dividing the number of successful alternations by the total number of possible alternations (i.e., total arm entries minus two). Time spent in each arm and the pattern of arm entries (i.e., spontaneous alternation) were recorded to assess short-term spatial working memory. To eliminate olfactory cues, the maze was thoroughly cleaned with 70% ethanol between trials.

## Statistical analysis

All statistical analyses were performed using GraphPad Prism. For comparisons between two groups in biochemical assays, N/C ratio, RT-qPCR, and animal behavior tests, multiple unpaired two-tailed $t$-tests were used unless otherwise specified. For comparisons across multiple groups in immunofluorescence staining assays and ePhys experiments, one-way ANOVA followed by post hoc tests (e.g., Tukey's or Dunnett's) was applied. In analyses involving multiple comparisons across repeated measures (e.g., paired-pulse ratios), multiple unpaired $t$-tests were performed. For linear trend comparisons (e.g., input-output curves), linear regression analysis was used to assess differences in slope between groups. Statistical significance was

defined as $p < 0.05$. The specific statistical tests used for each figure are detailed in the corresponding figure legends.

## Data availability

The datasets produced in this study are available in the following databases: Protein interaction IP-MS Data: PRIDE PXD060131. RNA-seq and Nanopore data: Gene Expression Omnibus GSE288599. Biochemistry dataset: Mendeley Data (https://doi.org/10.17632/9mckw4t9f3.2). Imaging dataset: BioImage Archive S-BIAD2140. Behavior dataset: BioImage Archive S-BIAD2143.

The source data of this paper are collected in the following database record: biostudies:S-SCDT-10_1038-S44318-025-00560-8.

## Peer review information

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

## Acknowledgements

We thank Mu-Ming Poo for discussions and critical reading of the manuscript. We thank the Genomics Core, Bioinformatics Core Facility (Institute of Molecular Biology, Academia Sinica) for RNA-Seq and rMATs analysis. We are grateful to the IMB Transgenic Core Facility for their work in generating paxillin-floxed and stop-floxed paxillin S119A knock-in mice. This work was supported in part by a grant from the National Science and Technology Council of Taiwan (113-2311-B-001-011-), and Academia Sinica Grand Challenge Program Seed Grant (AS-GCS-112-L02). We also thank Drs. Shu-Yu Lin and Yi-Yun Chen for technical assistance in LTQ-Orbitrap data acquisitions at the Academia Sinica Common Mass Spectrometry Facilities for Proteomics and protein modification analysis located at the Institute of Biological Chemistry, Academia Sinica, supported by Academia Sinica Core Facility and Innovative Instrument Project (AS-CFII-111-209).

## Author contributions

**Chien-Hsin Chu**: Data curation; Formal analysis; Validation; Investigation; Visualization; Methodology; Writing—original draft; Writing—review and editing. **Guan-Zhu Pan**: Conceptualization; Data curation; Formal analysis; Validation; Investigation; Visualization; Methodology; Writing—original draft; Writing—review and editing. **Ching-Yen Tsai**: Conceptualization; Resources; Investigation; Methodology. **Chen-Hsin Albert Yu**: Data curation; Software; Formal analysis; Methodology. **Hsin-Nan Lin**: Data curation; Software; Formal analysis; Methodology. **Hung-Lun Chiang**: Conceptualization; Resources; Methodology. **Chen Chen**: Data curation; Formal analysis; Validation; Investigation; Visualization; Methodology. **Li-Ching Chen**: Data curation; Formal analysis; Investigation; Methodology. **Xuan-Dieu Thi Pham**: Data curation; Methodology. **Chien-Ling Lin**: Conceptualization; Resources; Funding acquisition; Investigation; Methodology. **Pei-Lin Cheng**: Conceptualization; Resources; Data curation; Software; Formal analysis; Supervision; Funding acquisition; Validation; Investigation; Visualization; Methodology; Writing—original draft; Project administration; Writing—review and editing.

Source data underlying figure panels in this paper may have individual authorship assigned. Where available, figure panel/source data authorship is listed in the following database record: biostudies:S-SCDT-10_1038-S44318-025-00560-8.

## Disclosure and competing interests statement

The authors declare no competing interests.

# Expanded View Figures

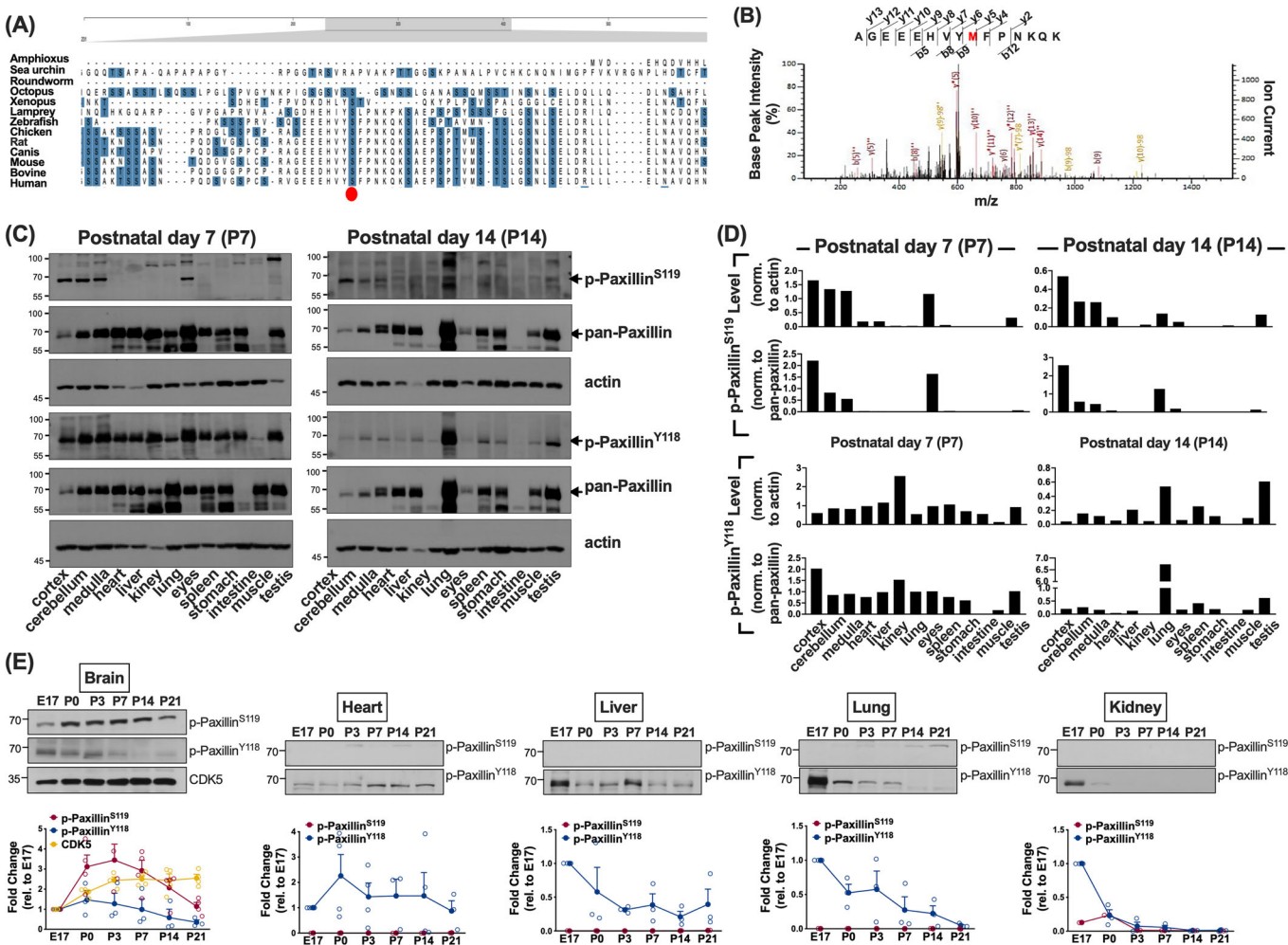

**Figure EV1. Identification of a brain-enriched phosphorylation site at paxillin serine119.**

(A) Paxillin S119 is phylogenetically conserved. Alignment of sequences found around paxillin S119 (red dot), illustrating conservation across indicated species. (B) Mass spectrometric identification of phosphopeptides in paxillin protein IP'd from P14 mouse brain lysates. The phosphopeptide AGEEEHVYMFPNKQK (phospho-paxillin$^{S119}$; p-Paxillin$^{S119}$) was identified by precursor ion scanning for neutral loss of phosphoric acid (M-H3PO4). M, peptide precursor ion. (C) Western blots show p-Paxillin$^{S119}$ enrichment in mouse brain lysates at P7 and P14. Lysates were prepared from multiple tissues, as indicated. The pan-Paxillin blots shown here were derived from membranes that were first probed with either anti–p-Paxillin$^{S119}$ or anti–p-Paxillin$^{Y118}$, and after antibody stripping were subsequently re-probed with anti–pan-Paxillin to serve as total Paxillin controls for the same samples. (D) Quantification of data shown in blots in (C). (E) Developmental profiles of p-Paxillin$^{S119}$ and p-Paxillin$^{Y118}$ across multiple tissues. Western blots showing p-Paxillin$^{S119}$ and p-Paxillin$^{Y118}$ levels over various developmental stages in various tissues. Graph shows average protein intensity ± SEM, compared with corresponding levels at E17 ($n = 3$–4 mice per group). Source data are available online for this figure.

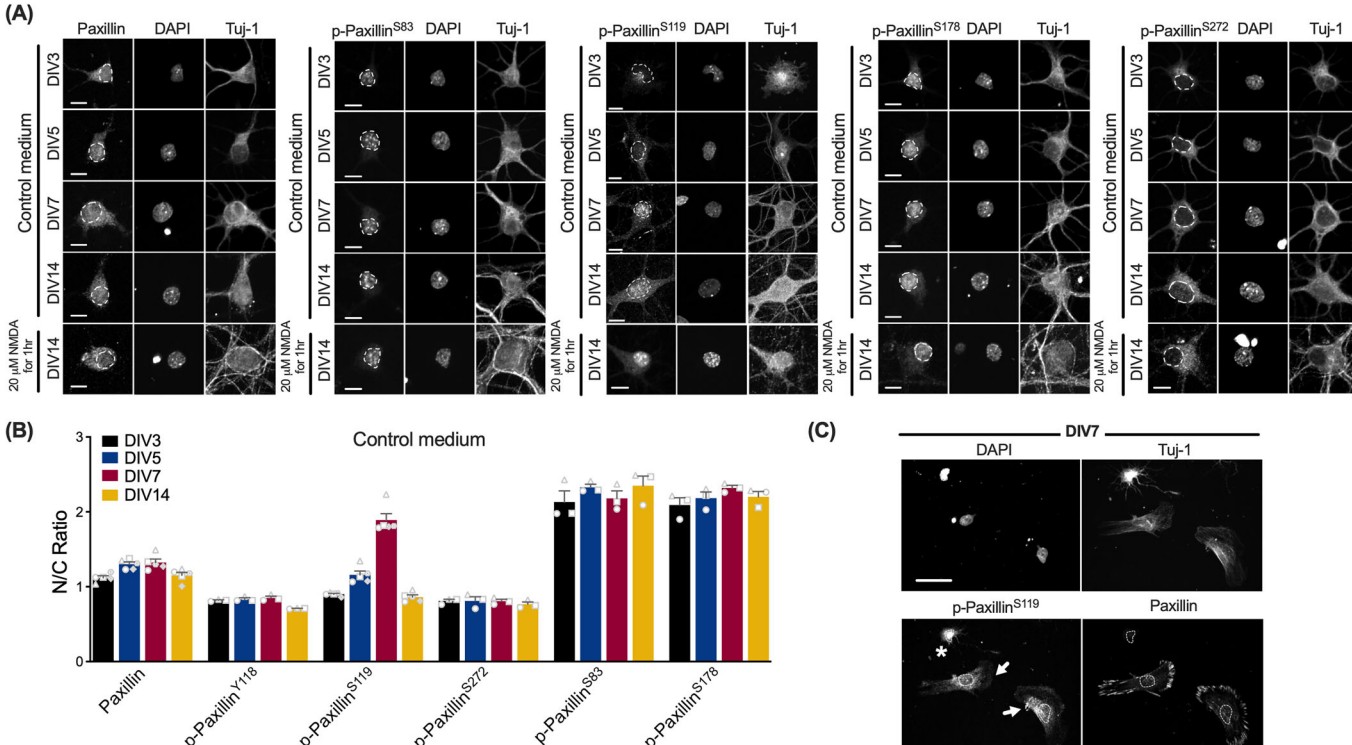

**Figure EV2. Site-specific phosphorylation determines paxillin subcellular localization in neurons and glia.**

(A, B) DIV7 neurons show transient increases in nuclear p-Paxillin$^{S119}$ levels. (A) Representative images of rat primary neuronal cultures stained with antibodies against different phosphorylated forms of paxillin at the indicated time points. Scale bar, 5 μm. (B) Histograms summarizing the nucleus-to-cytoplasm (N/C) ratios of various phosphorylated paxillin forms from experiments similar to those in (A). Data represent mean ± SEM ($n = 3$–5 independent cultures; >6 cells per group). (C) Differential distribution of p-Paxillin$^{S119}$ in neurons vs. glia. Representative image of a DIV7 primary neuronal culture showing p-Paxillin$^{S119}$ signals in a neuron (asterisk) versus in neuroglia (arrows). Scale bar, 10 μm.

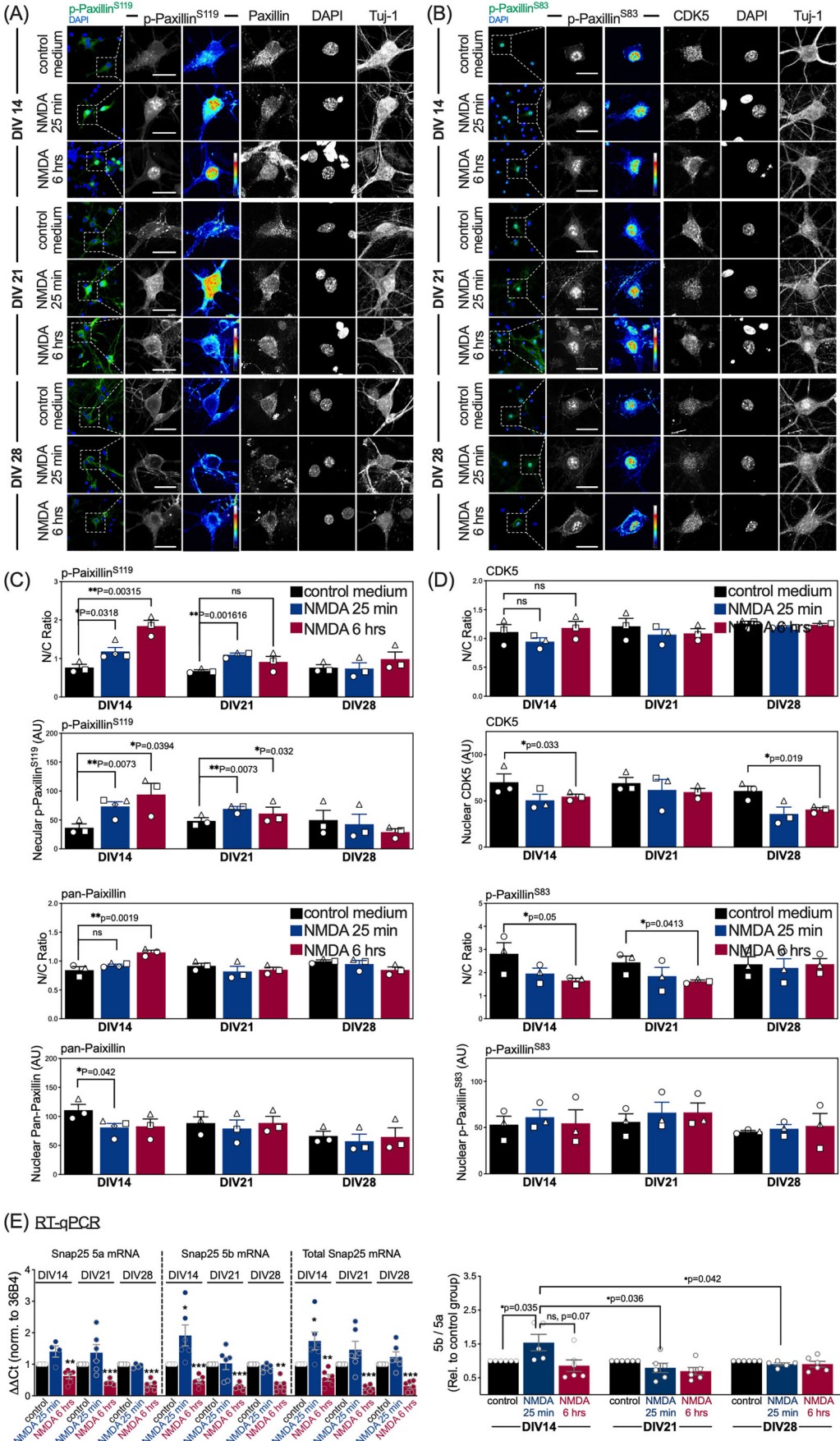

◀ **Figure EV3.  Both NMDA-induced nucleocytoplasmic translocation of p-Paxillin$^{S119}$ and Snap25 exon 5b-to-5a inclusion in neurons decrease after DIV14.**

(A–D) Subcellular distribution of endogenous p-Paxillin$^{S119}$, pan-Paxillin, p-Paxillin$^{S83}$, and CDK5 in primary cortical neurons at the indicated days in vitro (DIV), following acute ("25 min") or prolonged ("6 h") stimulation with 20 μM NMDA. (A, B) Representative images of neurons cultured in standard Neurobasal medium under the indicated conditions. Cells were immunostained for p-Paxillin$^{S119}$ (A, green), pan-Paxillin, p-Paxillin$^{S83}$ (B, green), CDK5 and the neuronal marker Tuj-1, with DAPI counterstain (blue) at the indicated conditions and time points. Scale bar, 15 μm. The intensity of p-Paxillin$^{S119}$ is also displayed using a "royal" look-up table, highlighting the range from low (blue) through medium (green) to high (red) intensity. (C, D) Histogram summarizing nuclear to cytoplasmic (N/C) ratio and nuclear abundance of p-Paxillin$^{S119}$, pan-Paxillin, p-Paxillin$^{S83}$, and CDK5 in primary neuronal cultures at indicated conditions and time points. Data represent mean ± SEM ($n = 3$ independent cultures; >30 cells per group; *$p < 0.05$, **$p < 0.01$, ***$p < 0.001$, ns not significant compared to control, by one-way ANOVA with Dunnett's multiple comparisons test). (E) Real-time qPCR analysis of Snap25 exon 5a, 5b, and total mRNA expression in neuronal cultures at the indicated time points under control or NMDA treatment conditions. Data represent mean ± SEM ($n = 5$–6 cultures per group; *$p < 0.05$, **$p < 0.01$, ***$p < 0.001$, ns not significant compared to DIV14 NMDA 25 min group, multiple $t$-tests). Source data are available online for this figure.

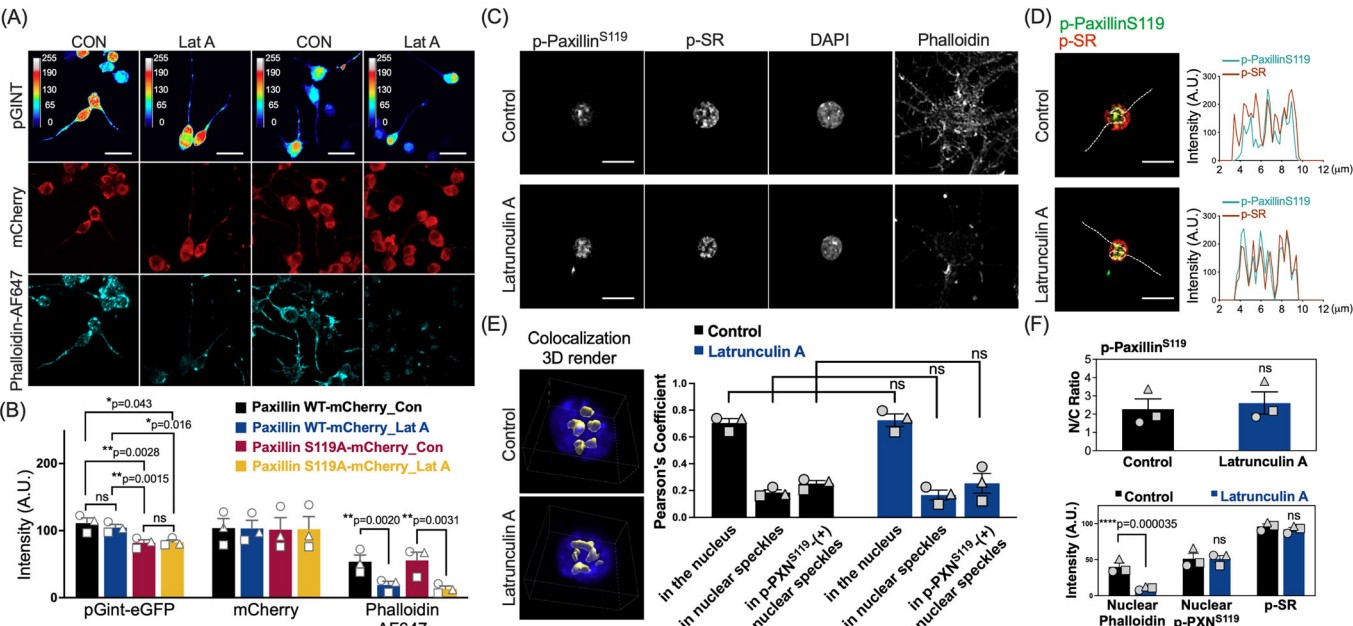

**Figure EV4.  Disruption of cytoplasmic actin polymerization does not alter mRNA splicing efficiency.**

(A) Representative images showing differentiated Neuro2A cells transduced with either mCherry-tagged paxillin wild-type (WT) or its S119A variant via lentiviral transduction, followed by transfection with the pGint splicing reporter. Cells were treated with or without the actin polymerization inhibitor Latrunculin A ("Lat A"; 330 nM for 1 h) and then stained with Phalloidin-AF647 to visualize F-actin. eGFP fluorescence intensity is shown using a "royal" look-up table, highlighting the range from low (blue) through medium (green) to high (red) intensity. Scale bar, 20 μm. (B) Plot summarizing average eGFP intensity, mCherry signal, and Phalloidin-AF647 staining (±SEM; $n = 21$–30 cells from three independent experiments; ****$p < 0.0001$ compared to WT by Two-way ANOVA with Tukey's multiple comparisons test). (C–F) p-Paxillin[S119] colocalization with phospho-SR proteins (p-SR) is maintained after disruption of actin polymerization. (C) Immunofluorescence images of DIV7 neurons treated with or without latrunculin A (330 nM, 1 h), stained for p-Paxillin[S119], p-SR, DAPI, and Phalloidin. (D) Fluorescence intensity profiles (right panel) across p-Paxillin[S119] puncta show colocalization with p-SR. Scale bar, 10 μm. (E) Left: 3D renderings showing colocalization of p-Paxillin[S119] with p-SR in nuclear speckles. Right: Bar graphs quantify Pearson's correlation coefficients (±SEM; $n = 3$ cultures; 15 cells per condition; ns not significant, by multiple unpaired t-tests) for p-Paxillin[S119] and p-SR colocalization within the nucleoplasm, nuclear speckles, and p-Paxillin[S119]-positive nuclear speckles. (F) Quantification of the N/C ratio of p-Paxillin[S119] (top), and nuclear intensity of p-Paxillin[S119], p-SR, and Phalloidin staining (bottom) in DIV7 neurons with or without Lat A treatment. Data represent mean (±SEM; $n = 3$ independent cultures; >30 cells per group; ****$p < 0.0001$; ns not significant, by multiple unpaired t-tests). Source data are available online for this figure.

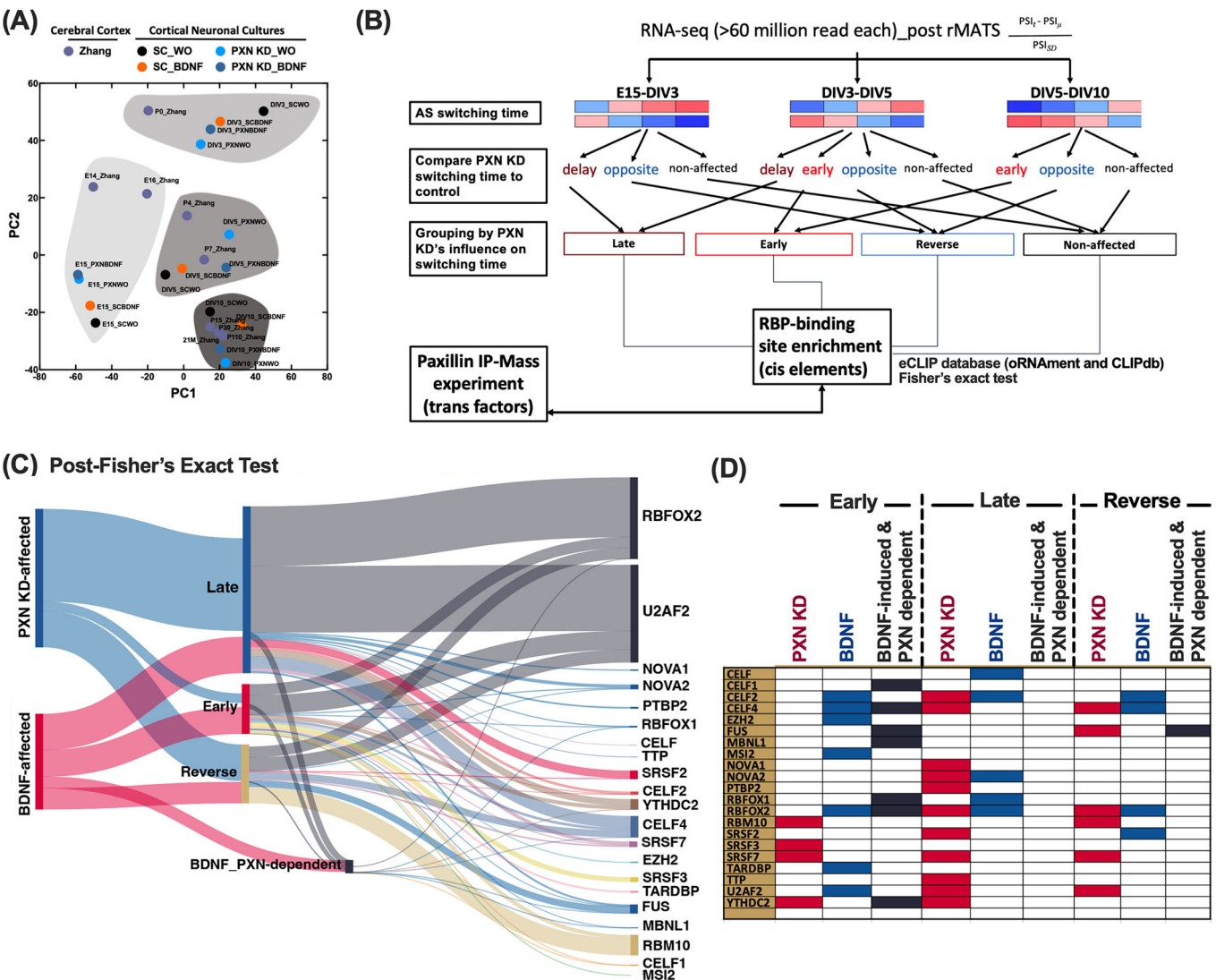

**Figure EV5. Identification of alternative splicing events associated with paxillin knockdown and/or BDNF treatment in primary neuronal cultures.**

(A) PCA analysis of cassette exon profiles. Cassette exon usage patterns from E15 mouse primary neuronal cultures at various DIV were compared with those seen in brain lysates ("Zhang"; dataset from (Weyn-Vanhentenryck et al, 2018) at corresponding postnatal days. Note that DIV3 clustered with P0, DIV5 with P4 and P7, and DIV19 with >P15. (B) Analysis pipeline for splicing event identification. Schematic illustrating workflow used to detect cassette exon switching events altered by paxillin knockdown ("PXN KD") and/or BDNF treatment, followed by prediction of trans-acting RNA-binding proteins (RBPs). (C) Sankey diagram of splicing time switches and RBPs. Diagram indicating how timing of cassette exon switching is influenced by PXN KD and/or BDNF treatment, and highlighting associated RBPs. Notably, most late-switched cis-elements affected by PXN KD contain U2AF2 and RBFOX2 binding sites. (D) Table listing RBPs significantly linked to the PXN KD- and/or BDNF-affected splicing events compared with those in the non-affected group ($p < 0.05$ by Fisher's exact test).

