## [Peer Review File · The EMBO Journal]

Nuclear paxillin functions as a molecular switch for alternative splicing in neurons during a critical period of brain development

Chien-Hsin Chu, Guan-Zhu Pan, Ching-Yen Tsai, Chen-Hsin Yu, Hsin-Nan Lin, Hung-Lun Chiang, Chen Chen, Li-Ching Chen, Xuan-Dieu Pham, Chien-Ling Lin, and Pei-Lin Cheng

Corresponding author(s): Pei-Lin Cheng (plcheng@imb.sinica.edu.tw)

Review Timeline:

Submission Date:	10th Mar 25
Editorial Decision:	7th Apr 25
Revision Received:	6th Jul 25
Editorial Decision:	5th Aug 25
Revision Received:	8th Aug 25
Accepted:	20th Aug 25

Editor: Ioannis Papaioannou

Transaction Report:

Dear Dr. Cheng,

Thank you for submitting your manuscript EMBOJ-2025-120719 for consideration by The EMBO Journal. It has now been seen by three experts in the field, and we have received the full set of their detailed and informative comments, which you can find below.

As you will see, all three referees indicate interest in the study and the findings, and explain that this is a comprehensive work offering novel and significant insights that would be of interest to the field. They also identify, however, a number of limitations and raise several concerns that should be addressed with additional experimental work, and they provide constructive suggestions for strengthening the manuscript further and increasing its impact on the field.

Given the referees' positive comments and recommendations, I would like to invite you to submit a thoroughly revised version of your manuscript taking the referees' suggestions on board, along with a detailed point-by-point response addressing all referees' comments. I should add that it is The EMBO Journal policy to allow only a single round of major revision, and acceptance of your manuscript will therefore depend on the completeness of your responses in this revised version. Please let me know if you have any questions or comments that you would like to discuss with me. If there are any major points you do not agree with or cannot address during your revision, I would encourage you to share them with me as early as possible to discuss how to proceed further in the most efficient way.

We generally allow three months as standard revision time (July 6, 2025). As a matter of policy, competing manuscripts published during this period will not negatively impact our assessment of the conceptual advance presented by your study. However, we request that you contact us as soon as possible upon publication of any related work, to discuss how to proceed. Should you foresee a problem in meeting this three-month deadline, please let us know in advance and we will be able to grant an extension.

Thank you for the opportunity to consider your work for publication in The EMBO Journal. I look forward to your revision.

Best regards,

Ioannis

Instructions for preparing your revised manuscript

1. When you are ready to submit the revision, please upload:

- A Word file of the manuscript text (including legends of main Figures, EV Figures and Tables). Please make sure that changes are highlighted (or "tracked") to be clearly visible.

- Individual production-quality figure files (one file per figure). When assembling your figures, please refer to our figure preparation guidelines in order to ensure proper formatting and readability in print as well as on screen:

If the data shown in a figure are obtained from n {less than or equal to} 2, please use scatter plots showing the individual data points.

- i. the name of the statistical test used to generate error bars and P values
- ii. the number (n) of independent experiments (please specify technical or biological replicates) underlying each data point (discussion of statistical methodology can be reported in the Materials and Methods section, but figure legends should contain a basic description of n , P, and the test applied)
- iii. the nature of the bars and error bars (s.d., s.e.m.).

- A point-by-point response to the referees' comments, with a detailed description of the changes made (as a word file). All referees' concerns must be fully addressed and their suggestions taken on board. When preparing your letter of response to the

referees' comments, please bear in mind that this will form part of the Review Process File and will therefore be available online to the community. Please note that you have the possibility to opt out of the transparent process at any stage prior to publication by letting the editorial office know (contact@embojournal.org); if you do opt out, the Review Process File link will point to the following statement: "No Review Process File is available with this article, as the authors have chosen not to make the review process public in this case.". For more details on our Transparent Editorial Process, please visit our website: <https://www.embopress.org/page/journal/14602075/authorguide#transparentprocess>

- Expanded View (EV) files (replacing Supplementary Information) that are collapsible/expandable online. A maximum of 5 EV Figures can be typeset. EV Figures should be cited as "Figure EV1, Figure EV2" etc. in the text, and their respective legends should be included in the manuscript file after the legends of regular figures. See detailed instructions regarding Expanded View files here:

- For the figures that you do NOT wish to display as Expanded View figures, they should be bundled together with their legends in a single PDF file called "Appendix", which should start with a short Table of Contents (including page numbers). Appendix figures should be referred to in the main text as: "Appendix Figure S1, Appendix Figure S2" etc. Please see detailed instructions here: <https://www.embopress.org/page/journal/14602075/authorguide#expandedview>

- A complete author checklist, which you can download from our author guidelines (<https://www.embopress.org/page/journal/14602075/authorguide>). Please note that the checklist will also be part of the Review Process File.

2. Please note that no statistics should be calculated and shown in Figures if $n=2$. Please also note that each p value should be reported as an exact value.

3. Before submitting your revision, primary datasets (and computer code, where appropriate) produced in this study need to be deposited in appropriate public databases (see <https://www.embopress.org/page/journal/14602075/authorguide#dataavailability>). The accession numbers, database, and the specific URLs (links) should be listed in a formal "Data availability" section (placed after Methods), following the example below:

"The RNA-seq datasets produced in this study are available in the following database:
Gene Expression Omnibus GSE46843 (<https://www.ncbi.nlm.nih.gov/geo/query/acc.cgi?acc=GSE46843>)"

*** All links should resolve to a page where the data can be accessed. ***

*** Please remember to provide in the Data availability section of your revised manuscript reviewer passwords if the datasets are not yet public. ***

*** The Data Availability Section is restricted to new primary data that are part of this study. In case you have no data that require deposition in a public database, please state so instead of referring to the database: "Our study includes no data deposited in public repositories." under the heading "Data availability". ***

4. The materials and methods need to be described in the manuscript using our structured methods format, which is now required for all research articles. According to this format, the Methods section includes a single "Reagents and Tools Table" - listing key reagents, experimental models, software and relevant equipment including their sources and relevant identifiers - followed by a "Methods and Protocols" section describing the methods. Please download and fill our Reagents and Tools Table template (.docx), which you can find in our author guide:

<https://www.embopress.org/page/journal/14602075/authorguide#structuredmethods>. When submitting your revised manuscript, please do not include the Reagents and Tools Table in the Methods section of the manuscript but instead upload it as a separate file choosing the file type "Reagent Table".

5. Please check that the title and the abstract of the manuscript are brief, yet explicit, even to non-specialists. The length of the title should not exceed 100 characters, and the abstract should be a single paragraph not exceeding 175 words.

6. Please also note our reference format: <https://www.embopress.org/page/journal/14602075/authorguide#referencesformat>.

8. Please remember: digital image enhancement is acceptable practice, as long as it accurately represents the original data and conforms to community standards. If a figure has been subjected to significant electronic manipulation, this must be noted in the figure legend or in the "Materials and Methods" section. The editors reserve the right to request original versions of figures and the original images that were used to assemble the figure.

9. Our journal encourages inclusion of data citations in the reference list to directly cite datasets that were obtained from public databases. Data citations in the article text are distinct from normal bibliographical citations and should directly link to the database records from which the data can be accessed. In the main text, data citations are formatted as follows: "Data ref: Smith et al, 2001" or "Data ref: NCBI Sequence Read Archive PRJNA342805, 2017". In the Reference list, data citations must be labeled with "[DATASET]". A data reference must provide the database name, accession number/identifiers, and a resolvable link to the landing page from which the data can be accessed at the end of the reference. Further instructions are available at: <https://www.embopress.org/page/journal/14602075/authorguide#referencesformat>.

10. We request authors to consider both actual and perceived competing interests. Please review our policy (<https://www.embopress.org/page/journal/14602075/authorguide#conflictsofinterest>) and update your competing interests statement if necessary. Please name this section 'Disclosure and competing interests statement' and place it after the Acknowledgements section.

11. Please note that all corresponding authors are required to provide an ORCID ID upon submission of a revised manuscript (<https://orcid.org/>). Please find instructions on how to link your ORCID ID to your account in our manuscript tracking system in our Author guidelines (<https://www.embopress.org/page/journal/14602075/authorguide#authorshipguidelines>).

12. We use CRediT to specify the contributions of each author in the journal submission system. CRediT replaces the author contribution section, which should be removed from the manuscript. Please use the free text box to provide more detailed descriptions. See also guide to authors: <https://www.embopress.org/page/journal/14602075/authorguide#authorshipguidelines>.

14. We would also welcome the submission of cover suggestions or motifs to be used by our Graphics Illustrator in designing a cover.

15. Please use the link below to submit your revision:
<https://emboj.msubmit.net/cgi-bin/main.plex>

Referee #1:

In this manuscript the authors examined the function of a novel, brain-specific phosphorylation site in paxillin, at the Serin 119 position. The authors show that this mediates nuclear translocation in neurons and enrichment in the splicing organelles in the nucleus, the speckles. The authors show that this phospho-site influences interaction with importin-b2 (e.g. decreased when this site is mutated to Alanin) and identify 3 non-canonical NLS-sites that much reduce nuclear localization when mutated. They then proceed towards understanding the kinases regulating this phosphorylation in maturing neurons and identify CDK5 in this function.

One of the most interesting aspects of this work is the transient window during which Paxillin translocates to the nucleus, in vitro and in vivo. In vitro this occurs between DIV 5 and 14 and is influenced by neuronal activity, and in vivo this occurs between P10 and P30 during the critical phase of increased plasticity in sensory areas. However, nuclear Paxillin is reduced in the somatosensory cortex in the row where whisker input is reduced by trimming.

The authors then proceed to explore the function of Paxillin in the speckles regarding splicing, identifying the interactors of Paxillin in the nuclear fraction going back to the in vitro models. They identify several splicing factors as interactors and, using CLIP, several target RNAs, including the synaptic protein SNAP25. The importance of PhosphoS119Paxillin regulating the splicing of SNAP25 is then beautifully demonstrated in vivo for the switch between the SNAP5a versus 5b isoform by inducible expression of the 119Alanin form interfering with this splicing switch in neurons.

Finally, the authors demonstrate in vivo, targeting postnatal neurogenesis in the hippocampus using the NestinCreERT2 mouse line, that the S119A mutation in Paxillin causes reduced synaptic efficiency in Schaffer collaterals, LTP reduction as well as behavioral phenotypes when functionally active TRAP neurons are targeted by this mutation. This very comprehensive work ranging from molecular to in vivo behavior analysis thus identifies a crucial role of Paxillin in splicing by transient nuclear translocation mediated by phosphorylation at S119.

Suggestions:

1) Throughout the Figures, please include the individual data points in the histograms, e.g. Figure 1F, 1G, 2B, D, H, J, 4H, 5G etc.

For analysis in cultures please use a culture batch as 1 biological replicate and perform statistics with n=several biological replicates.

2) An important open question is to which extent this function of Paxillin is a moonlighting function, or if nuclear Paxillin still interacts with actin and the nuclear cytoskeleton. Could the authors use binding deficient mutants to assess this?

3) If possible, it would be fantastic to know about the splicing changes in the NestinCreERT2 induced S119A mutation in a comprehensive manner to identify the critical players in this functionally relevant context.

Referee #2:

In this study Chu et al. report an unexpected role of paxillin in the regulation of alternative splicing during brain development. They show that paxillin phosphorylated at serine119 phosphorylation accumulates in the nucleus in activity-dependent manner and modulates alternative splicing during this period. The authors show that phosphorylated paxillin is present at nuclear speckles, and that it interacts with U2AFs and splicing factors. They identify distinct splicing events regulated by nuclear paxillin and show functional implications for synaptic plasticity. These findings are novel and of interest for the field. I have several comments that should be addressed.

Major:

- The data suggest that a broad range of different types neuronal activation results in paxillin phosphorylation and a shift in subcellular distribution. The outline of the study insinuates major relevance for a critical period during early synaptogenesis. This is not entirely covered by the data. The authors should address whether nucleocytoplasmic shuttling is indeed limited to a critical phase in development. This can be done in organotypic slices or primary neurons (>DIV 28) with chronic stimulation preferentially of NMDAR.

- Along these lines RNA sequencing (60+ million reads/sample) and rMATs analysis could be performed in mature neurons (+/- NMDA stimulation). If sustained activation of NMDAR is sufficient to induce similar alterations in alternative splicing this would change the interpretation of the results.

- Phosphorylation of paxillin at S119 increases its association with importin beta 2, suggesting phosphorylation as the switch to its nuclear import. Alternatively, also the paxillin phosphorylating kinases (such as PKA or CDK5) may be imported into the nucleus, leading to a nuclear increase of phosphorylated paxillin. Do pan-paxillin show an increase in the nucleus at DIV7, compared to the other timepoints in Fig 1 A? Can you also show an increase of pan-paxillin following NMDAR activation at DIV14?

- In Fig 3 & Sup 6 pan paxillin staining should be added to show that changes in phosphorylated paxillin are not due to changes in overall paxillin levels.

- Figure 6: p. 7 Mice with inducible deficiency in paxillin S119 phosphorylation (Nestin-CreERT2; paxillinS119A/fl or Nestin-CreERT2; paxillinS119A/+) and those with reduced paxillin expression (Nestin-CreERT2; paxillinfl/+) were administered tamoxifen at P5, P7, and P9, timed to precede anticipated nuclear translocation of the phosphorylated protein. Please check for nuclear accumulation of paxillin in control slices.

- In the text (p.7, 8) significant differences in fig. 6 B & C are mentioned, which were not indicated in the graphs. Which statistical test was used in fig. 6B?

- The representation of data in fig 7 is a bit confusing and misleading, as it is not clear which control belongs to which experiment. This becomes especially apparent in panel D, as group 'TRAP2; PXNS1229A/+' seems to show a clear increase when compared to the mean of the control group. I assume only control animals that were run in parallel with 'TRAP2; PXNS1229A/+' mice were considered in the analysis, thus the representation of data should also reflect this. Please split the graphs in a way it becomes clear which controls belong to which of the other 3 groups

- Regarding experiments in fig. 6-7, one cannot state that the electrophysiological and behavioral differences in Pax KD/119A conditions are necessarily developmental or acute. If tamoxifen was administered at a later time point (e.g. 1 week before testing) would the effects still be apparent?

Minor:

p.2: 'Paxillin (KD) in young neurons' - without brackets

p.3: 'Conversely, pretreatment of DIV3 or DIV5 cortical neurons' - typo. Rephrase the sentence. It is difficult to understand.

p.6: 'These findings support the idea that that paxillin modulates timing of AS programs that govern neuronal activity and synaptic strength.' - typo

Referee #3:

- general summary and opinion about the principal significance of the study, its questions and findings

Here the authors have shown the critical role of paxillin phosphorylation at serine (p-PaxillinS119) in regulating alternative splicing (AS) in neurons during the sensitive postnatal period. During this phase, neuronal circuits undergo refinement which coincides with widespread AS events which is essential for synaptic plasticity.

The authors showed that how neuronal activity-dependent phosphorylation of paxillin at S119 acts as a molecular switch in the nucleus, enabling the timely AS of genes that are crucial for synaptic function and plasticity. Upon NMDA receptor activation, p-

PaxillinS119 translocate to nuclear speckles, where it interacts with core components of the splicing machinery, including U2AF proteins and other RNA-binding proteins (RBPs). This interaction facilitates the AS of key synaptic genes such as Snap25, Anks, and the GABAB receptor, which are essential for neurotransmission and synaptic plasticity. Mice lacking p-PaxillinS119 exhibit delays in the AS transition from the Snap25-5a to the Snap25-5b isoform, which leads to impaired presynaptic neurotransmitter release at hippocampal Schaffer collateral-CA1 synapses. Consequently, these mice show deficits in short-term learning and memory, indicating the physiological importance of this regulatory mechanism.

Mechanistically, they have also showed that p-PaxillinS119 functions downstream of BDNF/cAMP/Cdk5 signaling to regulate AS in response to synaptic activity. Its nuclear localization is dependent on importin β 2 in immature neurons, but in mature neurons, NMDA receptor activation is required for its nuclear accumulation. This regulation ensures timely AS switching of genes crucial for synaptic plasticity.

This study highlights p-PaxillinS119 as a critical mediator linking synaptic activity to AS programs, enabling proper synaptic maturation during postnatal development.

- specific major concerns essential to be addressed to support the conclusions

Point 1. In the Figure 5B authors has showed upon KD of paxillin their more GO term enrichment related to synapse function, but there is no direct evidence of those changes in respect to experimental validation. It would be nice if the authors can perform the synaptosome fraction isolation and shows those splice form variant.

Point 2. All the behaviors studies' N number of mice were not sufficient, usually requires $N \geq 8$. Please add more biological replicates.

Point 3. All IF staining images are suggested to change into colorful. If authors have concern about color-blind readers, it is recommended that colorful versions of immunofluorescence images be placed in the main figure, and the black-and-white versions in the supplementary figures.

Point 4. All Bar plots should be changed into bar-dot plots to better display every sample.

- minor concerns that should be addressed

Point 5. Authors simply used "t-test" and "ANOVA" in Figure legends and Method details. Please add information of which types of t-tests (e.g multiple t-tests) and ANOVA (e.g. one-way ANOVA) were used to analysis.

Point 6. Some texts and symbols were half-hidden, such as Figure 2E's "+", and Figure 5E's "exon5b-M1 M2".

Point 7. In Figure 5D, "Sanp" should be corrected into "Snap".

Point 8. In Figure 7G, there were error in vertical axis numbers. For example, "0.2" should change into "20".

Point 9. Authors should list every antibodies information of IF, WB and IP in Method details-Plasmids, antibodies and materials section.

Point 10. Authors should adjust aspect ratio of the text "Data availability".

Point 11. In Method details-Mouse behavior assessments section, it's better to add description of standards of animal's selection for behaviors and gender information.

Referee's comments are in italics

Referee #1:

In this manuscript the authors examined the function of a novel, brain-specific phosphorylation site in Paxillin, at the Serin 119 position. The authors show that this mediates nuclear translocation in neurons and enrichment in the splicing organelles in the nucleus, the speckles. The authors show that this phospho-site influences interaction with importin-b2 (e.g. decreased when this site is mutated to Alanin) and identify 3 non-canonical NLS-sites that much reduce nuclear localization when mutated. They then proceed towards understanding the kinases regulating this phosphorylation in maturing neurons and identify CDK5 in this function.

One of the most interesting aspects of this work is the transient window during which Paxillin translocates to the nucleus, in vitro and in vivo. In vitro this occurs between DIV 5 and 14 and is influenced by neuronal activity, and in vivo this occurs between P10 and P30 during the critical phase of increased plasticity in sensory areas. However, nuclear Paxillin is reduced in the somatosensory cortex in the row where whisker input is reduced by trimming.

The authors then proceed to explore the function of Paxillin in the speckles regarding splicing, identifying the interactors of Paxillin in the nuclear fraction going back to the in vitro models. They identify several splicing factors as interactors and, using CLIP, several target RNAs, including the synaptic protein SNAP25. The importance of PhosphoS119Paxillin regulating the splicing of SNAP25 is then beautifully demonstrated in vivo for the switch between the SNAP5a versus 5b isoform by inducible expression of the 119Alanin form interfering with this splicing switch in neurons.

Finally, the authors demonstrate in vivo, targeting postnatal neurogenesis in the hippocampus using the NestinCreERT2 mouse line, that the S119A mutation in Paxillin causes reduced synaptic efficiency in Schaffer collaterals, LTP reduction as well as behavioral phenotypes when functionally active TRAP neurons are targeted by this mutation. This very comprehensive work ranging from molecular to in vivo behavior analysis thus identifies a crucial role of Paxillin in splicing by transient nuclear translocation mediated by phosphorylation at S119.

Suggestions:

1) Throughout the Figures, please include the individual data points in the histograms, e.g. Figure 1F, 1G, 2B, D, H, J, 4H, 5G etc.

For analysis in cultures please use a culture batch as 1 biological replicate and perform statistics with n=several biological replicates.

We have revised all relevant figures to display individual data points within histograms, as requested (e.g., **Figures 1F, 1G, 2B, 2D, 2H, 2J, 4H, 5G**, and others). Additionally, we ensured that each culture batch is treated as a single biological replicate, and all statistical analyses were re-performed using multiple independent biological replicates.

2) An important open question is to which extent this function of Paxillin is a moonlighting function, or if

nuclear Paxillin still interacts with actin and the nuclear cytoskeleton. Could the authors use binding deficient mutants to assess this?

We appreciate the reviewer's comment. To examine whether nuclear Paxillin's role in splicing regulation depends on interactions with the actin cytoskeleton, we conducted additional experiments in both Neuro2a cells expressing Paxillin variants and in primary neurons at DIV7 (see **new Figure EV4; Results section Page 6, Lines 15–21**). In both cases, we treated cells with Latrunculin A, a potent actin polymerization inhibitor, and observed marked disruption of the phalloidin-stained peripheral actin cytoskeleton. However, neither splicing activity (as assessed using the pGint splicing reporter assay) nor Paxillin's co-localization in nuclear speckles was significantly affected by Latrunculin A treatment. In contrast, expression of the S119 phosphorylation-deficient Paxillin mutant (Paxillin S119A) consistently decreased pGint splicing activity, regardless of actin disruption. These results indicate that Paxillin's ability to co-condense with splicing factors and nuclear actin (as detected by C4 and 1C7 antibodies, data not shown) within speckles and to modulate alternative splicing is largely independent of cytoplasmic actin dynamics under these conditions.

3) If possible, it would be fantastic to know about the splicing changes in the NestinCreERT2 induced S119A mutation in a comprehensive manner to identify the critical players in this functionally relevant context.

We appreciate the reviewer's enthusiasm for a more comprehensive view of the splicing changes associated with the Nestin-CreERT2-induced Paxillin S119A mutation. To address this, we performed additional Nanopore long-read cDNA sequencing of hippocampal tissue collected from Nestin-CreERT2; Paxillin^{fl/S119A} and control mice at P10, P15, and P21 (**new Appendix Figure S6; Results section Page 7, Lines 39–42**). This analysis revealed that Paxillin^{S119A}-associated isoform switching occurs at several key synaptic genes, including SNAP25 and NMDAR1. Consistent with our original short-read RNA-seq data from Paxillin knockdown neuronal cultures, generation of alternatively spliced isoforms of SNAP25 and NMDAR decreased in the Nestin-CreERT2; Paxillin^{fl/S119A} hippocampus. Furthermore, RNA-binding protein (RBP) motif enrichment analysis of affected splice variants and cassette exons again showed strong enrichment for RBPs previously implicated in analysis of Paxillin KD cells, including Rbfox2, FUS, Nova2, and U2AF2. These results provide additional evidence that Paxillin S119 phosphorylation modulates synaptic transcript isoforms *in vivo* during a critical postnatal period. Given the scope of this study, we include these key findings in **new Appendix Figure S6C**; however, we will address a full genome-wide analysis in future work.

Referee #2:

In this study Chu et al. report an unexpected role of paxillin in the regulation of alternative splicing during brain development. They show that paxillin phosphorylated at serine119 phosphorylation accumulates in the nucleus in activity-dependent manner and modulates alternative splicing during this period. The authors show that phosphorylated paxillin is present at nuclear speckles, and that it interacts with U2AFs and splicing factors. They identify distinct splicing events regulated by nuclear paxillin and show functional implications for synaptic plasticity. These findings are novel and of interest for the field. I have several comments that should be addressed.

Major:

- The data suggest that a broad range of different types neuronal activation results in paxillin phosphorylation and a shift in subcellular distribution. The outline of the study insinuates major relevance for a critical period during early synaptogenesis. This is not entirely covered by the data. The authors should address whether nucleocytoplasmic shuttling is indeed limited to a critical phase in development. This can be done in organotypic slices or primary neurons (>DIV 28) with chronic stimulation preferentially of NMDAR.

We thank the reviewer for this suggestion. To directly test whether nucleocytoplasmic shuttling of p-Paxillin^{S119} induced by neuronal activity is restricted to a defined developmental window, we performed additional experiments to assess responsiveness of primary neurons at later stages—namely, DIV14, DIV21, and DIV28—by examining p-Paxillin^{S119} nuclear translocation following NMDA receptor activation.

Due to significant excitotoxicity resulting from sustained NMDA exposure at later stages, we adjusted treatment conditions to 20 μ M NMDA for either acute (25-minute) or prolonged (6-hour) stimulation. As shown in the **new Figure EV3A and EV3C**, DIV14 neurons displayed robust nuclear accumulation of p-Paxillin^{S119} in both conditions, as quantified by the nuclear-to-cytoplasmic (N/C) ratio and nuclear signal intensity. In contrast, this responsiveness declined at DIV21 and was nearly absent by DIV28. These results support our conclusion that NMDA-induced nucleocytoplasmic shuttling of p-Paxillin^{S119} is developmentally regulated and largely confined to the sensitive period. This new evidence is now incorporated into the revised Results section (**Page 4, Lines 42-43 and Page 5, Lines 1-2**).

- Along these lines RNA sequencing (60+ million reads/sample) and rMATs analysis could be performed in mature neurons (+/- NMDA stimulation). If sustained activation of NMDAR is sufficient to induce similar alterations in alternative splicing this would change the interpretation of the results.

We thank the reviewer for this suggestion. While our original rMATs analysis focused on developing neurons (E15 to DIV15) and highlighted Snap25 as a representative gene (**Figure 5**), we conducted additional targeted isoform-specific RT-qPCR to assess NMDA-induced Snap25 splicing in more mature cultures (DIV14–DIV28) (see **new Figure EV3E; Results section Page 7, Lines 24–33**). We observed that although baseline Snap25-5b transcript levels gradually increased from DIV14 to DIV28 under control conditions, NMDA treatment induced a clear increase in Snap25-5b transcript levels and increased the 5b to 5a inclusion ratio at DIV14 but not at DIV21 or DIV28. Moreover, prolonged NMDA exposure (6

hours) reduced total Snap25 levels across all stages, likely due to transcript degradation, with no further induction of the 5b isoform. These results support our original conclusion that NMDA-induced alternative splicing of Snap25-5b is more robust during an early, activity-sensitive developmental window.

- Phosphorylation of paxillin at S119 increases its association with importin beta 2, suggesting phosphorylation as the switch to its nuclear import. Alternatively, also the paxillin phosphorylating kinases (such as PKA or CDK5) may be imported into the nucleus, leading to a nuclear increase of phosphorylated paxillin. Do pan-paxillin show an increase in the nucleus at DIV7, compared to the other timepoints in Fig 1 A? Can you also show an increase of pan-paxillin following NMDAR activation at DIV14?

To address the possibility that nuclear translocation of kinases such as CDK5 could underlie the observed nuclear accumulation of p-Paxillin^{S119}, we performed new experiments to examine CDK5 localization across developmental stages (DIV3–DIV28) and after NMDA receptor activation at DIV14 (**new Figure EV3; Result section Page 5, Lines 2-6**). Consistent with our previous findings, nuclear localization of p-Paxillin^{S119} peaked around DIV7 and was strongly inducible by NMDA treatment at DIV14. However, CDK5 levels did not show a matching nuclear-to-cytoplasmic (N/C) ratio pattern: although CDK5 expression increased with maturation, it remained largely cytoplasmic and did not parallel robust nuclear enrichment of phospho-Paxillin^{S119}. Together with our earlier data showing that S119 phosphorylation is required for Paxillin's importin β 2 interaction and that blocking importin function (with IPZ) retains p-Paxillin^{S119} in the cytoplasm, these results indicate that p-Paxillin^{S119} nuclear accumulation is driven primarily by its own phosphorylation and importin-mediated import rather than by nuclear translocation of CDK5.

Regarding pan-Paxillin, we observed a modest increase in its nuclear levels at DIV7 (see **Figures EV2A and EV2B**), but NMDA treatment at DIV14 did not significantly elevate nuclear pan-Paxillin (**Figure EV3A and EV3C**). This observation may reflect differential post-translational modification of other forms of Paxillin (e.g., phospho-Paxillin^{S83}) or limitations in detecting specific nuclear pools within condensates. While CDK5 nuclear import appears unlikely to explain these findings, we acknowledge that other nuclear kinases may still contribute to S119 phosphorylation under NMDA stimulation.

- In Fig 3 & Sup 6 pan paxillin staining should be added to show that changes in phosphorylated paxillin are not due to changes in overall paxillin levels.

In response, we have added pan-Paxillin staining and quantification to **new Appendix Figure S5 and revised Appendix Figure S4** (original Fig S6). The expression pattern and levels of p-Paxillin^{S119} remain consistent when normalized to total Paxillin, confirming that the observed changes in phosphorylation are not simply due to changes in overall Paxillin abundance.

- Figure 6: p. 7 Mice with inducible deficiency in paxillin S119 phosphorylation (Nestin-CreERT2;

paxillin^{S119A/fl} or Nestin-CreERT2; paxillin^{S119A/+}) and those with reduced paxillin expression (Nestin-CreERT2; paxillin^{fl/+}) were administered tamoxifen at P5, P7, and P9, timed to precede anticipated nuclear translocation of the phosphorylated protein. Please check for nuclear accumulation of paxillin in control slices.

We now include expression profiles indicating abundance of nuclear Paxillin and p-Paxillin^{S119} in relevant control brain sections. These data, presented in the **new Appendix Figure S8**, confirm expected nuclear localization of Paxillin in control animals, a pattern that shows an increase starting at P12 and that peaks around P21, with or without tamoxifen administration at P5, P7, and P9.

- In the text (p.7, 8) significant differences in fig. 6 B & C are mentioned, which were not indicated in the graphs. Which statistical test was used in fig. 6B?

We have now added statistical significance indicators to graphs shown in **Figures 6B and 6C**. For Figure 6B, a linear regression analysis in GraphPad Prism was used to compare the slopes between groups and assess statistical differences in fit. This information is now stated in the figure legend.

- The representation of data in fig 7 is a bit confusing and misleading, as it is not clear which control belongs to which experiment. This becomes especially apparent in panel D, as group 'TRAP2; PXNS1229A/+' seems to show a clear increase when compared to the mean of the control group. I assume only control animals that were run in parallel with 'TRAP2; PXNS1229A/+' mice were considered in the analysis, thus the representation of data should also reflect this. Please split the graphs in a way it becomes clear which controls belong to which of the other 3 groups

We appreciate this comment relevant to clarity of control group assignments. Our reason for using a combined control group stems from the genetic constraints of our experimental design. Because both homozygous Paxillin knockout and homozygous Paxillin S119A mutation are embryonically lethal, the Paxillin^{S119A} conditional knock-in line must be maintained as heterozygous. To ensure balanced genotype distribution and proper littermate controls, we consistently cross TRAP2; PXN^{S119A/+} mice with TRAP2; PXN^{fl/fl} or TRAP2; PXN^{fl/+} mice for each experiment, a breeding scheme that generates all possible experimental genotypes and corresponding controls within the same litters. Data were collected until each genotype group included at least five biological replicates.

We have now added a detailed explanation of this breeding strategy and control assignment to the Materials and Methods section (see **Page 11, Lines 21-26**) to state why the three experimental groups share the same set of littermate controls and why separate control groups cannot be displayed in split graphs. We hope this revision resolves the concern regarding data representation.

- Regarding experiments in fig. 6-7, one cannot state that the electrophysiological and behavioral differences in Pax KD/119A conditions are necessarily developmental or acute. If tamoxifen was administered at a later time point (e.g. 1 week before testing) would the effects still be apparent?

To address whether the observed behavioral deficits are necessarily developmental, we performed new experiments in TRAP2 adult mice, administering tamoxifen three times (once every other day) starting one week prior to behavioral testing to assess immediate functional consequences, as suggested. As shown in **new Appendix Figure S9**, mice with this acute manipulation exhibited comparable performance in the open field test, Y-maze test, and novel object recognition test. These tests assess the same three working memory behaviors that were clearly impaired in mice subjected to developmentally-timed TRAP2-driven Pax KD/119A conditions (**Figure 7**). These results suggest that the observed behavioral impairments are more likely linked to developmental disruption during the critical period rather than to acute loss of Paxillin function in adulthood.

However, we also note that acute tamoxifen induction is not feasible with the Nestin-CreERT2 mouse line, as the nestin promoter is neither neuron-specific nor sufficiently active at later time points, precluding us from using the same temporal restriction for the electrophysiological experiments in Nestin-CreERT2; PXN KD/S119A mice. We acknowledge this limitation and understand that future studies using alternative temporally restricted genetic tools will be needed to fully separate developmental from acute roles of Paxillin in regulating synaptic strength and plasticity.

Minor:

p.2: 'Paxillin (KD) in young neurons' - without brackets

p.3: 'Conversely, pretreatment of DIV3 or DIVR5 cortical neurons' - typo. Rephrase the sentence. It is difficult to understand.

We corrected “Paxillin (KD)” to “Paxillin KD” and changed “DIVR5” to “DIV5”. We also rephrased the sentence to improve clarity (**Page 3, Lines 33-35**). The revised version now reads:

"Conversely, pretreating DIV3 or DIV5 cortical neurons with LB100, an inhibitor of serine/threonine protein phosphatase 2A (PP2A), significantly increased both cytoplasmic and nuclear levels of p-Paxillin^{S119} when assessed at DIV7 (**Appendix Figure S2A**), likely due to sustained S119 phosphorylation."

p.6: 'These findings support the idea that that paxillin modulates timing of AS programs that govern neuronal activity and synaptic strength.' – typo

We thank the reviewer for noting these typos. We corrected the duplicated word “that” in the sentence.

Referee #3:

- general summary and opinion about the principal significance of the study, its questions and findings
Here the authors have shown the critical role of paxillin phosphorylation at serine (p-PaxillinS119) in regulating alternative splicing (AS) in neurons during the sensitive postnatal period. During this phase, neuronal circuits undergo refinement which coincides with widespread AS events which is essential for synaptic plasticity.

The authors showed that how neuronal activity-dependent phosphorylation of paxillin at S119 acts as a molecular switch in the nucleus, enabling the timely AS of genes that are crucial for synaptic function and plasticity. Upon NMDA receptor activation, p-PaxillinS119 translocate to nuclear speckles, where it interacts with core components of the splicing machinery, including U2AF proteins and other RNA-binding proteins (RBPs). This interaction facilitates the AS of key synaptic genes such as Snap25, Anks, and the GABA_B receptor, which are essential for neurotransmission and synaptic plasticity. Mice lacking p-PaxillinS119 exhibit delays in the AS transition from the Snap25-5a to the Snap25-5b isoform, which leads to impaired presynaptic neurotransmitter release at hippocampal Schaffer collateral-CA1 synapses. Consequently, these mice show deficits in short-term learning and memory, indicating the physiological importance of this regulatory mechanism.

Mechanistically, they have also showed that p-PaxillinS119 functions downstream of BDNF/cAMP/Cdk5 signaling to regulate AS in response to synaptic activity. Its nuclear localization is dependent on importin β 2 in immature neurons, but in mature neurons, NMDA receptor activation is required for its nuclear accumulation. This regulation ensures timely AS switching of genes crucial for synaptic plasticity.

This study highlights p-PaxillinS119 as a critical mediator linking synaptic activity to AS programs, enabling proper synaptic maturation during postnatal development.

- specific major concerns essential to be addressed to support the conclusions

Point 1. In the Figure 5B authors has showed upon KD of paxillin their more GO term enrichment related to synapse function, but there is no direct evidence of those changes in respect to experimental validation. It would be nice if the authors can perform the synaptosome fraction isolation and shows those splice form variant.

We appreciate the reviewer's comment. Due to lack of commercially available isoform-specific antibodies capable of distinguishing proteins translated from individual RNA splice variants of genes such as SNAP25 and NMDAR1 (NR1), we performed new experiments using an alternative approach. Specifically, we used antibodies that recognize distinct NR1 protein domains (namely, N1, C1, and C2/C2'), which are encoded by three alternatively spliced cassette exons of the NR1 gene. We then examined the relative abundance of these domain-specific NR1 protein variants in synaptosome fractions isolated from hippocampal tissue of Nestin-CreERT2; Paxillin^{S119A/fl} mice and littermate controls. As shown in our new Western blot analysis (see **new Appendix Figure S6A and S6B; Results section Page 6, Lines 39-42**), synaptosomes from Nestin-CreERT2; Paxillin^{S119A/fl} mice exhibited lower levels of NR1 variants containing the N1 and C1 domains, while levels of PSD95 and NR1 variants containing the C2' domain were comparable to controls. These results are consistent with our new Nanopore cDNA (long-

read) sequencing data (**new Appendix Figure S6C**), which similarly shows a reduced percent spliced-in (PSI) value for the N1 and C1 cassette exons, exon 5 and exon 22, respectively, in the Nestin-CreERT2; Paxillin^{S119A/fl} hippocampus, in the same mice that exhibited altered synaptic strength in our electrophysiology analysis (please see **Figure 6**). Together, these additional data provide experimental support for the functional relevance of the observed splicing changes at the level of synaptic proteins.

Point 2. All the behaviors studies' N number of mice were not sufficient, usually requires N{greater than or equal to}8. Please add more biological replicates.

In response, we increased the number of biological replicates for each behavioral assay, and all groups now include at least eight mice (**revised Figure 7**). Accordingly, we updated the figure legends and Materials and Methods section to reflect these revisions.

Point 3. All IF staining images are suggested to change into colorful. If authors have concern about color-blind readers, it is recommended that colorful versions of immunofluorescence images be placed in the main figure, and the black-and-white versions in the supplementary figures.

We thank the reviewer for this suggestion. We have now applied a color look-up table (Royal LUT in ImageJ) to the p-Paxillin^{S119} immunofluorescence staining channels for **Figures 1A, 1E, 1G, 2A, 2C, and 2I** to enhance visualization in the main figures.

Point 4. All Bar plots should be changed into bar-dot plots to better display every sample.

We updated all bar plots to bar-dot plots to display individual data points for each sample.

- minor concerns that should be addressed

Point 5. Authors simply used "t-test" and "ANOVA" in Figure legends and Method details. Please add information of which types of t-tests (e.g. multiple t-tests) and ANOVA (e.g. one-way ANOVA) were used to analysis.

We now specify the exact type of t-tests and/or ANOVA used in each analysis and have revised the figure legends and Method Details accordingly.

Point 6. Some texts and symbols were half-hidden, such as Figure 2E's "+", and Figure 5E's "exon5b-M1 M2".

These formatting issues have been corrected. Thank you.

Point 7. In Figure 5D, "Sanp" should be corrected into "Snap".

The typo has been corrected.

Point 8. In Figure 7G, there were error in vertical axis numbers. For example, "0.2" should change into "20".

We have corrected this axis labeling error.

Point 9. Authors should list every antibodies information of IF, WB and IP in Method details-Plasmids, antibodies and materials section.

We have now added a complete list of all antibodies used, including catalog numbers and sources, to Reagents_Tools_Table of the structured methods section, as suggested.

Point 10. Authors should adjust aspect ratio of the text "Data availability".

This formatting issue has been corrected.

Point 11. In Method details-Mouse behavior assessments section, it's better to add description of standards of animal's selection for behaviors and gender information.

We have now added detailed descriptions of animal selection criteria and specified the gender distribution in behavioral testing described in the Method Details section (Page 15, Lines 30-35).

Dear Pei-Lin,

Thank you again for submitting your revised manuscript (EMBOJ-2025-120719R) to The EMBO Journal for our consideration, and for your patience during peer review. Your manuscript has been sent back to the three original referees that had previously reviewed the initial version of your manuscript, and we have now received their comments, which you can find below.

I am very pleased to say that all three referees are satisfied with the revision, mentioning that almost all comments raised in the previous round of review have been successfully addressed and that the manuscript has been significantly strengthened by the addition of new data and analyses. There are only a few remaining issues that must be addressed in a final version of the manuscript before we can accept it for publication in The EMBO Journal:

1. Please show the individual data points as suggested by referee #1 and clearly show independent biological replicates in all listed Figure panels (referee #1); the statistics should indeed be calculated on the independent biological replicates, not technical replicates.
2. Please clearly mention and discuss in your revised manuscript the limitation identified by referee #2.
3. We agree with referee #3 that the manuscript would benefit from the addition of high- or super-resolution imaging data, but we will not request it for publication of the study.
4. Please consider adding a discussion on the translational potential and a figure summarizing the developmental dynamics of p-Paxillin^{S119}, as suggested by referee #4.

Please submit a revised version of your manuscript with all textual changes highlighted (and revised Figures, if necessary) along with a detailed list of changes/corrections/explanations.

There are also a few other changes and corrections we need you to make in the final version of your manuscript:

- The order of the manuscript sections must be corrected as follows: Title page - Abstract and Keywords - Introduction - Results - Discussion - Methods - Data Availability - Acknowledgements - Disclosure and Competing Interests Statement - References - Figure Legends - (main Tables with legends, if applicable) - Expanded View Figure Legends.
- Please add a list of up to 5 relevant keywords (preferably broad terms to enhance online search discoverability of your article) after the Abstract of your revised manuscript.
- Thank you for providing the referees access to the deposited datasets. Please make sure that all datasets will be publicly available at the time of publication. The reviewer account details for confidential access can now be removed from the Data availability statement of the manuscript. For each dataset, please make sure to list in this section the database, accession number-ID, and permanent specific URL.
- Please change the heading of your conflict-of-interest statement to "Disclosure and competing interests statement".
- The author contributions statement should be removed from the manuscript file. Instead, we use CRediT to specify the contributions of each author in the journal submission system. Please feel free to use the free text box to provide more detailed descriptions during submission. See also our guide to authors for more information:
<https://www.embopress.org/page/journal/14602075/authorguide#authorshipguidelines>.
- As per our journal's policy, "data not shown" (on page 4) is not permitted. All data referred to in the paper should be displayed in the main or Expanded View (EV) figures, or in the Appendix. Please add these data or change the text accordingly if these data are not central to the study and its conclusions, or properly cite the respective published sources if these data can be found elsewhere.
- The following funding information needs to be removed from the Comments box and provided instead as a separate entry: "Academia Sinica Core Facility and Innovative Instrument Project (AS-CFII-111-209)".
- We noticed that callouts for the following Figure panels are missing: 3ABD, 5A, 6A.
- Please correct "Supplemental Figure 2A": does this callout refer to Figure EV2A or to Appendix Figure S2A?
- The Appendix file needs heading "Appendix for:" followed by the manuscript's title on its title page, and then a Table of Contents including page numbers for the listed items; Appendix pages should be numbered.
- Please also note that the title of Appendix Figure S9 needs to be corrected in the Appendix file (currently labeled as "Appendix 9").
- Please note that EMBO press papers are accompanied online by:

A) a short (2 sentences) summary of the findings and their significance,
B) 2-5 short bullet points highlighting the key results, and
C) a synopsis image in .jpg or .png format that is exactly 550 pixels wide and 300-600 pixels high (the height is variable). Please note that all text in the image needs to be legible at the final size.
Please upload this information along with your revised manuscript (the text for A and B should be provided in a separate Word file).

- Please move sections "Animals" and "Cell culture" of your Methods to "Methods and Protocols"; more information on our structured methods format can be found here:
<https://www.embopress.org/page/journal/14602075/authorguide#structuredmethods>.

- During our routine pre-acceptance Figure checks, our Data Integrity analyst detected:

1. A splice site in Figure 4G ELAVL4 / INPUT.

2. Possible reuse within Figure EV1: pan-Paxillin, both for postnatal day 7 and 14.

Please check these Figures carefully, correct them if necessary, and upload source data for both along with clarification/description of any changes.

- During our routine data checks, our data editors have raised the following concern:

1. Statistics should not be shown when n=2 in Figure 5D. Please note that when n=2, only the individual data points should be shown, not their mean or any other statistics.

2. The exact p values should be provided in the legends of Figures 1H, 2B, D, H, J; 4D, E, G, H; 6B, C, D, E; EV3 D, EV4 B.

Please also note that as part of the EMBO publications' Transparent Editorial Process, The EMBO Journal publishes online a Peer Review File along with each accepted manuscript. This File will be published in conjunction with your paper and will include the referee reports, your point-by-point response and all pertinent correspondence relating to the manuscript. You can opt out of this by letting the editorial office know (contact@embojournal.org). If you do opt out, the Peer Review File link will point to the following statement: "No Peer Review File is available with this article, as the authors have chosen not to make the review process public in this case."

We look forward to seeing a final version of your manuscript as soon as possible. Please let us know if you have any questions and use this link to submit your revision: <https://emboj.msubmit.net/cgi-bin/main.plex>.

Best regards,

Ioannis

Referee #1:

The authors addressed all my comments and especially the further actin depolymerization experiments demonstrating the moonlighting function of Paxillin, namely that its nuclear function is independent of cytoplasmic actin. Also the additional splicing analysis greatly improved the impact and relevance of this work.

Unfortunately the presentation of individual data points and independent biological replicates (my comment 1 of the previous review) still needs some additions:

Individual data points are still missing in: Figure 1B,C

Independent biological replicates are not indicated in: Figures 1F, G, 2B, 2D, 2J, 4H, EV1E, EV2B, EV3C,D, EV4B, E, F (rather in these histograms the authors present cells measured, but don't indicate which come from different independent replicates, e.g. distinct culture batches; they indicate in the legends that they did independent culture replicates, but need to colour the dots accordingly and in principle perform the statistics on the independent replicates and not all cells)

With these additions I recommend this very interesting work and comprehensive analysis for publication

Referee #2:

I have reviewed the previous version of this manuscript. The authors have addressed convincingly most of my comments. I have only one comment and request.

- Unfortunately, nuclear levels of pan-paxillin neither show a significant and clear increase in DIV3, -5, -7 and -14 neurons, nor following NMDA treatment in DIV14 and -21 neurons, that is comparable with what the authors observe for phos-paxillin(Ser119). While this may be attributed to technical issues, such as dilution of the effect by unphosphorylated paxillin or potentially masking of the binding site of pan-paxillin by the phos-paxillin antibody, it suggests that paxillin may also be phosphorylated by kinases inside the nucleus to a large extent. This point should be mentioned and discussed in the paper.

Referee #3:

General Summary and Opinion

This study presents compelling evidence for a critical, previously underappreciated role of Paxillin phosphorylation at serine 119 (p-Paxillin^{S119}) in regulating alternative splicing (AS) during sensitive postnatal windows of neural circuit refinement. During this key developmental period, neuronal activity induces large-scale splicing transitions that produce isoforms essential for synaptic maturation and plasticity.

The authors demonstrate that NMDA receptor activation increases nuclear accumulation of p-Paxillin^{S119} in developing neurons, where it localizes to nuclear speckles and associates with splicing regulators, including U2AF components and RNA-binding proteins such as RbFox2 and NOVA1/2. Importantly, this process is shown to be downstream of BDNF/cAMP/Cdk5 signaling and operates in an activity-dependent and developmentally restricted manner. Nuclear import of p-Paxillin^{S119} is mediated by importin β 2 in immature neurons, whereas in mature neurons, synaptic activity becomes the dominant regulator.

Loss of S119 phosphorylation impairs developmentally timed splicing switches in genes critical for synaptic function, including Snap25, GABAb receptor subunits, and Ankyrins. The failure to transition from embryonic to mature isoforms (e.g., Snap25-5a to Snap25-5b) leads to functional deficits in synaptic transmission and results in impaired short-term learning and memory in mouse models.

Of particular note is the specificity of this regulation: while other phosphorylation sites on Paxillin (e.g., S83, S178, S273) may influence nuclear localization in non-neuronal cells, only S119 phosphorylation responds to NMDA receptor activation in differentiated neurons and shows developmentally regulated nuclear localization aligned with critical periods. The authors propose that p-Paxillin^{S119} functions as a sensor and mediator of synaptic activity, linking external input to nuclear AS programs in a feedback loop that promotes circuit maturation.

Overall, the study provides significant insights into how synaptic activity is transduced into nuclear splicing decisions and identifies p-Paxillin^{S119} as a key molecular switch coordinating postnatal brain development.

Specific Major Concerns

None. All previous major concerns have been adequately addressed by the authors.

Minor Concerns

None. All previously raised minor points have also been appropriately resolved.

Additional Non-Essential Suggestions for Improvement

It may strengthen the impact of the study if the authors include high-resolution imaging or super-resolution microscopy of p-Paxillin^{S119} within nuclear speckles, to better define its spatial dynamics.

A broader discussion on how this mechanism may intersect with neurodevelopmental disorders involving disrupted synaptic maturation or splicing (e.g., ASD, schizophrenia) could enhance the translational relevance.

Consider including a schematic or timeline figure summarizing the developmental dynamics of p-Paxillin^{S119} activity across different brain regions in relation to critical period onset.

Editor's comments are in italics

Dear Pei-Lin,

Thank you again for submitting your revised manuscript (EMBOJ-2025-120719R) to The EMBO Journal for our consideration, and for your patience during peer review. Your manuscript has been sent back to the three original referees that had previously reviewed the initial version of your manuscript, and we have now received their comments, which you can find below. I am very pleased to say that all three referees are satisfied with the revision, mentioning that almost all comments raised in the previous round of review have been successfully addressed and that the manuscript has been significantly strengthened by the addition of new data and analyses. There are only a few remaining issues that must be addressed in a final version of the manuscript before we can accept it for publication in The EMBO Journal:

1. Please show the individual data points as suggested by referee #1 and clearly show independent biological replicates in all listed Figure panels (referee #1); the statistics should indeed be calculated on the independent biological replicates, not technical replicates.

We have revised the figures to display mean values from independent biological replicates, with identical symbol shapes indicating the same replicate across experimental groups (Figures 1F, 1G/H, 2B, 2D, 2J, 4H, EV1E, EV2B, EV3C–D, EV4B, EV4E–F; referee #1). All statistical analyses, whether using multiple t-tests or multiple comparison tests, were performed on the number of independent biological replicates, not on the number of cells measured. The individual data points (cells measured) corresponding to each replicate are provided in the source data for the same set of figures (referee #1).

2. Please clearly mention and discuss in your revised manuscript the limitation identified by referee #2.

We have added a paragraph to discuss this matter (please see **Discussion, Page 10, Lines 24-29**).

3. We agree with referee #3 that the manuscript would benefit from the addition of high- or super-resolution imaging data, but we will not request it for publication of the study.

Thank you. We have included a higher-resolution image of p-Paxillin^{S119} within nuclear speckles in the synopsis image.

4. Please consider adding a discussion on the translational potential and a figure summarizing the developmental dynamics of p-Paxillin^{S119}, as suggested by referee #4.

We have incorporated additional discussion on the translational potential (**Page 11, Lines 21–26**) and added a synopsis figure illustrating the developmental dynamics of p-Paxillin^{S119}.

Please submit a revised version of your manuscript with all textual changes highlighted (and revised Figures, if necessary) along with a detailed list of changes/corrections/explanations.

There are also a few other changes and corrections we need you to make in the final version of your manuscript:

- The order of the manuscript sections must be corrected as follows: Title page - Abstract and Keywords - Introduction - Results - Discussion - Methods - Data Availability - Acknowledgements - Disclosure and Competing Interests Statement - References - Figure Legends - (main Tables with legends, if applicable) - Expanded View Figure Legends.

We have revised the manuscript to follow the requested section order exactly as instructed.

- Please add a list of up to 5 relevant keywords (preferably broad terms to enhance online search discoverability of your article) after the Abstract of your revised manuscript.

We have added a list of relevant keywords after the Abstract in the revised manuscript, as requested.

- Thank you for providing the referees access to the deposited datasets. Please make sure that all datasets will be publicly available at the time of publication. The reviewer account details for confidential access can now be removed from the Data availability statement of the manuscript. For each dataset, please make sure to list in this section the database, accession number-ID, and permanent specific URL.

We have made all the deposited datasets publicly available and updated the Data Availability statement accordingly.

- Please change the heading of your conflict-of-interest statement to “Disclosure and competing interests statement”.

We have changed the heading to “Disclosure and competing interests statement,” as instructed.

- The author contributions statement should be removed from the manuscript file. Instead, we use CRediT to specify the contributions of each author in the journal submission system. Please feel free to use the free text box to provide more detailed descriptions during submission. See also our guide to authors for more information: <https://www.embopress.org/page/journal/14602075/authorguide#authorshipguidelines>.

We have removed the author contributions statement, as instructed

- As per our journal's policy, "data not shown" (on page 4) is not permitted. All data referred to in the paper should be displayed in the main or Expanded View (EV) figures, or in the Appendix. Please add these data or change the text accordingly if these data are not central to the study and its conclusions, or properly cite the respective published sources if these data can be found elsewhere.

As the data are not central to the study, we have removed the "data not shown" statement.

- The following funding information needs to be removed from the Comments box and provided instead as a separate entry: "Academia Sinica Core Facility and Innovative Instrument Project (AS-CFII-111-209)".

We have removed "Academia Sinica Core Facility and Innovative Instrument Project (AS-CFII-111-209)" from the Comments, as instructed.

- We noticed that callouts for the following Figure panels are missing: 3ABD, 5A, 6A.

Thank you. We have ensured that Figure panels 3A–D, 5A, and 6A are now cited in the text.

- Please correct "Supplemental Figure 2A": does this callout refer to Figure EV2A or to Appendix Figure S2A?

We have corrected this callout to "Appendix Figure S1A".

- The Appendix file needs heading "Appendix for:" followed by the manuscript's title on its title page, and then a Table of Contents including page numbers for the listed items; Appendix pages should be numbered.

We have revised the Appendix format as instructed, adding the heading "Appendix for:" followed by the manuscript title, a Table of Contents with page numbers, and page numbering throughout.

- Please also note that the title of Appendix Figure S9 needs to be corrected in the Appendix file (currently labeled as "Appendix 9").

Thank you. We have corrected the title of Appendix Figure S9 in the Appendix file.

*- Please note that EMBO press papers are accompanied online by:
A) a short (2 sentences) summary of the findings and their significance,
B) 2-5 short bullet points highlighting the key results, and
C) a synopsis image in .jpg or .png format that is exactly 550 pixels wide and 300-600 pixels high (the height is variable). Please note that all text in the image needs to be legible at the final size.*

Please upload this information along with your revised manuscript (the text for A and B should be provided in a separate Word file).

We have provided the two-sentence summary, bullet points highlighting the key results, and the synopsis image in the required format, as instructed.

- Please move sections “Animals” and “Cell culture” of your Methods to “Methods and Protocols”; more information on our structured methods format can be found here: <https://www.embopress.org/page/journal/14602075/authorguide#structuredmethods>.

We have revised the format as instructed, moving the “Animals” and “Cell culture” sections to “Methods and Protocols.”

During our routine pre-acceptance Figure checks, our Data Integrity analyst detected:

- 1. A splice site in Figure 4G ELAVL4 / INPUT.*
- 2. Possible reuse within Figure EV1: pan-Paxillin, both for postnatal day 7 and 14. Please check these Figures carefully, correct them if necessary, and upload source data for both along with clarification/description of any changes.*

We have provided the source data for Figure 4G (ELAVL4 / INPUT) and Figure EV1 (pan-Paxillin), with the relevant positions highlighted on the original blots. In Figure 4G, the splice site arose from assembling lanes from the same gel and exposure in a non-contiguous order to match the experimental design; a dashed line has been added to indicate this. In Figure EV1, the similarity among panels reflects the use of the same blot, which was stripped of other antibodies (e.g., anti-p-Paxillin^{S119} or anti-p-Paxillin^{Y118}) and sequentially reprobbed for pan-Paxillin as an internal control in the same samples. Full, unprocessed source blots of pan-Paxillin are provided to verify data integrity.

- During our routine data checks, our data editors have raised the following concern:

- 1. Statistics should not be shown when n=2 in Figure 5D. Please note that when n=2, only the individual data points should be shown, not their mean or any other statistics.*

As suggested, we have removed the statistical comparisons and now display only the individual data points (n = 2), without the mean, in Figure 5D.

- 2. The exact p values should be provided in the legends of Figures 1H, 2B, D, H, J; 4D, E, G, H; 6B, C, D, E; EV3 D, EV4 B.*

The exact p values are now shown directly on the revised Figures 1H, 2B, 2D, 2H, 2J; 4D, 4E, 4G, 4H; 6B, 6C, 6D, 6E; EV3D, and EV4B.

Dear Pei-Lin,

Congratulations on an excellent manuscript! I am very pleased to inform you that it has been accepted for publication in The EMBO Journal. Thank you for comprehensively addressing the initially raised referee concerns and all editorial requests for changes and corrections.

If you have any questions, please do not hesitate to contact the Editorial Office. Thank you for your contribution to The EMBO Journal. Working with you has been a pleasure!

Best regards,

Ioannis
